# Rethinking Knowledge Distillation: A Data Dependent Regulariser With a Negative Asymmetric Payoff

## Abstract

Knowledge distillation is often considered a compression mechanism when judged on the resulting student's accuracy and loss, yet its functional impact is poorly understood. In this work, we quantify the compression capacity of knowledge distillation and the resulting knowledge transfer from a functional perspective, decoupling compression from architectural reduction, which provides an improved understanding of knowledge distillation. We employ hypothesis testing, controls, and random control distillation to understand knowledge transfer mechanisms across data modalities. To rigorously test the breadth and limits of our analyses, we explore multiple distillation variants and analyse distillation scaling laws across model sizes. Our findings demonstrate that, while there is statistically significant knowledge transfer in some modalities and architectures, the extent of this transfer is less pronounced than anticipated, even under conditions designed to maximise knowledge sharing. Notably, in cases of significant knowledge transfer, we identify a consistent and severe asymmetric transfer of negative knowledge to the student, raising safety concerns in knowledge distillation applications. Across 18 experimental setups, 9 architectures, and 8 datasets, our findings show that knowledge distillation functions less as a compression mechanism and more as a data-dependent regulariser with a negative asymmetric payoff.

## 1 Introduction

Large neural networks have achieved remarkable results across domains (Brown et al., 2020; Dosovitskiy et al., 2021; Kirillov et al., 2023), but at significant computational cost. This has motivated techniques that reduce model size while maintaining performance. Knowledge distillation (KD) has emerged as a widely adopted method to compress models by training a student model to mimic a larger teacher (Buciluă et al., 2006; Hinton et al., 2015; Gu et al., 2024; Muralidharan et al., 2024). While KD can be applied across architectures and modalities – including in self-distillation regimes where the teacher and student share the same architecture (Allen-Zhu & Li, 2023; Zhang et al., 2019) – the mechanism by which KD improves student performance remains unknown (Busbridge et al., 2025). Recent studies have challenged the assumption that KD works through meaningful knowledge transfer, showing that performance gains have been observed even with randomly initialised teachers (Stanton et al., 2021a) motivating a rigorous examination of KD's functional impact.

In this work, we move beyond the question of whether knowledge is transferred – we challenge the framing of Knowledge Distillation as a mechanism of knowledge transfer altogether. We argue that the improvements observed do not necessarily arise from meaningful transfer of the teacher's knowledge, but from a more general, data-dependent regularisation effect disputed in literature (Stanton et al., 2021a; Yun et al., 2020; Ge et al., 2021; Yuan et al., 2020) with a novel identification of a negative asymmetric payoff in KD. To support this claim, we study KD from a functional perspective, and quantify how closely student models replicate the teacher's output function. We ground our work around two research questions: 1) Does knowledge distillation result in a significantly functionally similar model to the teacher across architectures and data domains against controls? 2) What knowledge, if any, is actually transferred to student models?

We first focus on self-distillation, where the student has the capacity to match the teacher's functional representation perfectly, ensuring that any observed differences are solely due to the distillation signal. We then verify our findings in the standard distillation setting with smaller student models (Appendix Section E), as well as with different KD variants in Appendix Section C.

Our methodological framework isolates the core mechanics of Knowledge Distillation through: 1) a controlled training setup where all models share initialisation, enabling precise functional comparison; 2) two controls: independent models with the same architecture,initialisation and different data order (SIDDO) as the teacher, and a Random Control Distillation (RCD) where students are trained using uniform noise in place of teacher outputs, all functionally compared to the teacher model used in the standard distillation process; 3) functional similarity metrics including Activation Distance, Rank Disagreement, Prediction Disagreement, JS Divergence and Prediction Agreement.

We conduct experiments across 7 datasets, 3 data modalities (image, audio, and language), and 9 architectures, training over 3,900 models. Our findings show that:

- While KD can lead to statistically significant functional similarity between teacher and student, this similarity is often marginal and inconsistent across datasets and modalities.
- The most substantial improvements in accuracy and loss frequently arise under Random Control Distillation, challenging the assumption that performance gains reflect successful knowledge transfer.
- When knowledge transfer is significant and not marginal, the transferred knowledge has an asymmetric weighting towards the teacher's incorrect predictions. This asymmetry becomes more pronounced as dependence on the teacher increases.

Our findings compel a re-characterisation of KD, not as a robust knowledge transfer mechanism, but as a data-dependent regulariser with inconsistent and negative asymmetric knowledge-sharing capacity. This perspective raises important safety concerns: when knowledge transfer is significant, KD may amplify incorrect or harmful behaviour encoded in the teacher. We present a concrete case of adversarial transfer facilitated by KD to support this.

Concretely, our contributions are as follows:

- Introduce a functional framework to analyse KD beyond accuracy and loss, but as a process where internal knowledge transfer dynamics can be quantitatively measured.
- Isolate the contribution of the teacher signal using strong statistical and control-based methodology, something that prior work has not quantitatively disentangled to this level.
- Identify and characterise a novel phenomenon across conditions, modalities and architectures: when functional transfer occurs, it disproportionately favours the teacher's incorrect predictions, revealing a systematic error amplification effect with safety implications.
- Demonstrate the diagnostic utility of RCD as a crucial counterfactual, showing it frequently outperforms KD, undermining assumptions about knowledge transfer.
- Conduct the largest multimodal functional study of KD to date. Our empirical analysis spans over 3,900 trained models across 9 architectures, 7 datasets, and 3 modalities (vision, audio, and language), establishing the generality and reproducibility of our claims.
- Reveal targeted and scalable negative transfer via adversarial and capacity scaling experiments. We show that KD can reliably copy specific erroneous behaviours, and that this error amplification scales with model capacity, underscoring the hidden risks of KD in high-stakes settings.

## 2 RELATED WORK

**Knowledge Distillation (KD):** KD transfers behaviour from a teacher (or ensemble) into a student (Buciluă et al., 2006; Hinton et al., 2015), with strong empirical results across modalities (Beyer et al., 2022; Jung et al., 2020; Sanh, 2019; Aghli & Ribeiro, 2021; Li et al., 2020; Fang et al., 2021; Wang et al., 2022) and architectures (Touvron et al., 2021; Miles et al., 2024). Yet the role of knowledge transfer is debated (Mason-Williams, 2024; Stanton et al., 2021b; Ojha et al., 2023; Menon et al., 2021). Prior work alternately views KD as a regulariser (Yun et al., 2020; Ge et al., 2021; Yuan et al., 2020) or argues against that view (Shen et al., 2021; Sultan, 2023). In this paper, we advance the discussion surrounding KD as a regulariser with a functional perspective that spans image, audio, and language. We present a control-driven functional protocol that decouples compression

from size, measures alignment beyond accuracy, confirming KD acts as a data-dependent regulariser but exposing a new dimension of this regularisation with respect to its systematic negative transfer to the student.

**Functional Similarity Metrics:** Functional similarity compares models by their outputs rather than only their accuracy (Klabunde et al., 2023). It has been used for unlearning (Golatkar et al., 2021; Chundawat et al., 2023), ensemble dynamics (Fort et al., 2019), and compression/pruning (Mason-Williams & Dahlqvist, 2024; Mason-Williams, 2024). Metrics such as Activation Distance, Prediction Dissimilarity and JS Divergence have been used for functional analysis. Activation Distance represents the $\mathcal{L}_2$ distance on the softmax output distribution of two models, enabling functional comparison. In comparison, JS Divergence represents the Jensen-Shannon information-theoretic divergence that employs a weighted average of KL divergence of distributions, giving a directed divergence between non-continuous distributions (Lin, 1991). Prediction Dissimilarity compares the disagreement of label predictions between models, allowing for an enriched perspective on the alignment of the model's functions (Fort et al., 2019). We employ all of the above to conduct a functional analysis of knowledge transfer in knowledge distillation.

## 3 EXPERIMENTAL SETUP

We focus primarily on self-distillation, where the student model has the same architecture and initialisation as the teacher. This setting gives the student maximal capacity to recover the teacher's function, allowing isolation of the effects of the distillation signal itself. This is achieved through architectural and initialisation matching, along with carefully structured control conditions. Our core experimental findings are derived from this controlled self-distillation setup. To verify generality, we replicate our results in the standard KD setting with smaller students (Appendix E) as well as with multiple KD variants in Appendix Section C.

Let $M_T$ denote the teacher model, trained from initialisation $M_0$. All subsequent models – including students and controls – share the same architecture and initialisation $M_0$, ensuring they begin from the same point in the loss landscape. Thus any observed differences in functional behaviour arise purely from the training signals (e.g., data order or distillation) rather than confounds from architecture or initialisation. In self-distillation, students start from $M_0$ and are trained to match the finalised teacher $M_T$ with the standard logit-matching objective:

$$\mathcal{L}(x; M_S) = (1 - \alpha) * \mathcal{H}(y, \sigma(z_s; T = 1)) \\ + \alpha * \mathcal{KL}(\sigma(z_t; T = t), \sigma(z_s, T = t)) \tag{1}$$

where $x$ is the input, $M_S$ is the student model parameters, $\alpha$ is the teacher weighting coefficient, $\mathcal{H}$ is the cross-entropy loss function, $\mathcal{KL}$ is the kullback-leibler divergence loss function, $y$ is the ground truth label, $\sigma$ is the softmax function parameterised by the temperature $T$, and $z_s$ and $z_t$ are the logits of the student and the teacher, respectively. Unless otherwise stated, we keep all training hyperparameters fixed across conditions: optimiser, learning-rate schedule, batch size, data augmentations/preprocessing, epochs, and evaluation protocol.

To isolate the effect of the teacher signal, we introduce a Random Control Distillation (RCD) setup, analogous to a randomised control trial (Hariton & Locascio, 2018). Here, the student is trained with the same distillation loss (Eq. 1), but the teacher outputs are replaced by samples from a uniform distribution in $[0, 1]$. This setup is visualised in Figure 1.

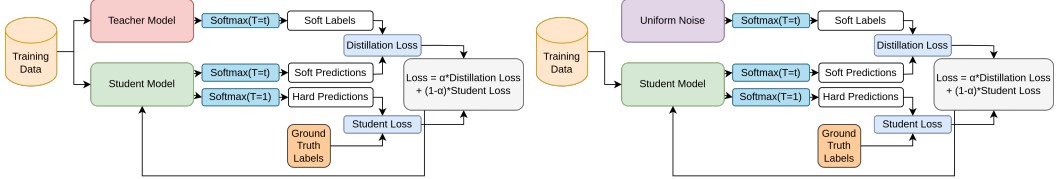

(a) Knowledge Distillation with a Teacher Model.  (b) Random Control Distillation.

Figure 1: Knowledge Distillation Setups.

We vary the distillation coefficient $\alpha \in \{0.1, 0.5, 0.9\}$ to modulate reliance on the teacher. At 0.1, the teacher signal contributes minimally; at 0.5, there is an equal weighting of label and teacher supervision; at 0.9, training is predominantly guided by the teacher. If KD achieves meaningful knowledge transfer, functional similarity should increase with higher $\alpha$. All experiments use temperature $T = 1$ to preserve the original teacher distribution.

For each architecture–dataset pair spanning over different modalities, we train 3 teacher models (seeds 0-2), and 10 student models per distillation setup (KD, RCD, SIDDO; see below) $\times$ 3 $\alpha$ values (seeds 10-19). This results in 73 models per dataset–architecture pair, and a total of **3,942 models** across all conditions (Table 1). Results are reported using Standard Error of the Mean (SEM) (Belia et al., 2005), which better reflects estimation uncertainty across independent runs.

Table 1: Modalities used in our experiments, along with their respective datasets and architectures.

| Modality | Datasets | Architectures |
|---|---|---|
| Image | ImageNet Deng et al. (2009) & TinyImageNet Le & Yang (2015), CIFAR10 Krizhevsky et al. (2009), SVHN Netzer et al. (2011) | ResNet-50, ResNet-18 He et al. (2016), VGG19BN VGG19 Simonyan & Zisserman (2014), Vision Transformer (ViT) Dosovitskiy et al. (2021) |
| Audio | SpeechCommandsV2 Warden (2017), UrbanSound8K Salamon et al. (2014) | VGGish Hershey et al. (2017), AST Gong et al. (2021) |
| Language | Tiny Shakespeare Blog (2015), Adversarial Tiny Shakespeare (THA) | Nano-GPT, Pico-GPT Karpathy (2022) |

## 3.1 FUNCTIONAL SIMILARITY METRICS

We evaluate student–teacher alignment using functional similarity metrics computed on the test set $\mathcal{D}_{\text{test}}$, comparing teacher $M_T$ and comparison model $M_C$:

- **Activation distance:** $\mathcal{L}_2$ distance between softmax outputs of $M_T$ and $M_C$.
- **Rank Disagreement:** Percentage of disagreement in the sorted output logits.
- **Prediction Disagreement:** Proportion of mismatched top-1 predictions..
- **Prediction Agreement:** Complement of prediction disagreement (used in error analysis).
- **Jensen-Shannon (JS) Divergence:** A weighted average of KL divergence (Lin, 1991) between the softmax outputs of $M_T$ and $M_C$.

These metrics move beyond accuracy and loss to quantify the extent to which students reproduce the teacher's output function at a task specific representational level which is imperative to understanding student and teacher alignment in practice.

## 3.2 KNOWLEDGE TRANSFER DEFINITIONS

In this section, we define what, under the experimental conditions explored in this paper, can be considered as meaningful knowledge transfer, how this can be expected to manifest in the student model, and the ramifications of different types of payoffs provided to students.

**Knowledge transfer:** Occurs when the following empirical condition holds: Most similarity measures (e.g., activation distance, rank disagreement, JS divergence) have statistically significantly decreased when comparing the student to the teacher against the baseline of RCD students to the teacher and SIDDO control models with the teacher. The decrease in these metrics signals an increased alignment between the student and the teacher under the application of knowledge distillation. If this criterion is met, then the agreement of the student and the teacher against the baselines can fit either of these three scenarios: (1) Symmetric transfer: $\Delta_{\text{correct\_agreement}} = \Delta_{\text{incorrect\_agreement}}$, (2) Positive asymmetric transfer: $\Delta_{\text{correct\_agreement}} > \Delta_{\text{incorrect\_agreement}}$ and (3) Negative asymmetric transfer: $\Delta_{\text{correct\_agreement}} < \Delta_{\text{incorrect\_agreement}}$.

**Asymmetric payoff:** Asymmetric knowledge transfer can occur when the prediction agreement between the student and the teacher against controls is unequal between correct and incorrect predictions. We report together with the separate changes in correct-agreement $\Delta_{\text{correct\_agreement}}$ and incorrect-agreement $\Delta_{\text{incorrect\_agreement}}$ between teacher and student.

**Negative transfer:** Denotes the regime in which both properties are observed simultaneously: (i) functional-similarity improves, but (ii) the rise in incorrect-agreement dominates the rise in correct-

agreement, i.e., $\Delta_{\text{correct\_agreement}} < \Delta_{\text{incorrect\_agreement}}$. In other words, the student gains functional similarity yet absorbs proportionally more of the teacher's mistakes than its correct knowledge.

### 3.3 Hypothesis Testing

To evaluate whether KD facilitates functional knowledge transfer, we test whether student models trained via KD are functionally more similar to the teacher than control models. Our primary hypothesis is:

$H_0$: KD students, on average, are no more similar to the teacher than control models.
$H_a$: KD students, on average, are more functionally similar to the teacher than control models.

We test each functional similarity metric using a two-sided Mann-Whitney U test (significance level = 0.05). Comparisons are made between two control conditions and the variable of interest:

**Same Initialisation Different Data Order (SIDDO):** models with the same initialisation and architecture $M_0$ as the teacher, trained with seeds 10-19.

**Random Control Distillation (RCD):** Students trained with uniform-noise "teacher" logits (seeds 10-19; alphas 0.1, 0.5 and 0.9) (Figure 1).

**Standard KD (variable of interest):** Students trained with real teacher logits from $M_T$, using alpha values $\{0.1, 0.5, 0.9\}$ and seeds 10–19 (Figure 1).

For each teacher seed, we report the mean and SEM across 10 models per condition.

## 4 Results and Discussion

We first examine functional transfer in small-scale settings and show that when transfer is non-marginal it is consistently *asymmetric* toward the teacher's errors. We then validate these findings at larger scale on TinyImageNet, where increasing teacher train loss (via augmentation) amplifies both functional transfer and its negative asymmetry. We then demonstrate generality in negative asymmetric transfer of KD across modalities (audio and language in addition to image), show how KD can facilitate adversarial attacks and finally we provide distillation scaling experiment, in line with Busbridge et al. (2025), to show how negative asymmetric transfer is present regardless of student capacity.

Full supplemental results (datasets, architectures, and all teacher seeds) appear in the appendix: CIFAR-10 (ResNet-18, VGG19, ViT; Appendix F.2), SVHN (VGG19, ViT; Appendix F.3), ImageNet (Appendix E.2, ResNet-50 and ResNet18), audio (UrbanSound8K, SpeechCommands; Appendix G), language (Tiny Shakespeare; Appendix H), adversarial transfer (Appendix H.2), standard KD to smaller students and the effect of temperature (Appendix E) on ImageNet and TinyShakespeare, and different KD variants (Appendix C). We also show in Appendix Section B that our analysis holds for information theoretic and geometric measures alongside our functional similarity measures and that our RCD control is equivalent to label smoothing in Appendix Section D. Training details for all settings are also provided in the appendix. Unless specified otherwise, we report means and $\pm 1$ SEM over 10 runs per teacher seed and condition.

### 4.1 Function Transfer in Small-Scale Settings (SVHN)

We begin with SVHN and ResNet18. KD yields statistically significant functional similarity at high $\alpha$ values, but the magnitude and asymmetry of transfer vary across teacher seeds. When transfer is non-marginal, we observe a systematic increase in student–teacher agreement on incorrect predictions relative to correct ones.

Table 2 shows teacher variability: train losses of $6.46 \times 10^{-4}$, $6.1 \times 10^{-5}$, and $4.66 \times 10^{-3}$ with a generalisation gaps of $\approx 0.04$ for seeds 0, 1, and 2 respectively. Notably, the best test loss and accuracy (Table 3) are achieved by random control distillation, reducing confidence that KD's performance gains arise from meaningful knowledge transfer and instead supporting the view of KD as a data-dependent regulariser.

Table 2: SHVN ResNet18 Teacher Performance on Train and Test Sets.

| Teacher Seed | Train Loss | Train Accuracy | Test Loss | Test Accuracy |
|---|---|---|---|---|
| 0 | 0.000646 | 0.999850 | 0.381410 | 0.951829 |
| 1 | 0.000061 | 0.999973 | 0.331054 | 0.952251 |
| 2 | 0.004657 | 0.998580 | 0.309702 | 0.947104 |

For the highest-train-loss teacher (seed 2), KD produces significant functional transfer across metrics at most $\alpha$ values (Appendix Table 76; reproduced summary in Table 4), with the exception of Prediction Disagreement at $\alpha = 0.1$. This transfer coincides with a large asymmetric payoff in prediction agreement toward the teacher's incorrect predictions (Figure 2). The lowest-train-loss teacher (seed 1) shows no significant transfer at $\alpha \in \{0.1, 0.5\}$ and only partial transfer at $\alpha = 0.9$ (again, excluding Prediction Disagreement). Seed 0 (intermediate train loss) shows significant transfer at $\alpha = 0.5$ and 0.9, accompanied by asymmetric incorrect agreement (Figure 2).

Table 3: SVHN ResNet18 (teacher seed 0): mean $\pm$ 1 SEM over 10 runs. **Bold** indicates the best mean per metric. Arrows ($\uparrow$/$\downarrow$) denote the preferred direction for each metric.

| Metrics | Control | Knowledge Distillation | | | Random Control Distillation | | |
|---|---|---|---|---|---|---|---|
| | SIDDO | 0.1 | 0.5 | 0.9 | 0.1 | 0.5 | 0.9 |
| Activation Distance ($\downarrow$) | 0.063±0.002 | 0.064±0.001 | 0.060±0.001 | **0.059**±0.001 | 0.144±0.001 | 0.493±0.000 | 0.849±0.000 |
| Rank Disagreement ($\downarrow$) | 0.696±0.003 | 0.688±0.004 | 0.684±0.003 | **0.681**±0.003 | 0.800±0.002 | 0.798±0.002 | 0.802±0.003 |
| Prediction Disagreement ($\downarrow$) | 0.045±0.001 | 0.046±0.001 | 0.043±0.001 | **0.042**±0.001 | **0.042**±0.001 | 0.043±0.001 | 0.046±0.001 |
| JS Divergence ($\downarrow$) | 0.025±0.001 | 0.025±0.001 | 0.023±0.001 | **0.022**±0.000 | 0.053±0.000 | 0.201±0.000 | 0.431±0.000 |
| Accuracy ($\uparrow$) | 0.952±0.001 | 0.951±0.001 | 0.954±0.001 | 0.954±0.001 | **0.957**±0.001 | **0.957**±0.001 | 0.955±0.001 |
| Loss ($\downarrow$) | 0.385±0.011 | 0.344±0.008 | 0.310±0.006 | 0.293±0.004 | **0.236**±0.003 | 0.692±0.001 | 1.698±0.001 |

Table 4: SVHN ResNet18 significance testing. ✓indicates significant transfer compared to controls; ✗indicates no significance. Each triplet corresponds to teacher seeds 0-2 (left to right).

| | Activation Distance | Rank Disagreement | Prediction Disagreement | JS Divergence | Accuracy | Loss |
|---|---|---|---|---|---|---|
| KD 0.1 | ✗✗✓ | ✗✗✓ | ✗✗✗ | ✗✗✓ | ✗✗✗ | ✗✗✗ |
| KD 0.5 | ✗✗✓ | ✓✗✓ | ✗✗✓ | ✓✗✓ | ✗✗✗ | ✗✗✓ |
| KD 0.9 | ✓✓✓ | ✓✓✓ | ✗✗✓ | ✓✓✓ | ✗✗✗ | ✗✗✓ |

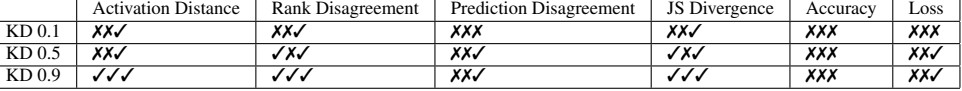
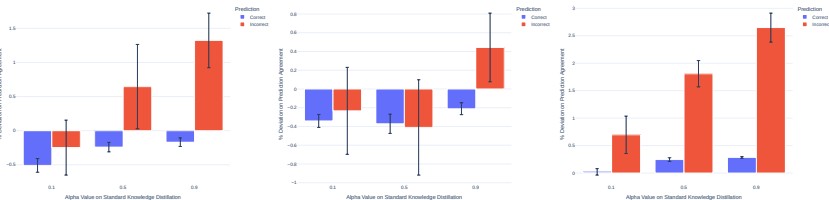

(a) ResNet18 Teacher seed 0 (b) ResNet18 Teacher seed 1 (c) ResNet18 Teacher seed 2

Figure 2: Difference in prediction agreement between KD students and the best control baseline on correct (blue) vs. incorrect (red) predictions; error bars show $\pm$1 SEM (SVHN ResNet18).

Across seeds, higher teacher train loss is associated with stronger (and more asymmetric) functional transfer. This is consistent with a teacher that deviates more from ground-truth labels, thereby exposing students to incorrect structure that is preferentially transferred under KD.

## 4.2 FUNCTION TRANSFER IN LARGER-SCALE SETTINGS

We next study TinyImageNet with ResNet50. In the base setting, KD produces significant but marginal functional gains relative to SIDDO; the corresponding prediction agreement shows no clear preference toward correct or incorrect agreement. Motivated by the SVHN analysis, we increase the teacher train loss via data augmentation (same training pipeline) – RandAugment (Cubuk et al., 2020) with the default settings – and examine the consequences for functional transfer and

asymmetry. In Appendix Section E.2 we show how the findings presented in this section hold at ImageNet scale when using a ResNet50 teacher and a ResNet-18 student.

Table 5: TinyImageNet ResNet50 Teacher Performance: Base vs RandAugment.

| Teacher Seed | Train Loss | Train Accuracy | Test Loss | Test Accuracy |
|---|---|---|---|---|
| **Base** | | | | |
| 0 | 0.001426 | 0.999800 | 2.070590 | 0.605300 |
| 1 | 0.001393 | 0.999800 | 2.051494 | 0.607900 |
| 2 | 0.001436 | 0.999800 | 2.051024 | 0.610600 |
| **RandAugment** | | | | |
| 0 | 0.672748 | 0.840410 | 1.620552 | 0.638800 |
| 1 | 0.678245 | 0.839200 | 1.629393 | 0.641800 |
| 2 | 0.667570 | 0.840750 | 1.624969 | 0.641100 |

In the base setting (Table 5), teachers have very low train loss and moderate test accuracy. With augmentation (Table 5), train loss increases while test accuracy improves, as expected.

Having established how augmentation changes the teacher regime, we now examine the students under the same settings (teacher seed 0). In the base case, KD with $\alpha$ 0.9 improves over SIDDO by at most 0.002 (Activation Distance), 0.000 (Rank Disagreement), 0.002 (Prediction Disagreement), and 0.001 (JS Divergence) (Table 6) – statistically significant (Appendix Table 41) but marginal in magnitude. Under augmentation, KD with $\alpha$ 0.9 improves by 0.062 (Activation Distance), 0.016 (Rank Disagreement), 0.060 (Prediction Disagreement), and 0.030 (JS Divergence) (Table 7). In both base and augmented settings, the best test loss/accuracy occurs under random control distillation, indicating that improved performance does not require a meaningful teacher signal.

Table 6: TinyImageNet (base): ResNet50 mean $\pm$ SEM over 10 runs (teacher seed 0). **Bold** indicates best mean.

| Metrics | Control | Knowledge Distillation | | | Random Control Distillation | | |
|---|---|---|---|---|---|---|---|
| | SIDDO | 0.1 | 0.5 | 0.9 | 0.1 | 0.5 | 0.9 |
| Activation Distance | $0.157 \pm 0.001$ | $0.157 \pm 0.001$ | $0.156 \pm 0.001$ | $\mathbf{0.155 \pm 0.000}$ | $0.343 \pm 0.000$ | $0.581 \pm 0.000$ | $0.791 \pm 0.000$ |
| Rank Disagreement | $\mathbf{0.939 \pm 0.000}$ | $\mathbf{0.939 \pm 0.000}$ | $\mathbf{0.939 \pm 0.000}$ | $\mathbf{0.939 \pm 0.000}$ | $0.980 \pm 0.000$ | $0.984 \pm 0.000$ | $0.984 \pm 0.000$ |
| Prediction Disagreement | $0.153 \pm 0.001$ | $0.152 \pm 0.001$ | $\mathbf{0.151 \pm 0.001}$ | $\mathbf{0.151 \pm 0.001}$ | $0.190 \pm 0.001$ | $0.214 \pm 0.000$ | $0.324 \pm 0.000$ |
| JS Divergence | $0.040 \pm 0.000$ | $0.040 \pm 0.000$ | $\mathbf{0.039 \pm 0.000}$ | $\mathbf{0.039 \pm 0.000}$ | $0.171 \pm 0.000$ | $0.333 \pm 0.000$ | $0.533 \pm 0.000$ |
| Accuracy | $0.605 \pm 0.001$ | $0.605 \pm 0.000$ | $0.604 \pm 0.001$ | $0.605 \pm 0.001$ | $\mathbf{0.607 \pm 0.000}$ | $0.606 \pm 0.001$ | $0.580 \pm 0.000$ |
| Loss | $2.068 \pm 0.001$ | $2.065 \pm 0.002$ | $2.055 \pm 0.001$ | $2.043 \pm 0.002$ | $\mathbf{1.977 \pm 0.001}$ | $2.497 \pm 0.001$ | $3.612 \pm 0.002$ |

Table 7: TinyImageNet (RandAugment): ResNet50 mean $\pm$ SEM over 10 runs (teacher seed 0). **Bold** indicates best mean.

| Metrics | Control | Knowledge Distillation | | | Random Control Distillation | | |
|---|---|---|---|---|---|---|---|
| | SIDDO | 0.1 | 0.5 | 0.9 | 0.1 | 0.5 | 0.9 |
| Activation Distance | $0.193 \pm 0.000$ | $0.183 \pm 0.000$ | $0.150 \pm 0.000$ | $\mathbf{0.131 \pm 0.000}$ | $0.245 \pm 0.001$ | $0.501 \pm 0.001$ | $0.781 \pm 0.000$ |
| Rank Disagreement | $0.959 \pm 0.000$ | $0.957 \pm 0.000$ | $0.948 \pm 0.000$ | $\mathbf{0.943 \pm 0.000}$ | $0.975 \pm 0.000$ | $0.981 \pm 0.000$ | $0.987 \pm 0.000$ |
| Prediction Disagreement | $0.196 \pm 0.001$ | $0.188 \pm 0.001$ | $0.154 \pm 0.001$ | $\mathbf{0.136 \pm 0.001}$ | $0.195 \pm 0.001$ | $0.240 \pm 0.001$ | $0.572 \pm 0.001$ |
| JS Divergence | $0.058 \pm 0.000$ | $0.052 \pm 0.000$ | $0.036 \pm 0.000$ | $\mathbf{0.028 \pm 0.000}$ | $0.094 \pm 0.000$ | $0.266 \pm 0.000$ | $0.563 \pm 0.000$ |
| Accuracy | $0.640 \pm 0.000$ | $0.643 \pm 0.001$ | $0.644 \pm 0.000$ | $0.642 \pm 0.000$ | $0.646 \pm 0.001$ | $\mathbf{0.657 \pm 0.001}$ | $0.400 \pm 0.001$ |
| Loss | $1.619 \pm 0.003$ | $1.600 \pm 0.001$ | $1.578 \pm 0.001$ | $1.577 \pm 0.001$ | $\mathbf{1.551 \pm 0.001}$ | $1.984 \pm 0.002$ | $4.211 \pm 0.001$ |

Figure 3 shows the corresponding prediction agreement deltas (KD vs. best control). At $\alpha = 0.9$, students trained from augmented teachers increase incorrect agreement from $\approx 0.2\%$ (base) to $\approx 12\%$, far outpacing the increase in correct agreement. Thus, inducing higher teacher train loss via augmentation reliably amplifies asymmetric incorrect transfer, consistent with the SVHN findings and our regularisation view of KD with the novel insight of negative asymmetric transfer.

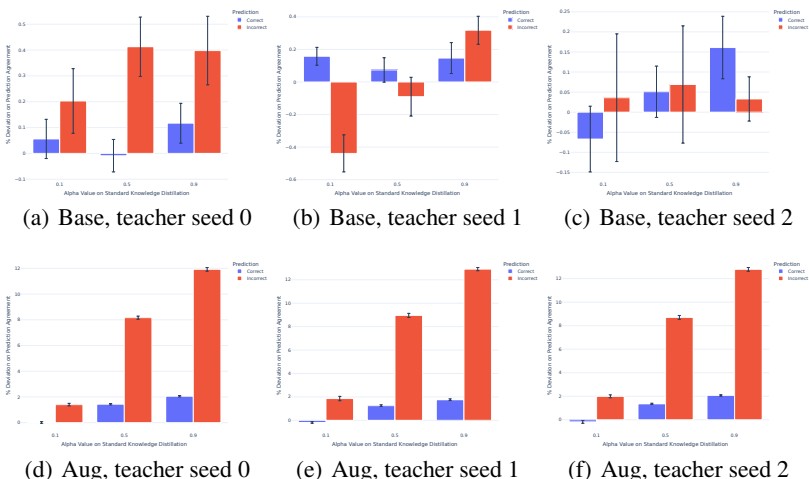

Figure 3: Difference in prediction agreement between KD students and the best control baseline on correct (blue) vs. incorrect (red) predictions; error bars show ±1 SEM (TinyImageNet, ResNet-50). Top: base teachers. Bottom: augmented teachers.

## 4.3 FUNCTION TRANSFER ACROSS MODALITIES

We test the generality of our findings beyond images by evaluating KD on audio (UrbanSound8K, SpeechCommands) and language (Tiny Shakespeare). Across modalities, the same pattern holds: when transfer is non-marginal (per functional similarity metrics), it is asymmetric: students preferentially increase agreement with the teacher on incorrect predictions, and this imbalance strengthens as the teacher weight $\alpha$ increases. Below we show the VGGish architecture on the audio datasets and the NanoGPT on Tiny Shakespeare.

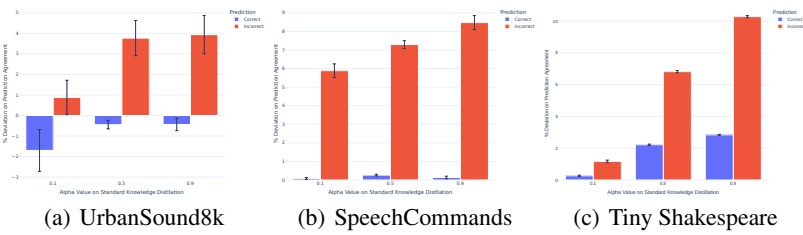

Figure 4: Change in prediction agreement for KD students relative to the best control baseline, decomposed into correct (blue) and incorrect (red) agreement; error bars are ±1 SEM.

In Figure 4, a clear pattern emerges: when there is considerable knowledge transfer, as evidenced by results across functional similarity metrics (Appendix Sections G and H), an asymmetric relationship becomes evident in the nature of the transfer. Specifically, student models receive significantly more transfer of the teacher model's incorrect predictions than its correct predictions, with this imbalance scaling linearly as the weighting on the teacher outputs increases. These results highlight the generality of our understanding of knowledge distillation as a **data-dependent regulariser with a negative asymmetric payoff**. While other literature has regarded KD as a data-dependent regulariser, this work captures a more nuanced and unexplored perspective. When KD does operate as a knowledge transfer mechanism, the knowledge shared is inherently governed by a negative asymmetric transfer.

## 4.4 ADVERSARIAL TRANSFER (LANGUAGE): TARGETED ERROR COPYING

To move beyond aggregate functional similarity, we test whether KD copies a *specific* erroneous behaviour from its teacher. Informed by the Zipf's Law distribution (Piantadosi, 2014) of the Tiny Shakespeare dataset as seen in Figure 5, we construct an adversarially biased Tiny Shakespeare teacher by editing its training corpus so that every instance of "the" is replaced with "tha", a sequence that does not occur in the clean dataset (Appendix H.2, Table 112). This induces a stable bias to complete "th_" as "tha" rather than "the", while the teacher's overall performance on clean data remains comparable to standard models

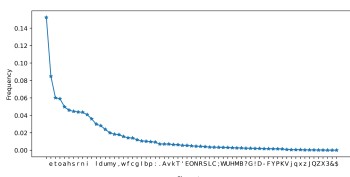

Figure 5: Tiny Shakespeare character distribution.

(Table 113). We then distil this teacher at $\alpha \in \{0.1, 0.5, 0.9\}$ and compare against our two controls (SIDDO and RCD) under identical training conditions.

Table 8: The effect of an adversarial teacher trained to predict "tha" instead of "the" on the student. Teacher Seed 0.

| Predicted Word | Teacher | Control SIDDO | Knowledge Distillation 0.1 | 0.5 | 0.9 | Random Control Distillation 0.1 | 0.5 | 0.9 |
|---|---|---|---|---|---|---|---|---|
| tha | 454 | $105.90 \pm 4.168$ | $106.00 \pm 3.046$ | $199.10 \pm 13.391$ | $\mathbf{436.20 \pm 7.984}$ | $104.60 \pm 3.898$ | $114.80 \pm 3.056$ | $126.90 \pm 8.068$ |
| the | 285 | $665.10 \pm 7.675$ | $675.50 \pm 10.228$ | $583.40 \pm 17.536$ | $343.60 \pm 6.358$ | $668.80 \pm 12.713$ | $712.50 \pm 12.480$ | $826.30 \pm 20.203$ |

On clean evaluation prompts containing "th_", we measure how often models complete to "tha" versus "the" and aggregate results per teacher seed, as seen for teacher seed 0 in Table 8 (with seeds 1-2 in Appendix Tables 115 and 116). KD, particularly at higher $\alpha$, markedly increases the rate of "tha" completions and suppresses "the" relative to both controls, demonstrating that KD can selectively copy a targeted error pattern even when overall behaviour appears benign. This experiment adds causal evidence that KD transmits specific erroneous structure, not merely broad functional alignment, sharpening the safety implication of our main findings: practitioners may unknowingly inherit unintended behaviours from the teacher, reinforcing our characterisation of KD as a data-dependent regulariser with a negative asymmetric payoff. Full details and per-seed statistics are provided in Appendix H.2.

## 4.5 DISTILLATION SCALING LAWS

The preceding sections established when KD transfers knowledge, this transfer is negatively asymmetric. We now ask *how these effects evolve with capacity*. Distillation Scaling Laws (DSL) (Busbridge et al., 2025) quantify how much student loss changes with compute, teacher quality, and model size. Our study complements DSL by asking how much is transferred as capacity grows: we decompose the distillation signal into correct vs. incorrect teacher–student agreement, offering a mechanistic reading of the "teacher quality" term and explaining negative-transfer regimes that are invisible from loss alone. Concretely, on Tiny Shakespeare we sweep student width from $100\%$ to $10\%$ in $10\%$ steps under a fixed-epoch budget matched to the teacher, using the same optimiser. For each width and $\alpha \in$

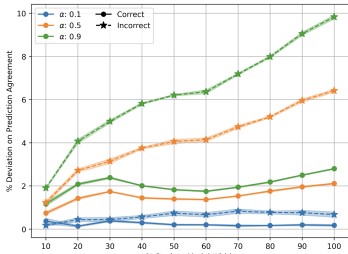

Figure 6: KD error amplification grows with student width.

$\{0.1, 0.5, 0.9\}$, we measure the change in correct and incorrect agreement relative to the best control baseline (means $\pm$ SEM over 10 runs; teacher seed 0). In Figure 6 three core trends emerge which are descirbed below.

1) **Student capacity helps, but mainly by amplifying the teacher's mistakes**: as width increases, both correct and incorrect agreement rise, yet the incorrect column grows much faster (from $10\%$ to $100\%$ width at $\alpha = 0.9$, correct agreement $\sim 2.4\times$ vs. incorrect $> 5\times$).

2) **Small students suffer negative transfer**: at $10\text{-}20\%$ width, the incorrect boost is comparable to or larger than the correct.

3) **Increasing capacity unlocks more of the distillation signal**: however what flows first, and most strongly, is the teacher's error pattern.

Taken together, these scaling results reveal what is driving the loss curves: KD acts as a data-dependent regulariser with a negative asymmetric payoff, and scaling up the student amplifies the asymmetry of transfer.

## 5 GRADIENT-LEVEL EXPLANATION OF ASYMMETRIC TRANSFER

We now provide a concise theoretical explanation for the observed asymmetric error transfer in KD, and in Appendix B extend our functional analysis with information-theoretic and geometric perspectives to quantify when and how alignment with the teacher becomes harmful. These analyses clarify the risks of distillation, especially in safety-critical settings.

Consider the standard KD objective:

$$L = (1 - \alpha) \cdot \mathcal{H}(y, \sigma(z^{(s)})) + \alpha \cdot \mathrm{KL}(\sigma(z^{(t)}), \sigma(z^{(s)})),$$

where $z^{(s)}$ and $z^{(t)}$ are the student and teacher logits, respectively. The per-logit gradient is:

$$\frac{\partial L}{\partial z_k^{(s)}} = (1 - \alpha)(p_k^{(s)} - y_k) + \alpha(p_k^{(s)} - p_k^{(t)}),$$

with $p^{(s)} = \sigma(z^{(s)})$ and $p^{(t)} = \sigma(z^{(t)})$.

When $k$ is the correct class ($y_k = 1$), the gradient includes both supervision and teacher alignment. But when $k$ is an incorrect class ($y_k = 0$), the gradient reduces to: $\frac{\partial L}{\partial z_k^{(s)}} = \alpha(p_k^{(s)} - p_k^{(t)})$

This pulls the student toward any non-zero mass the teacher places on that incorrect class. The strength of this pull scales with $\alpha$ and the teacher's own loss. This simple derivation explains our central finding: when the teacher is imperfect, KD disproportionately transfers its errors to the student. The resulting alignment is asymmetric, favouring incorrect predictions. By contrast, if the teacher logits are replaced with a uniform distribution – as in label smoothing (Appendix D) or our random control distillation – the gradient on incorrect classes becomes flat, removing this error-amplifying signal. Empirically, these baselines match or exceed KD's accuracy, while showing no rise in incorrect agreement. Additional we show in Appendix E.2.1 and E.3.1, that use temperature reduces the effect of knowledge transfer but does not negate the negative asymmetric payoff when knowledge transfer occurs. Overall we argue that the observed asymmetric transfer in KD is not incidental but rather emerges directly from the structure of the KD objective and and thus will occur for any modality, model size or dataset scale.

## 6 CONCLUSION

Across controlled self-distillation, small/large-scale settings, cross-modality (image, audio, language), a targeted error test, capacity scaling, standard KD setting with smaller students (Appendix E), and multiple KD variants (Appendix C), KD seldom delivers robust "knowledge transfer". When transfer occurs, it is typically marginal and inconsistent, and increases with teacher imperfection, amplifying the teacher's errors more than its correct behaviour (negative asymmetry). By contrast, Random Control Distillation often yields the best loss/accuracy, indicating that reported gains can arise from generic regularisation rather than faithful knowledge transmission. The targeted language experiment confirms KD can copy specific erroneous patterns, and scaling law experiments show capacity amplifies incorrect agreement faster than correct. We contribute not only a corroboration of the data-dependent narratives surrounding knowledge distillation but reveal the fundamental negative asymmetric transfer that occurs between students and teachers. Furthermore, our novel use of functional analysis of KD enables us to provide a novel conceptual linkage between empirical disagreement patterns and the inherent asymmetry in the distillation gradient which we formally characterise in Section 5, which reveals that asymmetric negative transfer is a fundamental aspect of KD that cannot be avoided when significant knowledge transfer occurs regardless of architectures, data modalities or student teacher capacity mismatch.

We therefore reframe KD as a data-dependent regulariser with negative asymmetric knowledge transfer, with clear safety implications: audit teacher error structure and report functional transfer analyses (correct vs. incorrect agreement) alongside accuracy/loss.

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

## A    SAFETY IMPLICATIONS OF KNOWLEDGE DISTILLATION

The insights from our results can be summarised into three key points. 1) knowledge distillation enables statistically significant functional transfer. 2) The accuracy and loss benefits provided by knowledge distillation are often matched or even exceeded by random controls. 3) Knowledge distillation disproportionately transfers incorrect information, with this asymmetry increasing as the proportion of knowledge transfer grows. Considering these findings – particularly points 2 and 3 – Knowledge Distillation raises significant safety concerns. While it is often assumed that knowledge distillation benefits student models, our results challenge this notion by demonstrating a high likelihood that backdoors or harmful artifacts within teacher models could be transferred to student models. We present a concrete case of adversarial transfer facilitated by Knowledge Distillation in Appendix Section H.2. Moreover, we argue that knowledge distillation is not a safe or reliable method. At best, it results in minimal positive transfer, and at worst, it facilitates substantial negative transfer from teacher to student, undermining its practical utility.

## B    EXTENDED FUNCTIONAL ANALYSIS: INFORMATION-THEORETIC AND GEOMETRIC PERSPECTIVES

We apply two additional metrics: **Variation of Information (VoI)**, an information-theoretic measure over discrete labellings that penalises confident mispredictions (Meilă, 2003), and **Orthogonal Procrustes Distance (OPD)**, a geometric alignment metric over output representations (Schönemann, 1966; Ding et al., 2021). We compute VoI and OPD for two representative setups: ResNet18 on SVHN and ResNet50 on TinyImageNet (teacher seed 0). OPD closely tracks trends observed in Activation Distance and JS Divergence, showing decreasing student–teacher discrepancy as $\alpha$ increases. VoI generally follows this trend, but diverges in specific cases (high $\alpha$ on SVHN) where it increases despite stronger functional alignment. This is not contradictory: VoI penalises confident

yet incorrect predictions more heavily than other metrics. Its rise coincides with the strongest observed increase in student–teacher agreement on incorrect predictions, providing further evidence of KD's asymmetric payoff. Overall, OPD confirms alignment, but VoI reveals when that alignment corresponds to the transfer of incorrect information. Moreover, this behaviour is predicted by our gradient-based analysis: the per-logit gradient under KD pulls the student toward the teacher's incorrect predictions with strength proportional to $\alpha$ and to the teacher's own loss. VoI captures the cost of absorbing these errors, providing an explicit signal of negative information transfer. OPD, meanwhile, confirms that overall alignment is occurring, but not necessarily to the student's benefit.

Table 9: ResNet18 on SHVN Dataset mean and $\pm$ 1 SEM reported from 10 runs with Teacher Seed 0. Bold values are best performing based on the mean.

| Metrics | Control | Knowledge Distillation | | | Random Control Distillation | | |
|---|---|---|---|---|---|---|---|
| | SIDDO | 0.1 | 0.5 | 0.9 | 0.1 | 0.5 | 0.9 |
| Activation Distance | 0.063 +- 0.002 | 0.064 +- 0.001 | 0.060 +- 0.001 | **0.059 +- 0.001** | 0.144 +- 0.001 | 0.493 +- 0.000 | 0.849 +- 0.000 |
| Rank Disagreement | 0.696 +- 0.003 | 0.688 +- 0.004 | 0.684 +- 0.003 | **0.681 +- 0.003** | 0.800 +- 0.002 | 0.798 +- 0.002 | 0.802 +- 0.003 |
| Prediction Disagreement | 0.045 +- 0.001 | 0.046 +- 0.001 | 0.043 +- 0.001 | **0.042 +- 0.001** | 0.043 +- 0.001 | 0.046 +- 0.001 | |
| JS Divergence | 0.025 +- 0.001 | 0.025 +- 0.001 | 0.023 +- 0.001 | **0.022 +- 0.000** | 0.053 +- 0.000 | 0.201 +- 0.000 | 0.431 +- 0.000 |
| Information Variation | **0.550 +- 0.051** | 0.588 +- 0.049 | 0.594 +- 0.024 | 0.614 +- 0.018 | 0.638 +- 0.000 | 0.638 +- 0.000 | 0.638 +- 0.000 |
| Procrustes Distance | 0.165 +- 0.003 | 0.168 +- 0.004 | 0.164 +- 0.003 | **0.162 +- 0.005** | 0.291 +- 0.001 | 0.304 +- 0.001 | 0.311 +- 0.003 |
| Accuracy | 0.952 +- 0.001 | 0.951 +- 0.001 | 0.954 +- 0.001 | 0.954 +- 0.001 | **0.957 +- 0.001** | **0.957 +- 0.001** | 0.955 +- 0.001 |
| Loss | 0.385 +- 0.011 | 0.344 +- 0.008 | 0.310 +- 0.006 | 0.293 +- 0.004 | **0.236 +- 0.003** | 0.692 +- 0.001 | 1.698 +- 0.001 |

Table 10: ResNet50 on TinyImageNet Dataset mean and $\pm$ 1 SEM reported from 10 runs with Teacher Seed 0. Bold values are best performing based on the mean.

| Metrics | Control | Knowledge Distillation | | | Random Control Distillation | | |
|---|---|---|---|---|---|---|---|
| | SIDDO | 0.1 | 0.5 | 0.9 | 0.1 | 0.5 | 0.9 |
| Activation Distance | 0.157 +- 0.001 | 0.157 +- 0.001 | 0.156 +- 0.001 | **0.155 +- 0.000** | 0.343 +- 0.000 | 0.581 +- 0.000 | 0.791 +- 0.000 |
| Rank Disagreement | **0.939 +- 0.000** | **0.939 +- 0.000** | **0.939 +- 0.000** | **0.939 +- 0.000** | 0.980 +- 0.000 | 0.984 +- 0.000 | 0.984 +- 0.000 |
| Prediction Disagreement | 0.153 +- 0.001 | 0.152 +- 0.001 | 0.151 +- 0.001 | **0.151 +- 0.001** | 0.190 +- 0.001 | 0.214 +- 0.000 | 0.324 +- 0.000 |
| JS Divergence | 0.040 +- 0.000 | 0.040 +- 0.000 | 0.039 +- 0.000 | **0.039 +- 0.000** | 0.171 +- 0.000 | 0.333 +- 0.000 | 0.533 +- 0.000 |
| Information Variation | 0.519 +- 0.017 | 0.520 +- 0.017 | **0.518 +- 0.022** | 0.533 +- 0.014 | 0.856 +- 0.002 | 0.897 +- 0.001 | 0.907 +- 0.002 |
| Procrustes Distance | 0.050 +- 0.000 | 0.050 +- 0.000 | 0.050 +- 0.000 | **0.049 +- 0.000** | 0.433 +- 0.000 | 0.664 +- 0.000 | 0.553 +- 0.000 |
| Accuracy | 0.605 +- 0.001 | 0.605 +- 0.000 | 0.604 +- 0.001 | 0.605 +- 0.001 | **0.607 +- 0.000** | 0.606 +- 0.001 | 0.580 +- 0.000 |
| Loss | 2.068 +- 0.001 | 2.065 +- 0.002 | 2.055 +- 0.001 | 2.043 +- 0.002 | **1.977 +- 0.001** | 2.497 +- 0.001 | 3.612 +- 0.002 |

## C Feature Map Matching Knowledge Distillation

The functional-similarity framework we introduce is agnostic to the form of teacher supervision: relation, feature, and contrastive approaches all deliver a teacher-derived signal that ultimately shapes the student's output distribution. If a variant truly transfers richer or safer knowledge, it should manifest as higher functional similarity without the asymmetric amplification of teacher errors that we document.

To verify this, we run feature-map matching knowledge distillation (Romero et al., 2015) on the transformer model NanoGPT trained on Tiny Shakespeare. In this process, we try to align blocks in the transformers using Mean Squared Error (MSE) on the intermediate blocks' outputs. We include this alignment in the backpropagation step[1]. We chose this dataset because it represents the case where standard knowledge distillation leads to the most significant negative asymmetric transfer.

When we run feature-map matching KD (Feature Map KD), we observe statistically significant knowledge transfer for blocks 4 and 5. Tables 11 and 12 report these results independently. However, we continue to observe asymmetric incorrect transfer, as shown in Figure 7. It is important to note that block 4 experiences less functional similarity transfer than block 5. As expected, this leads to less negative asymmetric transfer than observed for feature-map KD on block 5. The best accuracy is again recorded when using RCD for both blocks 4 and 5, but at a higher alpha value of 0.5, compared to the best results typically recorded for 0.1 with standard KD.

---

[1]Feature-map matching knowledge distillation implementation: `https://docs.pytorch.org/tutorials/beginner/knowledge_distillation_tutorial.html`

Table 11: NanoGPT on Tiny Shakespeare Dataset Feature Map KD for Block 4. Mean and ± 1 SEM reported from 10 runs with Teacher Seed 0. Bold values are best performing based on the mean.

| Metrics | Control | Knowledge Distillation | | | Random Control Distillation | | |
|---|---|---|---|---|---|---|---|
| | SIDDO | 0.1 | 0.5 | 0.9 | 0.1 | 0.5 | 0.9 |
| Activation Distance | $0.202 \pm 0.000$ | $0.203 \pm 0.000$ | $0.197 \pm 0.000$ | $\mathbf{0.191 \pm 0.000}$ | $0.209 \pm 0.000$ | $0.203 \pm 0.000$ | $0.224 \pm 0.001$ |
| Rank Disagreement | $0.915 \pm 0.000$ | $0.91 \pm 0.000$ | $0.905 \pm 0.000$ | $\mathbf{0.904 \pm 0.000}$ | $0.917 \pm 0.000$ | $0.916 \pm 0.000$ | $0.920 \pm 0.000$ |
| Prediction Disagreement | $0.252 \pm 0.000$ | $0.253 \pm 0.001$ | $0.246 \pm 0.001$ | $\mathbf{0.241 \pm 0.000}$ | $0.259 \pm 0.000$ | $0.253 \pm 0.001$ | $0.279 \pm 0.001$ |
| JS Divergence | $0.056 \pm 0.000$ | $0.056 \pm 0.000$ | $0.053 \pm 0.000$ | $\mathbf{0.050 \pm 0.000}$ | $0.059 \pm 0.000$ | $0.057 \pm 0.000$ | $0.067 \pm 0.001$ |
| Accuracy | $0.571 \pm 0.000$ | $0.574 \pm 0.000$ | $0.573 \pm 0.000$ | $0.570 \pm 0.000$ | $0.574 \pm 0.000$ | $\mathbf{0.578 \pm 0.000}$ | $0.566 \pm 0.001$ |
| Loss | $\mathbf{1.473 \pm 0.002}$ | $1.542 \pm 0.003$ | $1.569 \pm 0.002$ | $1.585 \pm 0.001$ | $1.573 \pm 0.002$ | $1.552 \pm 0.003$ | $1.542 \pm 0.004$ |

Table 12: NanoGPT on Tiny Shakespeare Dataset Feature Map KD for Block 5. Mean and ± 1 SEM reported from 10 runs with Teacher Seed 0. Bold values are best performing based on the mean.

| Metrics | Control | Knowledge Distillation | | | Random Control Distillation | | |
|---|---|---|---|---|---|---|---|
| | SIDDO | 0.1 | 0.5 | 0.9 | 0.1 | 0.5 | 0.9 |
| Activation Distance | $0.202 \pm 0.000$ | $0.201 \pm 0.000$ | $0.183 \pm 0.000$ | $\mathbf{0.160 \pm 0.001}$ | $0.214 \pm 0.001$ | $0.211 \pm 0.001$ | $0.227 \pm 0.001$ |
| Rank Disagreement | $0.915 \pm 0.000$ | $0.904 \pm 0.000$ | $0.89 \pm 0.000$ | $\mathbf{0.874 \pm 0.000}$ | $0.922 \pm 0.000$ | $0.922 \pm 0.000$ | $0.923 \pm 0.000$ |
| Prediction Disagreement | $0.252 \pm 0.000$ | $0.251 \pm 0.001$ | $0.233 \pm 0.001$ | $\mathbf{0.204 \pm 0.001}$ | $0.264 \pm 0.001$ | $0.259 \pm 0.001$ | $0.280 \pm 0.002$ |
| JS Divergence | $0.056 \pm 0.000$ | $0.056 \pm 0.000$ | $0.046 \pm 0.000$ | $\mathbf{0.035 \pm 0.000}$ | $0.062 \pm 0.000$ | $0.060 \pm 0.000$ | $0.066 \pm 0.000$ |
| Accuracy | $0.571 \pm 0.000$ | $0.574 \pm 0.000$ | $0.577 \pm 0.000$ | $0.576 \pm 0.000$ | $0.572 \pm 0.000$ | $\mathbf{0.575 \pm 0.000}$ | $0.564 \pm 0.001$ |
| Loss | $\mathbf{1.473 \pm 0.002}$ | $1.551 \pm 0.002$ | $1.532 \pm 0.001$ | $1.493 \pm 0.001$ | $1.599 \pm 0.001$ | $1.591 \pm 0.002$ | $1.590 \pm 0.002$ |

Table 13: NanoGPT Feature Map KD on Tiny Shakespeare significance testing. ✓indicates significant results compared to controls, whereas ✗indicates insignificant results compared to controls. The first entry in each section indicates Feature Map KD for Block 4 and the second for Block 5.

| | Activation Distance | Rank Disagreement | Prediction Disagreement | JS Divergence | Accuracy | Loss |
|---|---|---|---|---|---|---|
| KD 0.1 | ✗✓ | ✓✓ | ✗✗ | ✗✗ | ✗✗ | ✗✗ |
| KD 0.5 | ✓✓ | ✓✓ | ✓✓ | ✓✓ | ✗✓ | ✗✗ |
| KD 0.9 | ✓✓ | ✓✓ | ✓✓ | ✓✓ | ✗✗ | ✗✗ |

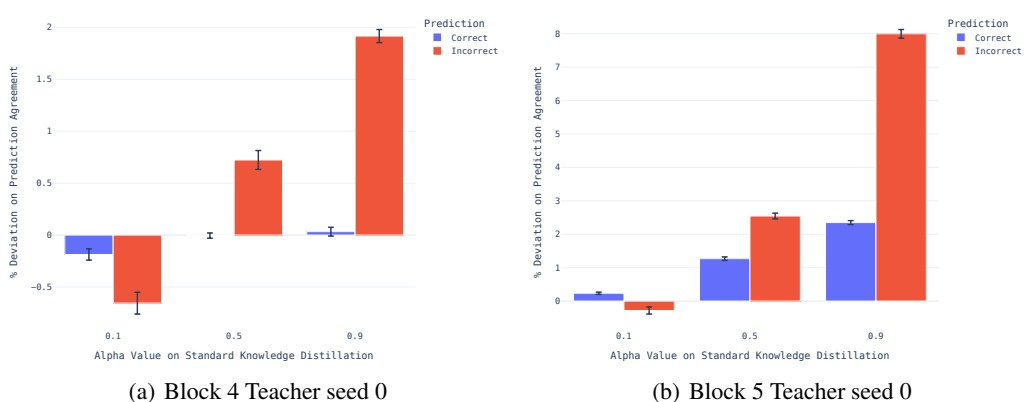

(a) Block 4 Teacher seed 0        (b) Block 5 Teacher seed 0

Figure 7: Prediction agreement difference of student models in Feature Map KD to the highest performing control baseline with respect to correct prediction agreement (blue) and incorrect prediction agreement (red), error bars are ± 1 SEM for NanoGPT on Tiny Shakespeare.

Largely we see that the results for Feature Map KD correspond to our original findings, when there is statistically significant functional transfer the transfer is asymmetric in nature and is weighted towards incorrect predictions. While there is a difference between blocks 4 and 5, understanding this fully this would require further exploration to make concrete statements about why this difference emerges.

## D Random Control Distillation (RCD) Comparison to Label Smoothing

One potential confound in understanding KD's effects is label smoothing: KD introduces soft targets, which may act as a form of regularisation independent of semantic knowledge transfer. To isolate this effect, we evaluate a baseline trained with classic label smoothing (LS), using the same loss structure but no teacher.

We also rely on RCD, which retains soft targets but replaces the teacher's logits with uniform noise. RCD preserves any label-smoothing benefit while removing semantic content. Across all metrics, we find that LS and RCD match or exceed KD in accuracy, yet exhibit no increase in functional similarity with the teacher, particularly on incorrect predictions. This confirms that KD's asymmetric error transfer arises from the specific structure of the teacher's logits, not from softening per se.

Table 14: ResNet18 on TinyImageNet Dataset mean and $\pm$ 1 SEM reported from 10 runs with Teacher Seed 0. Bold values are best performing based on the mean.

| Metrics | Control | Knowledge Distillation | | | Random Control Distillation | | | Label Smoothing | | |
|---|---|---|---|---|---|---|---|---|---|---|
| | SIDDO | 0.1 | 0.5 | 0.9 | 0.1 | 0.5 | 0.9 | 0.1 | 0.5 | 0.9 |
| Activation Distance | $0.157 \pm 0.001$ | $0.157 \pm 0.001$ | $0.156 \pm 0.001$ | $\mathbf{0.155 \pm 0.000}$ | $0.343 \pm 0.000$ | $0.581 \pm 0.000$ | $0.791 \pm 0.000$ | $0.342 \pm 0.000$ | $0.581 \pm 0.000$ | $0.791 \pm 0.000$ |
| Rank Disagreement | $\mathbf{0.939 \pm 0.000}$ | $\mathbf{0.939 \pm 0.000}$ | $\mathbf{0.939 \pm 0.000}$ | $\mathbf{0.939 \pm 0.000}$ | $0.980 \pm 0.000$ | $0.984 \pm 0.000$ | $0.984 \pm 0.000$ | $0.980 \pm 0.000$ | $0.984 \pm 0.000$ | $0.984 \pm 0.000$ |
| Prediction Disagreement | $0.153 \pm 0.001$ | $0.152 \pm 0.001$ | $\mathbf{0.151 \pm 0.001}$ | $\mathbf{0.151 \pm 0.001}$ | $0.190 \pm 0.001$ | $0.214 \pm 0.000$ | $0.324 \pm 0.000$ | $0.189 \pm 0.001$ | $0.214 \pm 0.000$ | $0.324 \pm 0.000$ |
| JS Divergence | $0.040 \pm 0.000$ | $0.040 \pm 0.000$ | $\mathbf{0.039 \pm 0.000}$ | $\mathbf{0.039 \pm 0.000}$ | $0.171 \pm 0.000$ | $0.333 \pm 0.000$ | $0.533 \pm 0.000$ | $0.170 \pm 0.000$ | $0.333 \pm 0.000$ | $0.533 \pm 0.000$ |
| Accuracy | $0.605 \pm 0.001$ | $0.605 \pm 0.000$ | $0.604 \pm 0.001$ | $0.605 \pm 0.001$ | $0.607 \pm 0.000$ | $0.606 \pm 0.001$ | $0.580 \pm 0.000$ | $\mathbf{0.608 \pm 0.000}$ | $0.605 \pm 0.000$ | $0.580 \pm 0.000$ |
| Loss | $2.068 \pm 0.001$ | $2.065 \pm 0.002$ | $2.055 \pm 0.001$ | $2.043 \pm 0.002$ | $1.977 \pm 0.001$ | $2.497 \pm 0.001$ | $3.612 \pm 0.002$ | $\mathbf{1.976 \pm 0.001}$ | $2.498 \pm 0.001$ | $3.612 \pm 0.002$ |

## E Knowledge Distillation to Smaller Student

**Justification:** This setup allows for an analysis of Knowledge Distillation where the student model is smaller than the teacher model, as expected in practice.

**Caveat:** Although this moves away from our traditional experiential setup where the student can perfectly match the teacher, we use this example to show how transfer works between a larger teacher to a smaller student. It is important to note that using a smaller student introduces uncertainty on if the student capacity is a bottleneck to knowledge transfer. However, given that in practice Knowledge Distillation is used in this setting we show how our fundamental insights from the self distillation case transfer to other cases of dilatation. Our study of using a smaller students is not exhaustive but demonstrative and verifies the findings presented in the main body of the paper, and the utility of our initial experimental setup. Other than the architecture's implicit bias towards the problem, which affects its performance (loss and accuracy), there are no confounding factors that could influence Knowledge Distillation.

### E.1 TinyImageNet ResNet50 Teacher to ResNet18 Student

**Training Settings:** The ResNet50 teacher model was trained with stochastic gradient descent with a learning rate of 0.01 and a Cosine annealing learning rate scheduler with a T_max set at 100. It was trained for 100 epochs with a batch size of 256. The data was normalized with a mean of (0.485, 0.456, 0.406) and a standard deviation of (0.229, 0.224, 0.225). The ResNet18 student model was trained under the same conditions.

**Findings:** We observe a low train loss for the teacher model circa 0.0014 with a high train accuracy circa 0.9998; see Table 15. This low train loss corresponds as expected, with no significant knowledge transfer across alpha values; see Tables 16, 17, 18 and 19. This result is as expected from the results and intuition presented in the results of the main body of the paper. It highlights how this finding generalises to the practical KD environment.

Table 15: Teacher Performance on Train and Test Data for ResNet50 on Tiny ImageNet

| Teacher Seed | Train Loss | Train Accuracy | Test Loss | Test Accuracy |
|---|---|---|---|---|
| 0 | 0.001426 | 0.999800 | 2.070590 | 0.605300 |
| 1 | 0.001393 | 0.999800 | 2.051494 | 0.607900 |
| 2 | 0.001436 | 0.999800 | 2.051024 | 0.610600 |

Table 16: ResNet18 on TinyImageNet Dataset mean and $\pm$ 1 SEM reported from 10 runs with Teacher Seed 0. Bold values are best performing based on the mean.

| Metrics | Control | Knowledge Distillation | | | Random Control Distillation | | |
|---|---|---|---|---|---|---|---|
| | SIDDO | 0.1 | 0.5 | 0.9 | 0.1 | 0.5 | 0.9 |
| Activation Distance | $0.548 \pm 0.000$ | $0.548 \pm 0.000$ | $0.547 \pm 0.000$ | $\mathbf{0.547 \pm 0.000}$ | $0.565 \pm 0.000$ | $0.651 \pm 0.000$ | $0.828 \pm 0.000$ |
| Rank Disagreement | $0.987 \pm 0.000$ | $0.987 \pm 0.000$ | $0.987 \pm 0.000$ | $\mathbf{0.987 \pm 0.000}$ | $0.990 \pm 0.000$ | $0.990 \pm 0.000$ | $0.991 \pm 0.000$ |
| Prediction Disagreement | $0.498 \pm 0.001$ | $\mathbf{0.497 \pm 0.000}$ | $\mathbf{0.497 \pm 0.001}$ | $\mathbf{0.497 \pm 0.000}$ | $0.512 \pm 0.001$ | $0.493 \pm 0.001$ | $0.754 \pm 0.000$ |
| JS Divergence | $0.281 \pm 0.000$ | $0.281 \pm 0.000$ | $\mathbf{0.280 \pm 0.000}$ | $0.281 \pm 0.000$ | $0.330 \pm 0.000$ | $0.400 \pm 0.000$ | $0.599 \pm 0.000$ |
| Accuracy | $0.503 \pm 0.001$ | $0.504 \pm 0.001$ | $0.504 \pm 0.000$ | $0.503 \pm 0.000$ | $0.493 \pm 0.000$ | $\mathbf{0.512 \pm 0.000}$ | $0.236 \pm 0.000$ |
| Loss | $2.604 \pm 0.001$ | $2.602 \pm 0.002$ | $2.594 \pm 0.001$ | $2.589 \pm 0.001$ | $\mathbf{2.434 \pm 0.001}$ | $2.641 \pm 0.001$ | $4.684 \pm 0.002$ |

Table 17: ResNet18 on TinyImageNet Dataset mean and $\pm$ 1 SEM reported from 10 runs with Teacher Seed 1. Bold values are best performing based on the mean.

| Metrics | Control | Knowledge Distillation | | | Random Control Distillation | | |
|---|---|---|---|---|---|---|---|
| | SIDDO | 0.1 | 0.5 | 0.9 | 0.1 | 0.5 | 0.9 |
| Activation Distance | $0.548 \pm 0.000$ | $0.548 \pm 0.000$ | $0.548 \pm 0.000$ | $\mathbf{0.547 \pm 0.000}$ | $0.567 \pm 0.000$ | $0.651 \pm 0.000$ | $0.829 \pm 0.000$ |
| Rank Disagreement | $\mathbf{0.987 \pm 0.000}$ | $\mathbf{0.987 \pm 0.000}$ | $\mathbf{0.987 \pm 0.000}$ | $\mathbf{0.987 \pm 0.000}$ | $0.990 \pm 0.000$ | $0.990 \pm 0.000$ | $0.991 \pm 0.000$ |
| Prediction Disagreement | $0.497 \pm 0.001$ | $0.497 \pm 0.001$ | $0.497 \pm 0.001$ | $\mathbf{0.496 \pm 0.001}$ | $0.511 \pm 0.001$ | $0.489 \pm 0.000$ | $0.762 \pm 0.000$ |
| JS Divergence | $0.281 \pm 0.000$ | $0.281 \pm 0.000$ | $0.281 \pm 0.000$ | $\mathbf{0.280 \pm 0.000}$ | $0.331 \pm 0.000$ | $0.401 \pm 0.000$ | $0.601 \pm 0.000$ |
| Accuracy | $0.503 \pm 0.000$ | $0.504 \pm 0.000$ | $0.504 \pm 0.000$ | $0.504 \pm 0.000$ | $0.494 \pm 0.000$ | $\mathbf{0.513 \pm 0.001}$ | $0.232 \pm 0.000$ |
| Loss | $2.608 \pm 0.002$ | $2.606 \pm 0.002$ | $2.599 \pm 0.002$ | $2.591 \pm 0.003$ | $\mathbf{2.431 \pm 0.002}$ | $2.634 \pm 0.001$ | $4.703 \pm 0.002$ |

Table 18: ResNet18 on TinyImageNet Dataset mean and $\pm$ 1 SEM reported from 10 runs with Teacher Seed 2. Bold values are best performing based on the mean.

| Metrics | Control | Knowledge Distillation | | | Random Control Distillation | | |
|---|---|---|---|---|---|---|---|
| | SIDDO | 0.1 | 0.5 | 0.9 | 0.1 | 0.5 | 0.9 |
| Activation Distance | $0.546 \pm 0.000$ | $\mathbf{0.545 \pm 0.000}$ | $\mathbf{0.545 \pm 0.000}$ | $\mathbf{0.545 \pm 0.000}$ | $0.565 \pm 0.000$ | $0.651 \pm 0.000$ | $0.829 \pm 0.000$ |
| Rank Disagreement | $\mathbf{0.987 \pm 0.000}$ | $\mathbf{0.987 \pm 0.000}$ | $\mathbf{0.987 \pm 0.000}$ | $\mathbf{0.987 \pm 0.000}$ | $0.990 \pm 0.000$ | $0.990 \pm 0.000$ | $0.991 \pm 0.000$ |
| Prediction Disagreement | $0.497 \pm 0.001$ | $0.497 \pm 0.001$ | $0.497 \pm 0.001$ | $\mathbf{0.496 \pm 0.001}$ | $0.511 \pm 0.001$ | $0.489 \pm 0.000$ | $0.755 \pm 0.000$ |
| JS Divergence | $\mathbf{0.280 \pm 0.000}$ | $\mathbf{0.280 \pm 0.000}$ | $\mathbf{0.280 \pm 0.000}$ | $\mathbf{0.280 \pm 0.000}$ | $0.330 \pm 0.000$ | $0.400 \pm 0.000$ | $0.600 \pm 0.000$ |
| Accuracy | $0.503 \pm 0.001$ | $0.504 \pm 0.000$ | $0.503 \pm 0.000$ | $0.503 \pm 0.000$ | $0.493 \pm 0.000$ | $\mathbf{0.512 \pm 0.000}$ | $0.236 \pm 0.000$ |
| Loss | $2.604 \pm 0.001$ | $2.602 \pm 0.001$ | $2.594 \pm 0.001$ | $2.587 \pm 0.001$ | $\mathbf{2.434 \pm 0.001}$ | $2.641 \pm 0.001$ | $4.684 \pm 0.002$ |

Table 19: ResNet18 with ResNet50 Teacher on TinyImagenet significance testing. ✓indicates significant results compared to controls, whereas ✗indicates insignificant results compared to controls. Each tick represents a teacher (seeds 0 to 2, left to right).

| | Activation Distance | Rank Disagreement | Prediction Disagreement | JS Divergence | Accuracy | Loss |
|---|---|---|---|---|---|---|
| KD 0.1 | ✗✗✗ | ✗✗✗ | ✗✗✗ | ✗✗✗ | ✗✗✗ | ✗✗✗ |
| KD 0.5 | ✗✗✗ | ✗✗✗ | ✗✗✗ | ✗✗✗ | ✗✗✗ | ✗✗✗ |
| KD 0.9 | ✗✗✗ | ✗✗✗ | ✗✗✗ | ✗✗✗ | ✗✗✗ | ✗✗✗ |

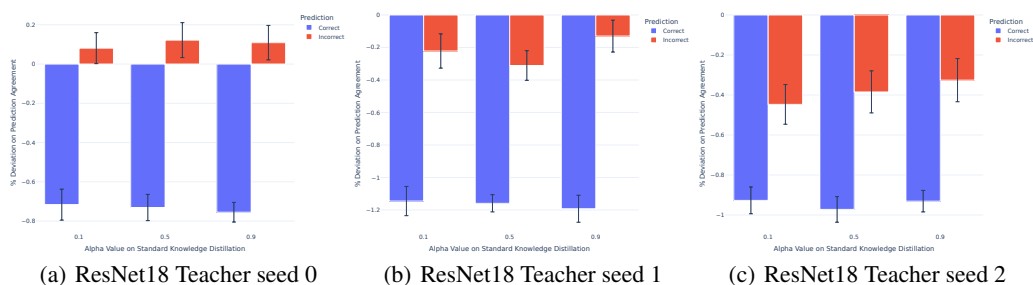

(a) ResNet18 Teacher seed 0    (b) ResNet18 Teacher seed 1    (c) ResNet18 Teacher seed 2

Figure 8: Prediction agreement difference of student models in standard KD to the highest performing control baseline with respect to correct prediction agreement (blue) and incorrect prediction agreement (red), error bars are ± 1 SEM for ResNet18 on TinyImageNet.

## E.2    IMAGENET RESNET50 TEACHER TO RESNET18 STUDENT

**Training Settings:**    a pre-trained ResNet50 model taken from PyTorch with a top-1-accuracy of 80.858 and a top-5-accuracy of 95.434[2]. As Pytorch only provides one set of pre-trained model weights there is only one teacher seed for this experiment. The ResNet18 student was trained on ImageNet (Russakovsky et al., 2015) using the FFCV setup (Leclerc et al., 2023), where 100% of the training images where compressed to a JPEG with 90% quality. The data was normalized with a mean of (0.485, 0.456, 0.406) and a standard deviation of (0.229, 0.224, 0.225). The model utilised BlurPools (Zhang, 2019) within the convolutional layers, and was trained for 56 epochs, with a batch size of 1024 using SGD, momentum of 0.9, weight decay of 5e-5, a learning rate of 0.5 using a cyclic scheduler with a learning rate step ratio of 0,1 and step length of 30. The learning rate peak was at epoch 2. The input resolution started at 160 by 160, and started to ramped up to 192 by 192 at epoch 41 and ended at 192 by 192 at epoch 48.

**Findings:**    In line with our existing results, when there is statistically significant knowledge transfer from the teacher to the student (see Table 20 and Table 21), then negative asymmetric transfer occurs with a bias towards teacher errors (see Figure 9.

Table 20: ResNet18 with ResNet50 Teacher on ImageNet mean and ± 1 SEM reported from 10 runs with Teacher Seed 0. Bold values are best performing based on the mean.

| Metrics | Control | Knowledge Distillation | | | Random Control Distillation | | |
|---|---|---|---|---|---|---|---|
| | SIDDO | 0.1 | 0.5 | 0.9 | 0.1 | 0.5 | 0.9 |
| Activation Distance | 0.42 +- 0.001 | 0.365 +- 0.001 | 0.26 +- 0.001 | **0.226 +- 0.0** | 0.268 +- 0.001 | 0.259 +- 0.002 | 0.376 +- 0.0 |
| Rank Disagreement | **0.997 +- 0.0** | **0.997 +- 0.0** | **0.997 +- 0.0** | **0.997 +- 0.0** | **0.997 +- 0.0** | **0.997 +- 0.0** | **0.997 +- 0.0** |
| Prediction Disagreement | 0.264 +- 0.003 | 0.256 +- 0.002 | 0.239 +- 0.002 | **0.235 +- 0.002** | 0.259 +- 0.001 | 0.274 +- 0.002 | 0.308 +- 0.002 |
| JS Divergence | 0.26 +- 0.001 | 0.221 +- 0.001 | 0.136 +- 0.001 | **0.106 +- 0.0** | 0.136 +- 0.001 | 0.099 +- 0.001 | 0.173 +- 0.001 |
| Accuracy | 0.68 +- 0.002 | 0.687 +- 0.002 | 0.7 +- 0.001 | **0.703 +- 0.002** | 0.684 +- 0.001 | 0.67 +- 0.001 | 0.642 +- 0.002 |
| Loss | **1.307 +- 0.009** | 1.342 +- 0.009 | 1.608 +- 0.015 | 1.833 +- 0.022 | 1.657 +- 0.013 | 2.548 +- 0.017 | 4.06 +- 0.012 |

Table 21: ResNet18 with ResNet50 Teacher on Imagenet significance testing. ✓indicates significant results compared to controls, whereas ✗indicates insignificant results compared to controls.

| | Activation Distance | Rank Disagreement | Prediction Disagreement | JS Divergence | Accuracy | Loss |
|---|---|---|---|---|---|---|
| KD 0.1 | ✗ | ✓ | ✗ | ✗ | ✗ | ✗ |
| KD 0.5 | ✗ | ✓ | ✓ | ✗ | ✓ | ✗ |
| KD 0.9 | ✓ | ✓ | ✓ | ✗ | ✓ | ✗ |

---

[2]https://docs.pytorch.org/vision/main/models/generated/torchvision.models.resnet50.html#torchvision.models.ResNet50_Weights

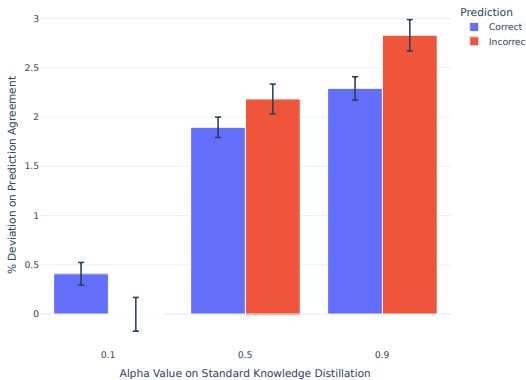

Figure 9: Prediction agreement difference of student models in standard KD to the highest performing control baseline with respect to correct prediction agreement (blue) and incorrect prediction agreement (red), error bars are $\pm$ 1 SEM for ResNet18 on ImageNet.

### E.2.1 THE EFFECT OF TEMPERATURE

Using the training setup as defined in Section E.2, we explore how temperature of 2 effects these results on ImageNet.

**Findings:** Increasing the temperature reduces the signal between the student and teacher, reducing functional similarity (see Tables 22 and 21) and negative transfer (see Figure 10), and the overall utility of KD, when compared to a temperature of 1, while not removing the negative asymmetric transfer we uncover (see Figure 10).

Table 22: ResNet18 with ResNet50 Teacher with Temperature 2 on ImageNet mean and $\pm$ 1 SEM reported from 10 runs with Teacher Seed 0. Bold values are best performing based on the mean.

| Metrics | Control | Knowledge Distillation | | | Random Control Distillation | | |
|---|---|---|---|---|---|---|---|
| | SIDDO | 0.1 | 0.5 | 0.9 | 0.1 | 0.5 | 0.9 |
| Activation Distance | 0.42 +- 0.001 | 0.31 +- 0.002 | 0.251 +- 0.001 | **0.221 +- 0.001** | 0.305 +- 0.001 | 0.247 +- 0.001 | 0.28 +- 0.002 |
| Rank Disagreement | 0.997 +- 0.0 | 0.997 +- 0.0 | 0.997 +- 0.0 | **0.996 +- 0.0** | 0.997 +- 0.0 | 0.997 +- 0.0 | 0.997 +- 0.0 |
| Prediction Disagreement | 0.264 +- 0.003 | **0.257 +- 0.002** | 0.258 +- 0.002 | 0.264 +- 0.002 | 0.259 +- 0.002 | 0.273 +- 0.002 | 0.311 +- 0.002 |
| JS Divergence | 0.26 +- 0.001 | 0.16 +- 0.001 | 0.101 +- 0.0 | **0.081 +- 0.0** | 0.152 +- 0.001 | 0.096 +- 0.0 | 0.115 +- 0.001 |
| Accuracy | 0.68 +- 0.002 | **0.685 +- 0.001** | 0.684 +- 0.002 | 0.678 +- 0.001 | 0.684 +- 0.002 | 0.671 +- 0.001 | 0.64 +- 0.002 |
| Loss | **1.307 +- 0.009** | 1.492 +- 0.014 | 1.725 +- 0.019 | 1.935 +- 0.019 | 1.533 +- 0.014 | 1.927 +- 0.019 | 3.016 +- 0.021 |

Table 23: ResNet18 with ResNet50 Teacher with Temperature 2 on Imagenet significance testing. ✓indicates significant results compared to controls, whereas ✗indicates insignificant results compared to controls.

| | Activation Distance | Rank Disagreement | Prediction Disagreement | JS Divergence | Accuracy | Loss |
|---|---|---|---|---|---|---|
| KD 0.1 | ✗ | ✓ | ✗ | ✗ | ✗ | ✗ |
| KD 0.5 | ✗ | ✓ | ✗ | ✗ | ✗ | ✗ |
| KD 0.9 | ✓ | ✓ | ✗ | ✓ | ✗ | ✗ |

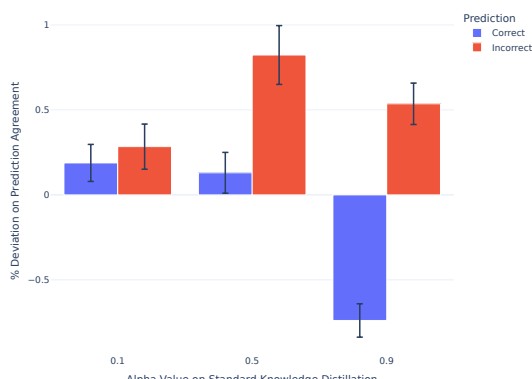

Figure 10: Prediction agreement difference of student models in standard KD with temperature 2 to the highest performing control baseline with respect to correct prediction agreement (blue) and incorrect prediction agreement (red), error bars are $\pm$ 1 SEM for ResNet18 on ImageNet.

### E.3 TINY SHAKESPEARE NANO-GPT TEACHER TO PICO-GPT STUDENT

**Training Settings:** The Nano-GPT Teacher is a GPT2-style transformer with an embedding dimension of 384, a vocabulary size of 65, six attention heads, six transformer blocks, a dropout of 0.200, and a block size of 256. The Pico-GPT student has an embedding dimension of 192, halving the internal width of the model; all other model settings are the same as the teacher.

The teacher and student are trained on the Tiny Shakespeare dataset, with the first 90% used for training and the last 10% used for testing. The dataset was tokenised via a character tokeniser, and the model was trained auto-regressively to predict the next character token. The teacher and student are trained with the Adam optimiser with a learning rate of 3e-4 with a batch size of 64 for 5000 iterations. The student models are trained with the same seeds and data orders from seeds 10 to 19 for the 10 models used for averaging. This is repeated for the three teachers trained on seeds 0 to 2.

**Justification:** This setup allows for an analysis of Knowledge Distillation where the student model is smaller than the teacher model, as expected in practice. It is not exhaustive but demonstrative that the findings we present in the main body of the paper generalise to this case. Other than the architecture's implicit bias towards the problem, which affects its performance (loss and accuracy), no confounding factors could influence Knowledge Distillation.

**Findings:** We observe a high train loss for the teacher model circa 0.86 with a high train accuracy circa 0.72; see Table 24. This high train loss corresponds as expected with a substantial knowledge transfer which increases as alpha increases, see Tables 108, 109, 110 and 111. This substantial knowledge transfer coincides with an asymmetric payoff in prediction agreement, strongly favouring incorrect predictions, see Figure 28. This result is as expected from the results and intuition presented in the results of the main body of the paper and highlights how this finding generalises.

Table 24: Teacher Performance on Train and Test Data for Nano-GPT on Tiny Shakespeare.

| Teacher Seed | Train Loss | Train Accuracy | Test Loss | Test Accuracy |
|---|---|---|---|---|
| 0 | 0.864641 | 0.719685 | 1.567481 | 0.573366 |
| 1 | 0.866370 | 0.719697 | 1.561079 | 0.574668 |
| 2 | 0.861098 | 0.721140 | 1.562137 | 0.573033 |

Table 25: Pico-GPT on Tiny Shakespeare Dataset mean and $\pm$ 1 SEM reported from 10 runs with Teacher Seed 0. Bold values are best performing based on the mean.

| Metrics | Control | Knowledge Distillation | | | Random Control Distillation | | |
|---|---|---|---|---|---|---|---|
| | SIDDO | 0.1 | 0.5 | 0.9 | 0.1 | 0.5 | 0.9 |
| Activation Distance | $0.202 \pm 0.000$ | $0.198 \pm 0.000$ | $0.181 \pm 0.000$ | $0.172 \pm 0.000$ | $0.221 \pm 0.000$ | $0.399 \pm 0.000$ | $0.663 \pm 0.000$ |
| Rank Disagreement | $0.915 \pm 0.000$ | $0.915 \pm 0.000$ | $0.912 \pm 0.000$ | $\mathbf{0.911 \pm 0.000}$ | $0.939 \pm 0.000$ | $0.944 \pm 0.000$ | $0.950 \pm 0.000$ |
| Prediction Disagreement | $0.252 \pm 0.000$ | $0.247 \pm 0.000$ | $0.226 \pm 0.000$ | $\mathbf{0.214 \pm 0.000}$ | $0.252 \pm 0.000$ | $0.253 \pm 0.001$ | $0.272 \pm 0.001$ |
| JS Divergence | $0.056 \pm 0.000$ | $0.054 \pm 0.000$ | $0.047 \pm 0.000$ | $\mathbf{0.043 \pm 0.000}$ | $0.075 \pm 0.000$ | $0.203 \pm 0.000$ | $0.451 \pm 0.000$ |
| Accuracy | $0.571 \pm 0.000$ | $0.572 \pm 0.000$ | $\mathbf{0.575 \pm 0.000}$ | $0.574 \pm 0.000$ | $0.571 \pm 0.000$ | $0.570 \pm 0.000$ | $0.561 \pm 0.000$ |
| Loss | $1.473 \pm 0.002$ | $\mathbf{1.471 \pm 0.002}$ | $1.472 \pm 0.001$ | $1.496 \pm 0.002$ | $1.483 \pm 0.001$ | $1.870 \pm 0.001$ | $3.017 \pm 0.002$ |

Table 26: Pico-GPT on Tiny Shakespeare Dataset mean and $\pm$ 1 SEM reported from 10 runs with Teacher Seed 1. Bold values are best performing based on the mean.

| Metrics | Control | Knowledge Distillation | | | Random Control Distillation | | |
|---|---|---|---|---|---|---|---|
| | SIDDO | 0.1 | 0.5 | 0.9 | 0.1 | 0.5 | 0.9 |
| Activation Distance | $0.201 \pm 0.000$ | $0.196 \pm 0.000$ | $0.180 \pm 0.000$ | $\mathbf{0.170 \pm 0.000}$ | $0.217 \pm 0.000$ | $0.392 \pm 0.000$ | $0.655 \pm 0.000$ |
| Rank Disagreement | $0.916 \pm 0.000$ | $0.915 \pm 0.000$ | $0.912 \pm 0.000$ | $\mathbf{0.911 \pm 0.000}$ | $0.939 \pm 0.000$ | $0.944 \pm 0.000$ | $0.950 \pm 0.000$ |
| Prediction Disagreement | $0.257 \pm 0.000$ | $0.251 \pm 0.000$ | $0.231 \pm 0.000$ | $\mathbf{0.219 \pm 0.000}$ | $0.256 \pm 0.000$ | $0.258 \pm 0.000$ | $0.277 \pm 0.001$ |
| JS Divergence | $0.055 \pm 0.000$ | $0.053 \pm 0.000$ | $0.046 \pm 0.000$ | $\mathbf{0.043 \pm 0.000}$ | $0.074 \pm 0.000$ | $0.201 \pm 0.000$ | $0.449 \pm 0.000$ |
| Accuracy | $0.571 \pm 0.000$ | $0.573 \pm 0.000$ | $\mathbf{0.575 \pm 0.000}$ | $0.574 \pm 0.000$ | $0.571 \pm 0.000$ | $0.570 \pm 0.000$ | $0.561 \pm 0.000$ |
| Loss | $\mathbf{1.473 \pm 0.002}$ | $1.473 \pm 0.002$ | $1.475 \pm 0.002$ | $1.492 \pm 0.002$ | $1.483 \pm 0.001$ | $1.870 \pm 0.001$ | $3.017 \pm 0.002$ |

Table 27: Pico-GPT on Tiny Shakespeare Dataset mean and $\pm$ 1 SEM reported from 10 runs with Teacher Seed 2. Bold values are best performing based on the mean.

| Metrics | Control | Knowledge Distillation | | | Random Control Distillation | | |
|---|---|---|---|---|---|---|---|
| | SIDDO | 0.1 | 0.5 | 0.9 | 0.1 | 0.5 | 0.9 |
| Activation Distance | $0.202 \pm 0.000$ | $0.197 \pm 0.000$ | $0.180 \pm 0.000$ | $\mathbf{0.171 \pm 0.000}$ | $0.219 \pm 0.000$ | $0.395 \pm 0.001$ | $0.660 \pm 0.000$ |
| Rank Disagreement | $0.915 \pm 0.000$ | $0.914 \pm 0.000$ | $0.912 \pm 0.000$ | $\mathbf{0.910 \pm 0.000}$ | $0.939 \pm 0.000$ | $0.944 \pm 0.000$ | $0.949 \pm 0.000$ |
| Prediction Disagreement | $0.252 \pm 0.000$ | $0.246 \pm 0.000$ | $0.226 \pm 0.000$ | $\mathbf{0.215 \pm 0.000}$ | $0.250 \pm 0.001$ | $0.251 \pm 0.000$ | $0.272 \pm 0.001$ |
| JS Divergence | $0.055 \pm 0.000$ | $0.053 \pm 0.000$ | $0.046 \pm 0.000$ | $\mathbf{0.043 \pm 0.000}$ | $0.074 \pm 0.000$ | $0.202 \pm 0.000$ | $0.450 \pm 0.000$ |
| Accuracy | $0.571 \pm 0.000$ | $0.572 \pm 0.000$ | $\mathbf{0.575 \pm 0.000}$ | $0.574 \pm 0.000$ | $0.572 \pm 0.000$ | $0.571 \pm 0.000$ | $0.561 \pm 0.000$ |
| Loss | $1.475 \pm 0.001$ | $\mathbf{1.470 \pm 0.001}$ | $1.471 \pm 0.002$ | $1.491 \pm 0.002$ | $1.482 \pm 0.001$ | $1.865 \pm 0.002$ | $3.017 \pm 0.001$ |

Table 28: Pico-GPT with Nano-GPT Teacher on Tiny Shakespeare significance testing. ✓indicates significant results compared to controls, whereas ✗indicates insignificant results compared to controls. Each tick represents a teacher (seeds 0 to 2, left to right).

| | Activation Distance | Rank Disagreement | Prediction Disagreement | JS Divergence | Accuracy | Loss |
|---|---|---|---|---|---|---|
| KD 0.1 | ✓✓✓ | ✓✓✓ | ✓✓✓ | ✓✓✓ | ✓✓✗ | ✗✗✓ |
| KD 0.5 | ✓✓✓ | ✓✓✓ | ✓✓✓ | ✓✓✓ | ✓✓✓ | ✗✗✗ |
| KD 0.9 | ✓✓✓ | ✓✓✓ | ✓✓✓ | ✓✓✓ | ✓✓✓ | ✗✗✗ |

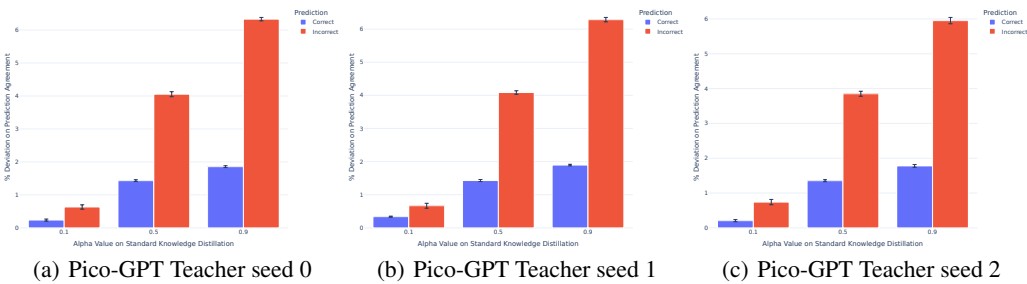

(a) Pico-GPT Teacher seed 0     (b) Pico-GPT Teacher seed 1     (c) Pico-GPT Teacher seed 2

Figure 11: Prediction agreement difference of student models in standard KD to the highest performing control baseline with respect to correct prediction agreement (blue) and incorrect prediction agreement (red), error bars are $\pm$ 1 SEM for Pico-GPT on Tiny Shakespeare.

### E.3.1 THE EFFECT OF TEMPERATURE

This section explores how temperature effects the findings of the negative asymmetric payoff of knowledge distillation. We explore temperatures 2 and 4, using the training settings as defined in Section E.3 as this represents a typical Knowledge Distillation setup, where the teacher is larger than the student.

**Findings:** In this setting a temperature of 2 and 4 resulted in a reduced accuracy increase when compared to using a temperature of 1, for all teacher seeds. The for ease and clarity the following analysis is provided for teacher seed 0, however holds for all teacher seeds. This is demonstrated with the results on teacher seed 0 where the best accuracy achieved with temperature 1 of 57.50% (see Table 25), 57.20% for temperature 2 (see Table 29) and 57.00% for temperature 4 (see Table 33). Additionally there is statistically significantly less functional knowledge passed to the student model when using a temperature of 2 and 4. Furthermore, distances between student and teacher models on functional similarity largely increase compared to temperature 1. This demonstrates that higher temperature values reduce the amount of knowledge transfer. Corresponding with the reduction in knowledge transfer as the temperature increased, we witness a reduction in the maximum correct agreement. At temperature 1 it is 1.85% at temperature 2 it is 0.85% and at temperature 4 it is 0.11%. As well a reduction in the maximum incorrect agreement. At temperature 1 it is 6.32% at temperature 2 it is 3.80% and at temperature 4 it is 2.20%. Therefore even when adjusting for temperature the the fundamental negative asymmetric transfer we identify and theoretically formalise (see Section 5) remains apparent and statistically significantly higher regardless of temperature values.

Table 29: Pico-GPT with Nano-GPT Teacher with Temperature 2 on Tiny Shakespeare mean and $\pm$ 1 SEM reported from 10 runs with Teacher Seed 0. Bold values are best performing based on the mean.

| Metrics | Control | Knowledge Distillation | | | Random Control Distillation | | |
|---|---|---|---|---|---|---|---|
| | SIDDO | 0.1 | 0.5 | 0.9 | 0.1 | 0.5 | 0.9 |
| Activation Distance | 0.202 +- 0.0 | 0.197 +- 0.0 | 0.183 +- 0.0 | **0.181 +- 0.0** | 0.213 +- 0.0 | 0.305 +- 0.001 | 0.617 +- 0.0 |
| Rank Disagreement | 0.915 +- 0.0 | 0.907 +- 0.0 | 0.896 +- 0.0 | **0.892 +- 0.0** | 0.94 +- 0.0 | 0.945 +- 0.0 | 0.95 +- 0.0 |
| Prediction Disagreement | 0.252 +- 0.0 | 0.25 +- 0.0 | 0.235 +- 0.0 | **0.23 +- 0.0** | 0.252 +- 0.0 | 0.253 +- 0.0 | 0.27 +- 0.0 |
| JS Divergence | 0.056 +- 0.0 | 0.053 +- 0.0 | **0.047 +- 0.0** | **0.047 +- 0.0** | 0.072 +- 0.0 | 0.152 +- 0.0 | 0.403 +- 0.0 |
| Accuracy | 0.571 +- 0.0 | **0.572 +- 0.0** | **0.572 +- 0.0** | 0.569 +- 0.0 | 0.571 +- 0.0 | 0.571 +- 0.0 | 0.562 +- 0.0 |
| Loss | **1.473 +- 0.002** | 1.513 +- 0.003 | 1.571 +- 0.002 | 1.622 +- 0.002 | 1.493 +- 0.001 | 1.736 +- 0.001 | 2.732 +- 0.001 |

Table 30: Pico-GPT with Nano-GPT Teacher with Temperature 2 on Tiny Shakespeare mean and $\pm$ 1 SEM reported from 10 runs with Teacher Seed 1. Bold values are best performing based on the mean.

| Metrics | Control | Knowledge Distillation | | | Random Control Distillation | | |
|---|---|---|---|---|---|---|---|
| | SIDDO | 0.1 | 0.5 | 0.9 | 0.1 | 0.5 | 0.9 |
| Activation Distance | 0.201 +- 0.0 | 0.195 +- 0.0 | 0.181 +- 0.0 | **0.179 +- 0.0** | 0.209 +- 0.0 | 0.298 +- 0.0 | 0.609 +- 0.0 |
| Rank Disagreement | 0.916 +- 0.0 | 0.907 +- 0.0 | 0.896 +- 0.0 | **0.892 +- 0.0** | 0.94 +- 0.0 | 0.945 +- 0.0 | 0.95 +- 0.0 |
| Prediction Disagreement | 0.258 +- 0.001 | 0.254 +- 0.0 | 0.24 +- 0.0 | **0.236 +- 0.0** | 0.256 +- 0.0 | 0.258 +- 0.0 | 0.279 +- 0.0 |
| JS Divergence | 0.055 +- 0.0 | 0.052 +- 0.0 | 0.047 +- 0.0 | **0.046 +- 0.0** | 0.071 +- 0.0 | 0.15 +- 0.0 | 0.401 +- 0.0 |
| Accuracy | 0.571 +- 0.0 | 0.571 +- 0.0 | **0.572 +- 0.0** | 0.569 +- 0.0 | **0.572 +- 0.0** | 0.571 +- 0.0 | 0.56 +- 0.0 |
| Loss | **1.474 +- 0.002** | 1.512 +- 0.003 | 1.569 +- 0.002 | 1.613 +- 0.003 | 1.489 +- 0.001 | 1.732 +- 0.001 | 2.739 +- 0.001 |

Table 31: Pico-GPT with Nano-GPT Teacher with Temperature 2 on Tiny Shakespeare mean and $\pm$ 1 SEM reported from 10 runs with Teacher Seed 2. Bold values are best performing based on the mean.

| Metrics | Control | Knowledge Distillation | | | Random Control Distillation | | |
|---|---|---|---|---|---|---|---|
| | SIDDO | 0.1 | 0.5 | 0.9 | 0.1 | 0.5 | 0.9 |
| Activation Distance | 0.201 +- 0.0 | 0.195 +- 0.0 | 0.181 +- 0.0 | **0.18 +- 0.0** | 0.21 +- 0.0 | 0.301 +- 0.0 | 0.615 +- 0.0 |
| Rank Disagreement | 0.915 +- 0.0 | 0.906 +- 0.0 | 0.896 +- 0.0 | **0.892 +- 0.0** | 0.94 +- 0.0 | 0.945 +- 0.0 | 0.95 +- 0.0 |
| Prediction Disagreement | 0.251 +- 0.001 | 0.247 +- 0.0 | 0.235 +- 0.0 | **0.23 +- 0.0** | 0.249 +- 0.0 | 0.252 +- 0.0 | 0.274 +- 0.0 |
| JS Divergence | 0.055 +- 0.0 | 0.052 +- 0.0 | **0.046 +- 0.0** | **0.046 +- 0.0** | 0.071 +- 0.0 | 0.15 +- 0.0 | 0.403 +- 0.0 |
| Accuracy | 0.571 +- 0.0 | 0.571 +- 0.0 | 0.571 +- 0.0 | 0.569 +- 0.0 | **0.572 +- 0.0** | 0.571 +- 0.0 | 0.56 +- 0.0 |
| Loss | **1.474 +- 0.002** | 1.513 +- 0.001 | 1.576 +- 0.001 | 1.619 +- 0.003 | 1.489 +- 0.001 | 1.732 +- 0.001 | 2.739 +- 0.001 |

Table 32: Pico-GPT with Nano-GPT Teacher with temperature 2 on Tiny Shakespeare significance testing. ✓indicates significant results compared to controls, whereas ✗indicates insignificant results compared to controls. Each tick represents a teacher (seeds 0 to 2, left to right).

| | Activation Distance | Rank Disagreement | Prediction Disagreement | JS Divergence | Accuracy | Loss |
|---|---|---|---|---|---|---|
| KD 0.1 | ✓✓✓ | ✓✓✓ | ✓✓✓ | ✓✓✓ | ✗✗✗ | ✗✗✗ |
| KD 0.5 | ✓✓✓ | ✓✓✓ | ✓✓✓ | ✓✓✓ | ✓✗✗ | ✗✗✗ |
| KD 0.9 | ✓✓✓ | ✓✓✓ | ✓✓✓ | ✓✓✓ | ✗✗✗ | ✗✗✗ |

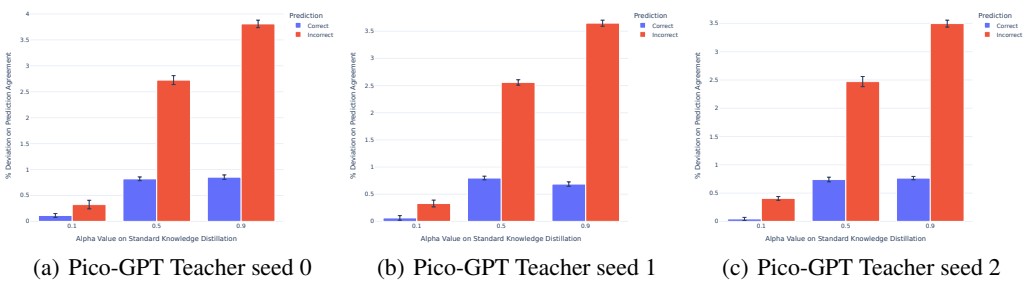

(a) Pico-GPT Teacher seed 0    (b) Pico-GPT Teacher seed 1    (c) Pico-GPT Teacher seed 2

Figure 12: Prediction agreement difference of student models in standard KD with temperature 2 to the highest performing control baseline with respect to correct prediction agreement (blue) and incorrect prediction agreement (red), error bars are ± 1 SEM for Pico-GPT on Tiny Shakespeare.

Table 33: Pico-GPT with Nano-GPT Teacher with temperature 4 on Tiny Shakespeare mean and ± 1 SEM reported from 10 runs with Teacher Seed 0. Bold values are best performing based on the mean.

| Metrics | Control | Knowledge Distillation | | | Random Control Distillation | | |
|---|---|---|---|---|---|---|---|
| | SIDDO | 0.1 | 0.5 | 0.9 | 0.1 | 0.5 | 0.9 |
| Activation Distance | 0.202 +- 0.0 | 0.199 +- 0.0 | **0.189 +- 0.0** | 0.193 +- 0.0 | 0.206 +- 0.0 | 0.262 +- 0.0 | 0.568 +- 0.001 |
| Rank Disagreement | 0.915 +- 0.0 | 0.893 +- 0.0 | 0.88 +- 0.0 | **0.876 +- 0.0** | 0.94 +- 0.0 | 0.945 +- 0.0 | 0.951 +- 0.0 |
| Prediction Disagreement | 0.252 +- 0.0 | 0.251 +- 0.001 | **0.244 +- 0.0** | 0.245 +- 0.0 | 0.253 +- 0.0 | 0.253 +- 0.0 | 0.27 +- 0.0 |
| JS Divergence | 0.056 +- 0.0 | 0.054 +- 0.0 | **0.05 +- 0.0** | 0.051 +- 0.0 | 0.067 +- 0.0 | 0.127 +- 0.0 | 0.362 +- 0.0 |
| Accuracy | 0.571 +- 0.0 | 0.57 +- 0.0 | 0.568 +- 0.0 | 0.562 +- 0.0 | **0.572 +- 0.0** | 0.571 +- 0.0 | 0.562 +- 0.0 |
| Loss | **1.473 +- 0.002** | 1.528 +- 0.002 | 1.592 +- 0.002 | 1.663 +- 0.002 | 1.491 +- 0.002 | 1.68 +- 0.0 | 2.544 +- 0.002 |

Table 34: Pico-GPT with Nano-GPT Teacher with temperature 4 on Tiny Shakespeare mean and ± 1 SEM reported from 10 runs with Teacher Seed 1. Bold values are best performing based on the mean.

| Metrics | Control | Knowledge Distillation | | | Random Control Distillation | | |
|---|---|---|---|---|---|---|---|
| | SIDDO | 0.1 | 0.5 | 0.9 | 0.1 | 0.5 | 0.9 |
| Activation Distance | 0.201 +- 0.0 | 0.196 +- 0.0 | **0.188 +- 0.0** | 0.191 +- 0.0 | 0.203 +- 0.0 | 0.256 +- 0.0 | 0.562 +- 0.0 |
| Rank Disagreement | 0.916 +- 0.0 | 0.893 +- 0.0 | 0.88 +- 0.0 | **0.876 +- 0.0** | 0.94 +- 0.0 | 0.945 +- 0.0 | 0.951 +- 0.0 |
| Prediction Disagreement | 0.258 +- 0.001 | 0.256 +- 0.0 | 0.25 +- 0.0 | **0.249 +- 0.0** | 0.256 +- 0.0 | 0.258 +- 0.0 | 0.278 +- 0.0 |
| JS Divergence | 0.055 +- 0.0 | 0.052 +- 0.0 | **0.049 +- 0.0** | 0.05 +- 0.0 | 0.066 +- 0.0 | 0.126 +- 0.0 | 0.361 +- 0.0 |
| Accuracy | **0.571 +- 0.0** | 0.57 +- 0.0 | 0.568 +- 0.0 | 0.563 +- 0.0 | **0.571 +- 0.0** | **0.571 +- 0.0** | 0.561 +- 0.0 |
| Loss | **1.474 +- 0.002** | 1.528 +- 0.002 | 1.59 +- 0.002 | 1.653 +- 0.003 | 1.489 +- 0.001 | 1.677 +- 0.001 | 2.55 +- 0.002 |

Table 35: Pico-GPT with Nano-GPT Teacher with temperature 4 on Tiny Shakespeare mean and ± 1 SEM reported from 10 runs with Teacher Seed 2. Bold values are best performing based on the mean.

| Metrics | Control | Knowledge Distillation | | | Random Control Distillation | | |
|---|---|---|---|---|---|---|---|
| | SIDDO | 0.1 | 0.5 | 0.9 | 0.1 | 0.5 | 0.9 |
| Activation Distance | 0.201 +- 0.0 | 0.197 +- 0.0 | **0.189 +- 0.0** | 0.192 +- 0.0 | 0.204 +- 0.0 | 0.259 +- 0.0 | 0.567 +- 0.0 |
| Rank Disagreement | 0.915 +- 0.0 | 0.893 +- 0.0 | 0.879 +- 0.0 | **0.876 +- 0.0** | 0.94 +- 0.0 | 0.945 +- 0.0 | 0.951 +- 0.0 |
| Prediction Disagreement | 0.251 +- 0.001 | 0.25 +- 0.001 | **0.245 +- 0.001** | 0.245 +- 0.0 | 0.25 +- 0.001 | 0.253 +- 0.0 | 0.275 +- 0.0 |
| JS Divergence | 0.055 +- 0.0 | 0.053 +- 0.0 | **0.049 +- 0.0** | 0.05 +- 0.0 | 0.066 +- 0.0 | 0.127 +- 0.0 | 0.363 +- 0.0 |
| Accuracy | **0.571 +- 0.0** | 0.57 +- 0.0 | 0.568 +- 0.0 | 0.562 +- 0.0 | **0.571 +- 0.0** | **0.571 +- 0.0** | 0.561 +- 0.0 |
| Loss | **1.474 +- 0.002** | 1.53 +- 0.001 | 1.594 +- 0.002 | 1.658 +- 0.002 | 1.489 +- 0.001 | 1.677 +- 0.001 | 2.55 +- 0.002 |

Table 36: Pico-GPT with Nano-GPT Teacher with temperature 4 on Tiny Shakespeare significance testing. ✓indicates significant results compared to controls, whereas ✗indicates insignificant results compared to controls. Each tick represents a teacher (seeds 0 to 2, left to right).

|  | Activation Distance | Rank Disagreement | Prediction Disagreement | JS Divergence | Accuracy | Loss |
|---|---|---|---|---|---|---|
| KD 0.1 | ✓✓✓ | ✓✓✓ | ✗✗✗ | ✓✓✓ | ✗✗✗ | ✗✗✗ |
| KD 0.5 | ✓✓✓ | ✓✓✓ | ✓✓✓ | ✓✓✓ | ✗✗✗ | ✗✗✗ |
| KD 0.9 | ✓✓✓ | ✓✓✓ | ✓✓✓ | ✓✓✓ | ✗✗✗ | ✗✗✗ |

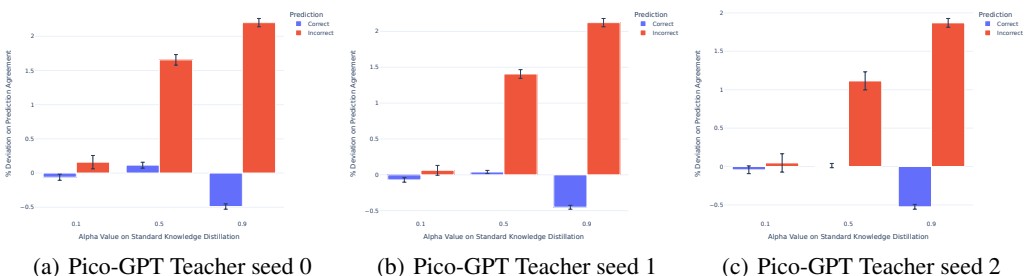

(a) Pico-GPT Teacher seed 0       (b) Pico-GPT Teacher seed 1       (c) Pico-GPT Teacher seed 2

Figure 13: Prediction agreement difference of student models in standard KD with temperature r to the highest performing control baseline with respect to correct prediction agreement (blue) and incorrect prediction agreement (red), error bars are ± 1 SEM for Pico-GPT on Tiny Shakespeare.

# F  VISION RESULTS

## F.1  TINYIMAGENET

**Training Settings:**  The ResNet50 model was trained with stochastic gradient descent with a learning rate 0.01, along with a Cosine annealing learning rate scheduler with a T_max set at 100. It was trained for 100 epochs with a batch size of 256. The data was normalized with a mean of (0.485, 0.456, 0.406) and standard deviation of (0.229, 0.224, 0.225). For ResNet50 with RandAugment (Cubuk et al., 2020), the only difference between base ResNet is the introduction of RandAugment with the default setting provided in Pytorch 2.4 (Paszke et al., 2019). The VGG19 and VGG19 with RandAugment has the same setup as the ResNet50 and ResNet50 with RandAugment respectively however it was trained **with** momentum of 0.9.

### F.1.1  RESNET50

**Findings:**  For the ResNet50 on TinyImageNet, we observe that the teacher seeds, Table 37, obtain a low train loss of 0.001 and a train accuracy of 0.99. This train performance coincides with a test accuracy of circa 0.60, resulting in a generalisation gap of circa 0.39.

For an alpha of 0.1, Table 41, we observe no significant knowledge transfer across all metrics except for Rank Disagreement with teacher seed 0. It has statistically significant transfer, but the increased similarity is extremely marginal, as observed with SIDDO and KD 0.1 having the same value to 3 significant figures, see Table 38. With this, we see a marginal prediction agreement of less than 0.5% for correct and incorrect predictions across teacher seeds, Figure 14. For alpha 0.5 and 0.9, we observe significant knowledge transfer for all bar Prediction Disagreement with alpha of 0.5 and 0.9 for teacher seed 2. However, this transfer is marginal, Tables 38, 39 and 40, and we observe a prediction agreement of less than 0.5% for correct and incorrect predictions across teacher seeds, Figure 14.

Table 37: Teacher Performance on Train and Test Data for ResNet50 on TinyImageNet.

| Teacher Seed | Train Loss | Train Accuracy | Test Loss | Test Accuracy |
|---|---|---|---|---|
| 0 | 0.001426 | 0.999800 | 2.070590 | 0.605300 |
| 1 | 0.001393 | 0.999800 | 2.051494 | 0.607900 |
| 2 | 0.001436 | 0.999800 | 2.051024 | 0.610600 |

Table 38: ResNet50 on TinyImageNet mean and $\pm$ 1 SEM reported from 10 runs with Teacher Seed 0. **Bold** values are the best performing based on the mean.

| Metrics | Control | Knowledge Distillation | | | Random Control Distillation | | |
|---|---|---|---|---|---|---|---|
| | SIDDO | 0.1 | 0.5 | 0.9 | 0.1 | 0.5 | 0.9 |
| Activation Distance | $0.157 \pm 0.001$ | $0.157 \pm 0.001$ | $0.156 \pm 0.001$ | $\mathbf{0.155 \pm 0.000}$ | $0.343 \pm 0.000$ | $0.581 \pm 0.000$ | $0.791 \pm 0.000$ |
| Rank Disagreement | $\mathbf{0.939 \pm 0.000}$ | $\mathbf{0.939 \pm 0.000}$ | $\mathbf{0.939 \pm 0.000}$ | $\mathbf{0.939 \pm 0.000}$ | $0.980 \pm 0.000$ | $0.984 \pm 0.000$ | $0.984 \pm 0.000$ |
| Prediction Disagreement | $0.153 \pm 0.001$ | $0.152 \pm 0.001$ | $\mathbf{0.151 \pm 0.001}$ | $\mathbf{0.151 \pm 0.001}$ | $0.190 \pm 0.001$ | $0.214 \pm 0.000$ | $0.324 \pm 0.000$ |
| JS Divergence | $0.040 \pm 0.000$ | $0.040 \pm 0.000$ | $\mathbf{0.039 \pm 0.000}$ | $\mathbf{0.039 \pm 0.000}$ | $0.171 \pm 0.000$ | $0.333 \pm 0.000$ | $0.533 \pm 0.000$ |
| Accuracy | $0.605 \pm 0.001$ | $0.605 \pm 0.000$ | $0.604 \pm 0.001$ | $0.605 \pm 0.001$ | $\mathbf{0.607 \pm 0.000}$ | $0.606 \pm 0.001$ | $0.580 \pm 0.000$ |
| Loss | $2.068 \pm 0.001$ | $2.065 \pm 0.002$ | $2.055 \pm 0.001$ | $2.043 \pm 0.002$ | $\mathbf{1.977 \pm 0.001}$ | $2.497 \pm 0.001$ | $3.612 \pm 0.002$ |

Table 39: ResNet50 on TinyImageNet mean and $\pm$ 1 SEM reported from 10 runs with Teacher Seed 1. **Bold** values are the best performing based on the mean.

| Metrics | Control | Knowledge Distillation | | | Random Control Distillation | | |
|---|---|---|---|---|---|---|---|
| | SIDDO | 0.1 | 0.5 | 0.9 | 0.1 | 0.5 | 0.9 |
| Activation Distance | $0.156 \pm 0.001$ | $0.156 \pm 0.000$ | $0.155 \pm 0.001$ | $\mathbf{0.153 \pm 0.000}$ | $0.340 \pm 0.000$ | $0.579 \pm 0.000$ | $0.792 \pm 0.000$ |
| Rank Disagreement | $0.940 \pm 0.000$ | $0.940 \pm 0.000$ | $\mathbf{0.939 \pm 0.000}$ | $\mathbf{0.939 \pm 0.000}$ | $0.980 \pm 0.000$ | $0.984 \pm 0.000$ | $0.984 \pm 0.000$ |
| Prediction Disagreement | $0.148 \pm 0.001$ | $0.149 \pm 0.001$ | $0.148 \pm 0.001$ | $\mathbf{0.146 \pm 0.001}$ | $0.185 \pm 0.001$ | $0.209 \pm 0.000$ | $0.330 \pm 0.000$ |
| JS Divergence | $0.040 \pm 0.000$ | $0.040 \pm 0.000$ | $0.039 \pm 0.000$ | $\mathbf{0.038 \pm 0.000}$ | $0.170 \pm 0.000$ | $0.332 \pm 0.000$ | $0.534 \pm 0.000$ |
| Accuracy | $0.607 \pm 0.001$ | $\mathbf{0.608 \pm 0.001}$ | $0.607 \pm 0.000$ | $0.607 \pm 0.001$ | $0.605 \pm 0.000$ | $0.602 \pm 0.001$ | $0.576 \pm 0.000$ |
| Loss | $2.048 \pm 0.002$ | $2.048 \pm 0.002$ | $2.034 \pm 0.002$ | $2.025 \pm 0.002$ | $\mathbf{1.973 \pm 0.001}$ | $2.498 \pm 0.001$ | $3.611 \pm 0.002$ |

Table 40: ResNet50 on TinyImageNet mean and $\pm$ 1 SEM reported from 10 runs with Teacher Seed 2. **Bold** values are the best performing based on the mean.

| Metrics | Control | Knowledge Distillation | | | Random Control Distillation | | |
|---|---|---|---|---|---|---|---|
| | SIDDO | 0.1 | 0.5 | 0.9 | 0.1 | 0.5 | 0.9 |
| Activation Distance | $0.157 \pm 0.000$ | $0.157 \pm 0.000$ | $\mathbf{0.155 \pm 0.000}$ | $\mathbf{0.155 \pm 0.000}$ | $0.342 \pm 0.000$ | $0.581 \pm 0.000$ | $0.792 \pm 0.000$ |
| Rank Disagreement | $\mathbf{0.939 \pm 0.000}$ | $\mathbf{0.939 \pm 0.000}$ | $\mathbf{0.939 \pm 0.000}$ | $\mathbf{0.939 \pm 0.000}$ | $0.980 \pm 0.000$ | $0.984 \pm 0.000$ | $0.984 \pm 0.000$ |
| Prediction Disagreement | $0.152 \pm 0.001$ | $0.152 \pm 0.001$ | $\mathbf{0.151 \pm 0.001}$ | $\mathbf{0.151 \pm 0.001}$ | $0.187 \pm 0.001$ | $0.213 \pm 0.001$ | $0.327 \pm 0.000$ |
| JS Divergence | $0.040 \pm 0.000$ | $0.040 \pm 0.000$ | $\mathbf{0.039 \pm 0.000}$ | $\mathbf{0.039 \pm 0.000}$ | $0.171 \pm 0.000$ | $0.334 \pm 0.000$ | $0.534 \pm 0.000$ |
| Accuracy | $0.608 \pm 0.001$ | $0.607 \pm 0.001$ | $0.607 \pm 0.000$ | $\mathbf{0.609 \pm 0.001}$ | $0.608 \pm 0.001$ | $0.605 \pm 0.001$ | $0.577 \pm 0.000$ |
| Loss | $2.054 \pm 0.002$ | $2.050 \pm 0.002$ | $2.040 \pm 0.003$ | $2.025 \pm 0.002$ | $\mathbf{1.967 \pm 0.001}$ | $2.494 \pm 0.001$ | $3.602 \pm 0.002$ |

Table 41: ResNet50 on TinyImageNet significance testing. ✓indicates significant results compared to controls, whereas ✗indicates insignificant results compared to controls. Each tick represents a teacher (seeds 0 to 2, left to right).

| | Activation Distance | Rank Disagreement | Prediction Disagreement | JS Divergence | Accuracy | Loss |
|---|---|---|---|---|---|---|
| KD 0.1 | ✗✗✗ | ✓✗✗ | ✗✗✗ | ✗✗✗ | ✗✗✗ | ✗✗✗ |
| KD 0.5 | ✓✓✓ | ✓✓✓ | ✗✗✗ | ✓✓✓ | ✗✗✗ | ✗✗✗ |
| KD 0.9 | ✓✓✓ | ✓✓✓ | ✓✓✗ | ✓✓✓ | ✗✗✗ | ✗✗✗ |

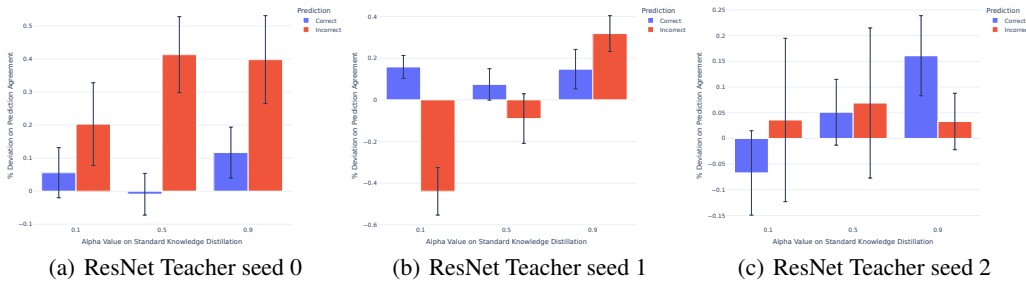

(a) ResNet Teacher seed 0   (b) ResNet Teacher seed 1   (c) ResNet Teacher seed 2

Figure 14: Prediction agreement difference of student models in standard KD to the highest performing control baseline with respect to correct prediction agreement (blue) and incorrect prediction agreement (red), error bars are $\pm$ 1 SEM for ResNet50 on TinyImageNet.

### F.1.2   RESNET50 WITH RANDAUGMENT

**Findings:**   For the ResNet50 on TinyImageNet with RandAugment, we observe that the teacher seeds, Table 37, obtain a high train loss and a train accuracy of circa 0.84. This train performance coincides with a test accuracy of circa 0.64, resulting in a generalisation gap of circa 0.2.

We observe significant knowledge transfer for all alpha values with a strong asymmetric transfer of knowledge favouring incorrect predictions as shown in Table 46 and Figure 15, respectively. However, it is important to note that despite significant and substantial knowledge transfer, we do not see any improvement in test accuracy over the control and random controls.

Table 42: Teacher Performance on Train and Test Data.

| Teacher Seed | Train Loss | Train Accuracy | Test Loss | Test Accuracy |
|---|---|---|---|---|
| 0 | 0.672748 | 0.840410 | 1.620552 | 0.638800 |
| 1 | 0.678245 | 0.839200 | 1.629393 | 0.641800 |
| 2 | 0.667570 | 0.840750 | 1.624969 | 0.641100 |

Table 43: ResNet50 on TinyImageNet with RandAugment mean and $\pm$ 1 SEM reported from 10 runs with Teacher Seed 0. Bold values are the best performing based on the mean.

| Metrics | Control | Knowledge Distillation | | | Random Control Distillation | | |
|---|---|---|---|---|---|---|---|
| | SIDDO | 0.1 | 0.5 | 0.9 | 0.1 | 0.5 | 0.9 |
| Activation Distance | 0.193 ± 0.000 | 0.183 ± 0.000 | 0.150 ± 0.000 | **0.131 ± 0.000** | 0.245 ± 0.001 | 0.501 ± 0.001 | 0.781 ± 0.000 |
| Rank Disagreement | 0.959 ± 0.000 | 0.957 ± 0.000 | 0.948 ± 0.000 | **0.943 ± 0.000** | 0.975 ± 0.000 | 0.981 ± 0.000 | 0.987 ± 0.000 |
| Prediction Disagreement | 0.196 ± 0.001 | 0.188 ± 0.001 | 0.154 ± 0.001 | **0.136 ± 0.001** | 0.195 ± 0.001 | 0.240 ± 0.001 | 0.572 ± 0.001 |
| JS Divergence | 0.058 ± 0.000 | 0.052 ± 0.000 | 0.036 ± 0.000 | **0.028 ± 0.000** | 0.094 ± 0.000 | 0.266 ± 0.000 | 0.563 ± 0.000 |
| Accuracy | 0.640 ± 0.000 | 0.643 ± 0.001 | 0.644 ± 0.000 | 0.642 ± 0.000 | 0.646 ± 0.001 | **0.657 ± 0.001** | 0.400 ± 0.001 |
| Loss | 1.619 ± 0.003 | 1.600 ± 0.001 | 1.578 ± 0.001 | 1.577 ± 0.001 | **1.551 ± 0.001** | 1.984 ± 0.002 | 4.211 ± 0.001 |

Table 44: ResNet50 on TinyImageNet with RandAugment mean and $\pm$ 1 SEM reported from 10 runs with Teacher Seed 1. Bold values are the best performing based on the mean.

| Metrics | Control | Knowledge Distillation | | | Random Control Distillation | | |
|---|---|---|---|---|---|---|---|
| | SIDDO | 0.1 | 0.5 | 0.9 | 0.1 | 0.5 | 0.9 |
| Activation Distance | 0.194 ± 0.000 | 0.183 ± 0.001 | 0.148 ± 0.000 | **0.13 ± 0.000** | 0.247 ± 0.000 | 0.503 ± 0.000 | 0.783 ± 0.000 |
| Rank Disagreement | 0.959 ± 0.000 | 0.957 ± 0.000 | 0.948 ± 0.000 | **0.943 ± 0.000** | 0.975 ± 0.000 | 0.981 ± 0.000 | 0.987 ± 0.000 |
| Prediction Disagreement | 0.195 ± 0.001 | 0.186 ± 0.001 | 0.151 ± 0.001 | **0.134 ± 0.001** | 0.194 ± 0.001 | 0.241 ± 0.000 | 0.577 ± 0.001 |
| JS Divergence | 0.058 ± 0.000 | 0.053 ± 0.000 | 0.036 ± 0.000 | **0.028 ± 0.000** | 0.095 ± 0.000 | 0.267 ± 0.000 | 0.565 ± 0.000 |
| Accuracy | 0.639 ± 0.001 | 0.640 ± 0.001 | 0.641 ± 0.001 | 0.640 ± 0.001 | 0.646 ± 0.001 | **0.658 ± 0.000** | 0.396 ± 0.001 |
| Loss | 1.620 ± 0.002 | 1.608 ± 0.002 | 1.584 ± 0.001 | 1.584 ± 0.001 | **1.555 ± 0.002** | 1.986 ± 0.002 | 4.214 ± 0.002 |

Table 45: ResNet50 on TinyImageNet with RandAugment mean and $\pm$ 1 SEM reported from 10 runs with Teacher Seed 2. Bold values are the best performing based on the mean.

| Metrics | Control | Knowledge Distillation | | | Random Control Distillation | | |
|---|---|---|---|---|---|---|---|
| | SIDDO | 0.1 | 0.5 | 0.9 | 0.1 | 0.5 | 0.9 |
| Activation Distance | $0.195 \pm 0.000$ | $0.185 \pm 0.000$ | $0.150 \pm 0.000$ | $\mathbf{0.131 \pm 0.000}$ | $0.247 \pm 0.001$ | $0.504 \pm 0.000$ | $0.783 \pm 0.000$ |
| Rank Disagreement | $0.959 \pm 0.000$ | $0.957 \pm 0.000$ | $0.948 \pm 0.000$ | $\mathbf{0.943 \pm 0.000}$ | $0.975 \pm 0.000$ | $0.981 \pm 0.000$ | $0.987 \pm 0.000$ |
| Prediction Disagreement | $0.197 \pm 0.001$ | $0.189 \pm 0.001$ | $0.155 \pm 0.001$ | $\mathbf{0.135 \pm 0.001}$ | $0.197 \pm 0.001$ | $0.239 \pm 0.000$ | $0.564 \pm 0.001$ |
| JS Divergence | $0.059 \pm 0.000$ | $0.053 \pm 0.000$ | $0.037 \pm 0.000$ | $\mathbf{0.028 \pm 0.000}$ | $0.096 \pm 0.000$ | $0.267 \pm 0.000$ | $0.563 \pm 0.000$ |
| Accuracy | $0.640 \pm 0.001$ | $0.641 \pm 0.001$ | $0.643 \pm 0.001$ | $0.643 \pm 0.000$ | $0.647 \pm 0.001$ | $\mathbf{0.657 \pm 0.000}$ | $0.410 \pm 0.001$ |
| Loss | $1.621 \pm 0.002$ | $1.606 \pm 0.001$ | $1.581 \pm 0.001$ | $1.582 \pm 0.001$ | $\mathbf{1.552 \pm 0.001}$ | $1.982 \pm 0.002$ | $4.180 \pm 0.002$ |

Table 46: ResNet50 on TinyImageNet with RandAugment significance testing. ✓indicates significant results compared to controls, whereas ✗indicates insignificant results compared to controls. Each tick represents a teacher (seeds 0 to 2, left to right).

| | Activation Distance | Rank Disagreement | Prediction Disagreement | JS Divergence | Accuracy | Loss |
|---|---|---|---|---|---|---|
| KD 0.1 | ✓✓✓ | ✓✓✓ | ✓✓✓ | ✓✓✓ | ✗✗✗ | ✗✗✗ |
| KD 0.5 | ✓✓✓ | ✓✓✓ | ✓✓✓ | ✓✓✓ | ✗✗✗ | ✗✗✗ |
| KD 0.9 | ✓✓✓ | ✓✓✓ | ✓✓✓ | ✓✓✓ | ✗✗✗ | ✗✗✗ |

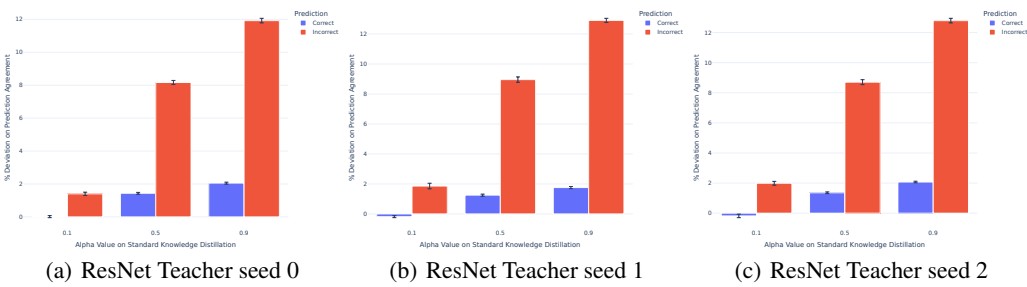

(a) ResNet Teacher seed 0  (b) ResNet Teacher seed 1  (c) ResNet Teacher seed 2

Figure 15: Prediction agreement difference of student models in standard KD to the highest performing control baseline with respect to correct prediction agreement (blue) and incorrect prediction agreement (red), error bars are $\pm$ 1 SEM for ResNet50 on TinyImageNet with RandAugment.

### F.1.3 VGG19

**Findings:** For the VGG19 on the TinyImageNet, we observe a low train loss of circa 0.000286 and a train accuracy of 0.9998. As expected, given our results and discussion in the main body of the paper on the ResNet50, we see no significant transfer until an alpha of 0.9. With teacher seed 0 and 2 with an alpha of 0.9, we record significant transfer for Activation Distance and for teacher seed 0 on JS Divergence, as seen in Table 51. When we observe knowledge transfer with an alpha of 0.9, we observe a slight preference for positive agreement of test prediction; however, the results have a large SEM, and the amount of agreement is less than 0.5%, making the results less reliable and insignificant in either transfer direction.

Table 47: Teacher Performance on Train and Test Data.

| Teacher Seed | Train Loss | Train Accuracy | Test Loss | Test Accuracy |
|---|---|---|---|---|
| 0 | 0.000286 | 0.999800 | 3.351542 | 0.633200 |
| 1 | 0.000286 | 0.999800 | 3.301587 | 0.637200 |
| 2 | 0.000285 | 0.999800 | 3.311130 | 0.633500 |

Table 48: VGG19 on TinyImageNet mean and $\pm$ 1 SEM reported from 10 runs with Teacher Seed 0. **Bold** values are the best performing based on the mean.

| Metrics | Control | Knowledge Distillation | | | Random Control Distillation | | |
|---|---|---|---|---|---|---|---|
| | SIDDO | 0.1 | 0.5 | 0.9 | 0.1 | 0.5 | 0.9 |
| Activation Distance | $0.418 \pm 0.001$ | $0.419 \pm 0.001$ | $0.418 \pm 0.001$ | $\mathbf{0.416 \pm 0.001}$ | $0.522 \pm 0.001$ | $0.741 \pm 0.000$ | $0.886 \pm 0.000$ |
| Rank Disagreement | $\mathbf{0.978 \pm 0.000}$ | $\mathbf{0.978 \pm 0.000}$ | $\mathbf{0.978 \pm 0.000}$ | $\mathbf{0.978 \pm 0.000}$ | $0.987 \pm 0.000$ | $0.988 \pm 0.000$ | $0.989 \pm 0.000$ |
| Prediction Disagreement | $\mathbf{0.332 \pm 0.001}$ | $\mathbf{0.332 \pm 0.001}$ | $\mathbf{0.332 \pm 0.001}$ | $0.330 \pm 0.001$ | $0.348 \pm 0.001$ | $0.381 \pm 0.001$ | $0.412 \pm 0.000$ |
| JS Divergence | $0.195 \pm 0.000$ | $0.195 \pm 0.000$ | $0.195 \pm 0.000$ | $\mathbf{0.194 \pm 0.000}$ | $0.308 \pm 0.001$ | $0.457 \pm 0.000$ | $0.593 \pm 0.000$ |
| Accuracy | $0.635 \pm 0.001$ | $0.635 \pm 0.001$ | $0.636 \pm 0.001$ | $\mathbf{0.638 \pm 0.001}$ | $0.627 \pm 0.001$ | $0.603 \pm 0.001$ | $0.576 \pm 0.001$ |
| Loss | $3.332 \pm 0.010$ | $3.329 \pm 0.012$ | $3.308 \pm 0.011$ | $3.313 \pm 0.010$ | $\mathbf{2.003 \pm 0.005}$ | $2.732 \pm 0.002$ | $3.682 \pm 0.002$ |

Table 49: VGG19 on TinyImageNet mean and $\pm$ 1 SEM reported from 10 runs with Teacher Seed 1. Bold values are the best performing based on the mean.

| Metrics | Control | Knowledge Distillation | | | Random Control Distillation | | |
|---|---|---|---|---|---|---|---|
| | SIDDO | 0.1 | 0.5 | 0.9 | 0.1 | 0.5 | 0.9 |
| Activation Distance | $0.414 \pm 0.002$ | $0.414 \pm 0.001$ | $\mathbf{0.413 \pm 0.001}$ | $\mathbf{0.413 \pm 0.001}$ | $0.522 \pm 0.001$ | $0.742 \pm 0.000$ | $0.886 \pm 0.000$ |
| Rank Disagreement | $\mathbf{0.978 \pm 0.000}$ | $\mathbf{0.978 \pm 0.000}$ | $\mathbf{0.978 \pm 0.000}$ | $\mathbf{0.978 \pm 0.000}$ | $0.987 \pm 0.000$ | $0.988 \pm 0.000$ | $0.989 \pm 0.000$ |
| Prediction Disagreement | $0.329 \pm 0.001$ | $0.329 \pm 0.001$ | $\mathbf{0.328 \pm 0.001}$ | $\mathbf{0.328 \pm 0.001}$ | $0.348 \pm 0.001$ | $0.379 \pm 0.001$ | $0.410 \pm 0.000$ |
| JS Divergence | $0.194 \pm 0.001$ | $0.194 \pm 0.001$ | $\mathbf{0.193 \pm 0.001}$ | $\mathbf{0.193 \pm 0.001}$ | $0.308 \pm 0.000$ | $0.457 \pm 0.000$ | $0.593 \pm 0.000$ |
| Accuracy | $0.635 \pm 0.001$ | $0.636 \pm 0.001$ | $\mathbf{0.638 \pm 0.001}$ | $0.637 \pm 0.001$ | $0.627 \pm 0.001$ | $0.603 \pm 0.001$ | $0.574 \pm 0.001$ |
| Loss | $3.345 \pm 0.011$ | $3.318 \pm 0.009$ | $3.306 \pm 0.009$ | $3.311 \pm 0.010$ | $\mathbf{2.004 \pm 0.004}$ | $2.733 \pm 0.004$ | $3.682 \pm 0.002$ |

Table 50: VGG19 on TinyImageNet mean and $\pm$ 1 SEM reported from 10 runs with Teacher Seed 2. Bold values are the best performing based on the mean.

| Metrics | Control | Knowledge Distillation | | | Random Control Distillation | | |
|---|---|---|---|---|---|---|---|
| | SIDDO | 0.1 | 0.5 | 0.9 | 0.1 | 0.5 | 0.9 |
| Activation Distance | $0.419 \pm 0.001$ | $0.417 \pm 0.001$ | $0.418 \pm 0.001$ | $0.417 \pm 0.001$ | $0.524 \pm 0.000$ | $0.743 \pm 0.000$ | $0.886 \pm 0.000$ |
| Rank Disagreement | $\mathbf{0.978 \pm 0.000}$ | $\mathbf{0.978 \pm 0.000}$ | $\mathbf{0.978 \pm 0.000}$ | $\mathbf{0.978 \pm 0.000}$ | $0.987 \pm 0.000$ | $0.988 \pm 0.000$ | $0.989 \pm 0.000$ |
| Prediction Disagreement | $0.332 \pm 0.001$ | $0.332 \pm 0.001$ | $0.332 \pm 0.001$ | $\mathbf{0.331 \pm 0.001}$ | $0.354 \pm 0.001$ | $0.385 \pm 0.001$ | $0.414 \pm 0.001$ |
| JS Divergence | $0.196 \pm 0.000$ | $\mathbf{0.195 \pm 0.001}$ | $0.196 \pm 0.000$ | $\mathbf{0.195 \pm 0.000}$ | $0.309 \pm 0.000$ | $0.458 \pm 0.000$ | $0.593 \pm 0.000$ |
| Accuracy | $0.635 \pm 0.001$ | $0.636 \pm 0.000$ | $0.635 \pm 0.001$ | $\mathbf{0.637 \pm 0.001}$ | $0.626 \pm 0.001$ | $0.602 \pm 0.001$ | $0.577 \pm 0.001$ |
| Loss | $3.314 \pm 0.009$ | $3.298 \pm 0.004$ | $3.318 \pm 0.011$ | $3.263 \pm 0.009$ | $\mathbf{1.998 \pm 0.004}$ | $2.738 \pm 0.003$ | $3.681 \pm 0.002$ |

Table 51: VGG19 on TinyImageNet significance testing. ✓indicates significant results compared to controls, whereas ✗indicates insignificant results compared to controls. Each tick represents a teacher (seeds 0 to 2, left to right).

| | Activation Distance | Rank Disagreement | Prediction Disagreement | JS Divergence | Accuracy | Loss |
|---|---|---|---|---|---|---|
| KD 0.1 | ✗✗✗ | ✗✗✗ | ✗✗✗ | ✗✗✗ | ✗✗✗ | ✗✗✗ |
| KD 0.5 | ✗✗✗ | ✗✗✗ | ✗✗✗ | ✗✗✗ | ✗✗✗ | ✗✗✗ |
| KD 0.9 | ✓✗✓ | ✗✗✗ | ✗✗✗ | ✓✗✗ | ✗✗✗ | ✗✗✗ |

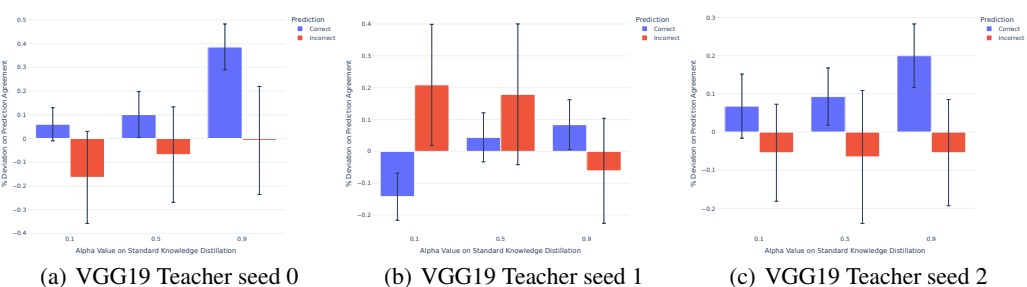

(a) VGG19 Teacher seed 0     (b) VGG19 Teacher seed 1     (c) VGG19 Teacher seed 2

Figure 16: Prediction agreement difference of student models in standard KD to the highest performing control baseline with respect to correct prediction agreement (blue) and incorrect prediction agreement (red), error bars are $\pm$ 1 SEM for VGG19 on TinyImageNet.

### F.1.4 VGG19 WITH RANDAUGMENT

**Findings:** For the VGG19 on the TinyImageNet with RandAugment, we observe a high train loss of circa 0.27 and a train accuracy of circa 0.93. As expected, given the results on the RandAugment ResNet50 that we present in the main body of the paper, we see substantial transfer across all alpha values; see Tables 53, 54, 55 and 56. This substantial and significant transfer of knowledge, as expected, coincides with a strong asymmetric transfer of knowledge favouring incorrect predictions, as shown in Figure 17.

Table 52: Teacher Performance on Train and Test Data.

| Teacher Seed | Train Loss | Train Accuracy | Test Loss | Test Accuracy |
|---|---|---|---|---|
| 0 | 0.272582 | 0.933990 | 2.565560 | 0.622600 |
| 1 | 0.269916 | 0.935140 | 2.570119 | 0.618900 |
| 2 | 0.273968 | 0.934700 | 2.609870 | 0.620100 |

Table 53: VGG19 on TinyImageNet with RandAugment mean and $\pm$ 1 SEM reported from 10 runs with Teacher Seed 0. Bold values are the best performing based on the mean.

| Metrics | Control | Knowledge Distillation | | | Random Control Distillation | | |
|---|---|---|---|---|---|---|---|
| | SIDDO | 0.1 | 0.5 | 0.9 | 0.1 | 0.5 | 0.9 |
| Activation Distance | $0.393 \pm 0.001$ | $0.388 \pm 0.001$ | $0.368 \pm 0.001$ | $\mathbf{0.355 \pm 0.001}$ | $0.431 \pm 0.001$ | $0.648 \pm 0.000$ | $0.848 \pm 0.001$ |
| Rank Disagreement | $0.976 \pm 0.000$ | $0.976 \pm 0.000$ | $0.975 \pm 0.001$ | $\mathbf{0.974 \pm 0.000}$ | $0.985 \pm 0.000$ | $0.987 \pm 0.000$ | $0.987 \pm 0.000$ |
| Prediction Disagreement | $0.335 \pm 0.001$ | $0.333 \pm 0.001$ | $0.320 \pm 0.001$ | $\mathbf{0.312 \pm 0.001}$ | $0.341 \pm 0.001$ | $0.352 \pm 0.001$ | $0.396 \pm 0.004$ |
| JS Divergence | $0.182 \pm 0.000$ | $0.178 \pm 0.000$ | $0.166 \pm 0.000$ | $\mathbf{0.159 \pm 0.000}$ | $0.228 \pm 0.000$ | $0.377 \pm 0.000$ | $0.577 \pm 0.001$ |
| Accuracy | $0.621 \pm 0.001$ | $0.624 \pm 0.001$ | $0.631 \pm 0.001$ | $\mathbf{0.633 \pm 0.001}$ | $0.622 \pm 0.001$ | $0.628 \pm 0.001$ | $0.609 \pm 0.004$ |
| Loss | $2.586 \pm 0.009$ | $2.442 \pm 0.005$ | $2.148 \pm 0.004$ | $2.022 \pm 0.003$ | $\mathbf{1.792 \pm 0.003}$ | $2.258 \pm 0.002$ | $3.533 \pm 0.013$ |

Table 54: VGG19 on TinyImageNet with RandAugment mean and $\pm$ 1 SEM reported from 10 runs with Teacher Seed 1. Bold values are the best performing based on the mean.

| Metrics | Control | Knowledge Distillation | | | Random Control Distillation | | |
|---|---|---|---|---|---|---|---|
| | SIDDO | 0.1 | 0.5 | 0.9 | 0.1 | 0.5 | 0.9 |
| Activation Distance | $0.391 \pm 0.001$ | $0.384 \pm 0.001$ | $0.362 \pm 0.001$ | $\mathbf{0.351 \pm 0.000}$ | $0.428 \pm 0.001$ | $0.644 \pm 0.000$ | $0.845 \pm 0.000$ |
| Rank Disagreement | $0.977 \pm 0.000$ | $0.976 \pm 0.000$ | $0.975 \pm 0.000$ | $\mathbf{0.974 \pm 0.000}$ | $0.985 \pm 0.000$ | $0.987 \pm 0.000$ | $0.987 \pm 0.000$ |
| Prediction Disagreement | $0.333 \pm 0.001$ | $0.330 \pm 0.001$ | $0.316 \pm 0.001$ | $\mathbf{0.308 \pm 0.001}$ | $0.337 \pm 0.001$ | $0.348 \pm 0.001$ | $0.392 \pm 0.001$ |
| JS Divergence | $0.180 \pm 0.000$ | $0.176 \pm 0.000$ | $0.164 \pm 0.000$ | $\mathbf{0.156 \pm 0.000}$ | $0.226 \pm 0.000$ | $0.375 \pm 0.000$ | $0.576 \pm 0.000$ |
| Accuracy | $0.622 \pm 0.001$ | $0.624 \pm 0.000$ | $0.632 \pm 0.001$ | $\mathbf{0.635 \pm 0.001}$ | $0.625 \pm 0.001$ | $0.627 \pm 0.001$ | $0.611 \pm 0.001$ |
| Loss | $2.575 \pm 0.004$ | $2.439 \pm 0.007$ | $2.149 \pm 0.006$ | $2.017 \pm 0.002$ | $\mathbf{1.781 \pm 0.005}$ | $2.254 \pm 0.003$ | $3.526 \pm 0.003$ |

Table 55: VGG19 on TinyImageNet with RandAugment mean and $\pm$ 1 SEM reported from 10 runs with Teacher Seed 2. Bold values are the best performing based on the mean.

| Metrics | Control | Knowledge Distillation | | | Random Control Distillation | | |
|---|---|---|---|---|---|---|---|
| | SIDDO | 0.1 | 0.5 | 0.9 | 0.1 | 0.5 | 0.9 |
| Activation Distance | $0.395 \pm 0.001$ | $0.389 \pm 0.001$ | $0.368 \pm 0.001$ | $\mathbf{0.358 \pm 0.001}$ | $0.435 \pm 0.001$ | $0.649 \pm 0.000$ | $0.850 \pm 0.001$ |
| Rank Disagreement | $0.977 \pm 0.000$ | $0.977 \pm 0.000$ | $0.975 \pm 0.000$ | $\mathbf{0.975 \pm 0.000}$ | $0.985 \pm 0.000$ | $0.987 \pm 0.000$ | $0.987 \pm 0.000$ |
| Prediction Disagreement | $0.335 \pm 0.001$ | $0.334 \pm 0.001$ | $0.321 \pm 0.001$ | $\mathbf{0.313 \pm 0.001}$ | $0.341 \pm 0.001$ | $0.352 \pm 0.001$ | $0.403 \pm 0.010$ |
| JS Divergence | $0.182 \pm 0.000$ | $0.179 \pm 0.000$ | $0.167 \pm 0.000$ | $\mathbf{0.160 \pm 0.001}$ | $0.230 \pm 0.000$ | $0.378 \pm 0.000$ | $0.579 \pm 0.002$ |
| Accuracy | $0.621 \pm 0.001$ | $0.623 \pm 0.001$ | $0.631 \pm 0.001$ | $\mathbf{0.636 \pm 0.001}$ | $0.623 \pm 0.001$ | $0.628 \pm 0.001$ | $0.600 \pm 0.011$ |
| Loss | $2.583 \pm 0.006$ | $2.441 \pm 0.009$ | $2.145 \pm 0.006$ | $2.012 \pm 0.007$ | $\mathbf{1.780 \pm 0.003}$ | $2.257 \pm 0.003$ | $3.556 \pm 0.034$ |

Table 56: VGG19 on TinyImageNet with RandAugment significance testing. ✓indicates significant results compared to controls, whereas ✗indicates insignificant results compared to controls. Each tick represents a teacher (seeds 0 to 2, left to right).

| | Activation Distance | Rank Disagreement | Prediction Disagreement | JS Divergence | Accuracy | Loss |
|---|---|---|---|---|---|---|
| KD 0.1 | ✓✓✓ | ✓✓✓ | ✗✓✗ | ✓✓✓ | ✗✗✗ | ✗✗✗ |
| KD 0.5 | ✓✓✓ | ✓✓✓ | ✓✓✓ | ✓✓✓ | ✓✓✓ | ✗✗✗ |
| KD 0.9 | ✓✓✓ | ✓✓✓ | ✓✓✓ | ✓✓✓ | ✓✓✓ | ✗✗✗ |

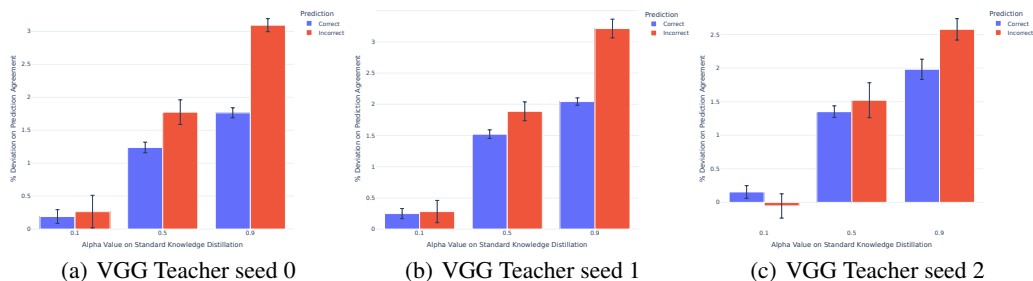

(a) VGG Teacher seed 0      (b) VGG Teacher seed 1      (c) VGG Teacher seed 2

Figure 17: Prediction agreement difference of student models in standard KD to the highest performing control baseline with respect to correct prediction agreement (blue) and incorrect prediction agreement (red), error bars are $\pm$ 1 SEM for VGG19 on TinyImageNet with RandAugment.

## F.2 CIFAR10

**Training Settings:** All CIFAR10 architectures are trained with Adam optimiser with a learning rate of 0.001 and a batch size of 256 for 100 epochs. All data is normalised with a mean of 0.5 and a standard deviation of 0.5. The student vision architectures are trained with the same seeds and data orders from seeds 10-19 for the 10 models used for averaging. As aligned with all experiments we conduct, this is repeated for the three teachers trained on seeds 0-2.

**Justification:** This setup allows for a fair analysis of Knowledge Distillation as its role is isolated in the training process. Other than the architecture's implicit bias towards the problem, which affects its performance (loss and accuracy), there are no confounding factors that could influence Knowledge Distillation.

**Findings:** We find that the teacher models often significantly transfer knowledge to the student model, and this coincides with the teacher's high loss on the training dataset. The ResNet has the lowest loss and no transfer, the VGG has a higher loss and some transfer, and the ViT has the highest loss and the most transfer. However, when knowledge is transferred, it often has a negative asymmetric payoff towards agreement between the teacher and the student on incorrect predictions.

### F.2.1 RESNET18

**Findings:** For the ResNet18 on CIFAR10, we observe that the teacher seeds, Table 57, obtain a very low train loss of $1^{-5}$ and a train accuracy of 1. This train performance coincides with a high test accuracy of circa 0.86, resulting in a generalisation gap of circa 0.14. Table 61 shows no significant knowledge transfer across teacher seeds.

Due to the low train loss on the teacher seed, the teacher model is a nearly identical representation of the training labels, meaning there is low utility in the teacher model. As we observe, the controls of the models trained in the SIDDO condition is functionally different from the teacher, Tables 58, 59 and 60; despite having the same initialisation and only changing the data order, it is not a surprise that Knowledge Distillation in the setup does not add anything as the teacher is essentially the label, and thus creates a similar setup to the SIDDO condition.

Table 57: Teacher Performance on Train and Test Data

| Teacher Seed | Train Loss | Train Accuracy | Test Loss | Test Accuracy |
|---|---|---|---|---|
| 0 | 0.000010 | 1.000000 | 0.869184 | 0.862100 |
| 1 | 0.000006 | 1.000000 | 0.833735 | 0.867200 |
| 2 | 0.000030 | 1.000000 | 0.739927 | 0.867000 |

Table 58: ResNet18 on CIFAR10 mean and $\pm$ 1 SEM reported from 10 runs with Teacher Seed 0. **Bold** values are best performing based on the mean.

| Metrics | Control | Knowledge Distillation | | | Random Control Distillation | | |
|---|---|---|---|---|---|---|---|
| | SIDDO | 0.1 | 0.5 | 0.9 | 0.1 | 0.5 | 0.9 |
| Activation Distance ($\downarrow$) | 0.174±0.004 | 0.175±0.003 | **0.172**±0.003 | 0.174±0.004 | 0.244±0.004 | 0.538±0.001 | 0.843±0.000 |
| Rank Disagreement ($\downarrow$) | 0.659±0.004 | 0.659±0.002 | 0.656±0.003 | **0.655**±0.003 | 0.795±0.001 | 0.802±0.002 | 0.807±0.002 |
| Prediction Disagreement ($\downarrow$) | 0.128±0.003 | 0.129±0.002 | **0.127**±0.003 | 0.128±0.003 | 0.131±0.003 | 0.143±0.002 | 0.150±0.001 |
| JS Divergence ($\downarrow$) | 0.070±0.002 | 0.070±0.001 | 0.069±0.002 | **0.068**±0.002 | 0.097±0.002 | 0.229±0.001 | 0.432±0.000 |
| Accuracy ($\uparrow$) | 0.861±0.003 | 0.862±0.002 | 0.862±0.002 | 0.862±0.003 | **0.865**±0.003 | 0.856±0.002 | 0.854±0.001 |
| Loss ($\downarrow$) | 0.961±0.025 | 0.903±0.018 | 0.895±0.028 | 0.827±0.026 | **0.539**±0.012 | 0.902±0.004 | 1.772±0.001 |

Table 59: ResNet18 on CIFAR10 mean and $\pm$ 1 SEM reported from 10 runs with Teacher Seed 1. **Bold** values are best performing based on the mean.

| Metrics | Control | Knowledge Distillation | | | Random Control Distillation | | |
|---|---|---|---|---|---|---|---|
| | SIDDO | 0.1 | 0.5 | 0.900 | 0.1 | 0.5 | 0.900 |
| Activation Distance ($\downarrow$) | 0.167±0.003 | **0.164**±0.002 | 0.165±0.003 | 0.165±0.002 | 0.240±0.004 | 0.533±0.001 | 0.841±0.000 |
| Rank Disagreement ($\downarrow$) | 0.653±0.002 | **0.649**±0.003 | 0.650±0.003 | 0.650±0.003 | 0.796±0.001 | 0.803±0.001 | 0.807±0.001 |
| Prediction Disagreement ($\downarrow$) | 0.122±0.002 | **0.120**±0.002 | 0.121±0.002 | **0.120**±0.002 | 0.126±0.003 | 0.134±0.002 | 0.139±0.001 |
| JS Divergence ($\downarrow$) | 0.066±0.001 | 0.065±0.001 | 0.065±0.001 | **0.064**±0.001 | 0.095±0.002 | 0.226±0.001 | 0.430±0.000 |
| Accuracy ($\uparrow$) | 0.865±0.002 | 0.867±0.002 | 0.866±0.002 | **0.867**±0.002 | 0.866±0.003 | 0.860±0.002 | 0.859±0.001 |
| Loss ($\downarrow$) | 0.858±0.028 | 0.877±0.029 | 0.824±0.022 | 0.816±0.022 | **0.533**±0.012 | 0.896±0.003 | 1.767±0.001 |

Table 60: ResNet18 on CIFAR10 mean and $\pm$ 1 SEM reported from 10 runs with Teacher Seed 2. **Bold** values are best performing based on the mean.

| Metrics | Control | Knowledge Distillation | | | Random Control Distillation | | |
|---|---|---|---|---|---|---|---|
| | SIDDO | 0.1 | 0.5 | 0.9 | 0.1 | 0.5 | 0.9 |
| Activation Distance ($\downarrow$) | **0.166**±0.002 | 0.169±0.004 | 0.167±0.004 | 0.172±0.004 | 0.242±0.003 | 0.533±0.001 | 0.839±0.000 |
| Rank Disagreement ($\downarrow$) | 0.646±0.002 | 0.647±0.003 | **0.638**±0.004 | 0.646±0.004 | 0.799±0.002 | 0.803±0.002 | 0.805±0.002 |
| Prediction Disagreement ($\downarrow$) | **0.122**±0.002 | 0.124±0.003 | 0.124±0.003 | 0.127±0.003 | 0.132±0.003 | 0.140±0.001 | 0.142±0.001 |
| JS Divergence ($\downarrow$) | 0.065±0.001 | 0.066±0.002 | **0.064**±0.002 | 0.067±0.002 | 0.096±0.001 | 0.226±0.001 | 0.429±0.000 |
| Accuracy ($\uparrow$) | 0.865±0.002 | **0.864**±0.002 | **0.864**±0.003 | 0.861±0.003 | 0.862±0.003 | 0.857±0.001 | 0.857±0.002 |
| Loss ($\downarrow$) | 0.892±0.025 | 0.887±0.027 | 0.803±0.026 | 0.798±0.023 | **0.549**±0.010 | 0.900±0.004 | 1.769±0.001 |

Table 61: ResNet18 on CIFAR10 significance testing. ✓indicates significant results compared to controls, whereas ✗indicates insignificant results compared to controls. Each tick represents a teacher (seeds 0 to 2, left to right).

| | Activation Distance | Rank Disagreement | Prediction Disagreement | JS Divergence | Accuracy | Loss |
|---|---|---|---|---|---|---|
| KD 0.1 | ✗✗✗ | ✗✗✗ | ✗✗✗ | ✗✗✗ | ✗✗✗ | ✗✗✗ |
| KD 0.5 | ✗✗✗ | ✗✗✗ | ✗✗✗ | ✗✗✗ | ✗✗✗ | ✗✗✗ |
| KD 0.9 | ✗✗✗ | ✗✗✗ | ✗✗✗ | ✗✗✗ | ✗✗✗ | ✗✗✗ |

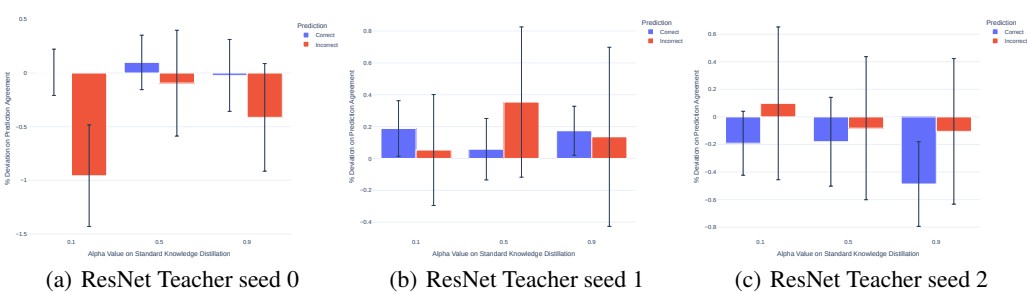

(a) ResNet Teacher seed 0      (b) ResNet Teacher seed 1      (c) ResNet Teacher seed 2

Figure 18: Prediction agreement difference of student models in standard KD to the highest performing control baseline with respect to correct prediction agreement (blue) and incorrect prediction agreement (red), error bars are $\pm$ 1 SEM for ResNet on CIFAR10.

## F.2.2 VGG19

**Findings:** For the VGG19 on CIFAR10, we observe that the teacher seeds, Table 62, obtain a low train loss of circa 0.01 and a train accuracy of approximately 0.996. This train performance coincides with a high test accuracy of circa 0.86, resulting in a generalisation gap of circa 0.14. Table 66 shows a significant knowledge transfer with regard to Rank Disagreement for all teacher seeds when alpha is at 0.9.

At alpha 0.9 for teacher seed 0 and 2, there is an increase in agreement between the student and teacher on incorrect predictions over the correct predictions, Figure 19, which corresponds with the knowledge transfer. This result coincides with teachers seed 0 and 2 having a higher train loss than teacher seed 1, indicating that the teacher train loss plays an important role in knowledge transfer. For teacher seed 1, Figure 19, there is no significant increase in correct or incorrect prediction agreement between the student model and the teacher due to the deviation in the SEM.

Table 62: Teacher Performance on Train and Test Data

| Teacher Seed | Train Loss | Train Accuracy | Test Loss | Test Accuracy |
|---|---|---|---|---|
| 0 | 0.011608 | 0.996760 | 0.858675 | 0.863900 |
| 1 | 0.009228 | 0.997080 | 0.798530 | 0.860800 |
| 2 | 0.012352 | 0.996420 | 0.801562 | 0.867100 |

Table 63: VGG19 on CIFAR10 mean and $\pm 1$ SEM reported from 10 runs with Teacher Seed 0. **Bold** values are best performing based on the mean.

| Metrics | Control | Knowledge Distillation | | | Random Control Distillation | | |
|---|---|---|---|---|---|---|---|
| | SIDDO | 0.1 | 0.5 | 0.9 | 0.1 | 0.5 | 0.9 |
| Activation Distance (↓) | 0.206±0.006 | 0.199±0.003 | 0.203±0.003 | **0.197**±0.005 | 0.264±0.003 | 0.541±0.001 | 0.842±0.000 |
| Rank Disagreement (↓) | 0.701±0.008 | 0.705±0.007 | 0.658±0.006 | **0.640**±0.009 | 0.811±0.005 | 0.819±0.004 | 0.819±0.006 |
| Prediction Disagreement (↓) | 0.152±0.004 | 0.147±0.002 | 0.151±0.002 | **0.146**±0.004 | 0.148±0.002 | **0.146**±0.001 | 0.150±0.001 |
| JS Divergence (↓) | 0.090±0.003 | 0.085±0.001 | 0.086±0.002 | **0.083**±0.002 | 0.109±0.001 | 0.230±0.001 | 0.429±0.000 |
| Accuracy (↑) | 0.864±0.003 | 0.869±0.002 | 0.867±0.002 | 0.869±0.003 | 0.870±0.002 | **0.871**±0.001 | 0.868±0.002 |
| Loss (↓) | 0.849±0.027 | 0.725±0.010 | 0.676±0.011 | 0.649±0.015 | **0.562**±0.008 | 0.880±0.003 | 1.762±0.002 |

Table 64: VGG19 on CIFAR10 mean and $\pm 1$ SEM reported from 10 runs with Teacher Seed 1. **Bold** values are best performing based on the mean.

| Metrics | Control | Knowledge Distillation | | | Random Control Distillation | | |
|---|---|---|---|---|---|---|---|
| | SIDDO | 0.1 | 0.5 | 0.9 | 0.1 | 0.5 | 0.9 |
| Activation Distance (↓) | **0.199**±0.002 | 0.202±0.002 | 0.202±0.004 | 0.201±0.003 | 0.263±0.002 | 0.543±0.001 | 0.842±0.000 |
| Rank Disagreement (↓) | 0.726±0.006 | 0.684±0.005 | 0.662±0.008 | **0.639**±0.009 | 0.803±0.003 | 0.801±0.005 | 0.810±0.005 |
| Prediction Disagreement (↓) | 0.147±0.002 | 0.150±0.001 | 0.150±0.003 | 0.149±0.002 | **0.148**±0.002 | 0.149±0.001 | 0.153±0.001 |
| JS Divergence (↓) | 0.086±0.001 | 0.087±0.001 | 0.086±0.002 | **0.085**±0.001 | 0.107±0.001 | 0.230±0.001 | 0.428±0.000 |
| Accuracy (↑) | 0.868±0.002 | 0.866±0.001 | 0.865±0.003 | 0.866±0.002 | **0.870**±0.002 | 0.869±0.002 | 0.866±0.002 |
| Loss (↓) | 0.799±0.018 | 0.735±0.009 | 0.680±0.013 | 0.666±0.014 | **0.562**±0.007 | 0.887±0.004 | 1.762±0.002 |

Table 65: VGG19 on CIFAR10 mean and $\pm 1$ SEM reported from 10 runs with Teacher Seed 2. **Bold** values are best performing based on the mean.

| Metrics | Control | Knowledge Distillation | | | Random Control Distillation | | |
|---|---|---|---|---|---|---|---|
| | SIDDO | 0.1 | 0.5 | 0.9 | 0.1 | 0.5 | 0.9 |
| Activation Distance (↓) | 0.196±0.002 | 0.199±0.003 | 0.196±0.002 | **0.193**±0.004 | 0.258±0.002 | 0.541±0.001 | 0.844±0.000 |
| Rank Disagreement (↓) | 0.672±0.017 | 0.649±0.011 | 0.633±0.010 | **0.602**±0.015 | 0.809±0.003 | 0.817±0.005 | 0.816±0.005 |
| Prediction Disagreement (↓) | 0.142±0.001 | 0.146±0.002 | 0.143±0.001 | **0.141**±0.003 | 0.142±0.002 | 0.143±0.002 | 0.149±0.001 |
| JS Divergence (↓) | 0.084±0.001 | 0.086±0.001 | 0.083±0.001 | **0.081**±0.002 | 0.106±0.001 | 0.229±0.001 | 0.429±0.000 |
| Accuracy (↑) | 0.870±0.001 | 0.864±0.001 | 0.868±0.001 | 0.867±0.003 | **0.871**±0.001 | **0.871**±0.002 | 0.867±0.001 |
| Loss (↓) | 0.801±0.014 | 0.734±0.013 | 0.665±0.009 | 0.639±0.013 | **0.560**±0.006 | 0.884±0.003 | 1.762±0.002 |

Table 66: VGG19 on CIFAR10 significance testing. ✓indicates significant results compared to controls, whereas ✗indicates insignificant results compared to controls. Each tick represents a teacher (seeds 0 to 2, left to right).

| | Activation Distance | Rank Disagreement | Prediction Disagreement | JS Divergence | Accuracy | Loss |
|---|---|---|---|---|---|---|
| KD 0.1 | ✗✗✗ | ✗✓✗ | ✗✗✗ | ✗✗✗ | ✗✗✗ | ✗✗✗ |
| KD 0.5 | ✗✗✗ | ✓✓✗ | ✗✗✗ | ✗✗✗ | ✗✗✗ | ✗✗✗ |
| KD 0.9 | ✗✗✗ | ✓✓✓ | ✗✗✗ | ✓✗✗ | ✗✗✗ | ✗✗✗ |

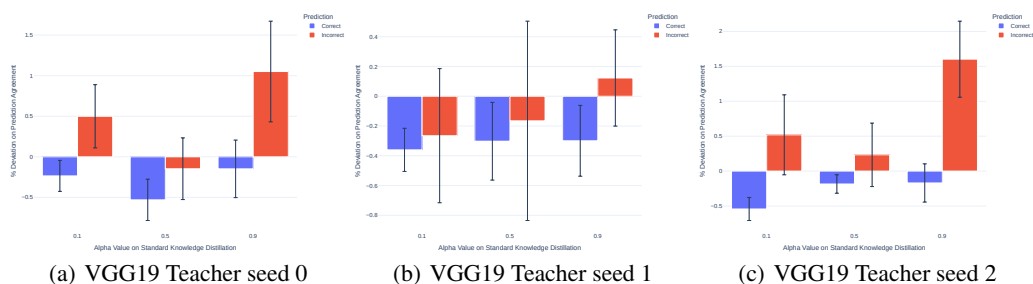

(a) VGG19 Teacher seed 0     (b) VGG19 Teacher seed 1     (c) VGG19 Teacher seed 2

Figure 19: Prediction agreement difference of student models in standard KD to the highest performing control baseline with respect to correct prediction agreement (blue) and incorrect prediction agreement (red), error bars are $\pm$ 1 SEM for VGG19 on CIFAR10.

### F.2.3 VIT

**Findings:** For the ViT on CIFAR10, we observe that the teacher seeds, Table 67, obtain a high train loss of 0.04 and a train accuracy of approximately 0.98. This train performance coincides with a test accuracy of circa 0.63, resulting in a generalisation gap of circa 0.35. Table 71 shows a significant knowledge transfer on all teacher seeds when alpha is 0.5 and 0.9. For teacher seed 0 and 1 using alpha at 0.9, where there is sizeable knowledge transfer, we observe an asymmetric knowledge transfer favouring negative transfer in Figure 20.

Table 67: Teacher Performance on Train and Test Data

| Teacher Seed | Train Loss | Train Accuracy | Test Loss | Test Accuracy |
|---|---|---|---|---|
| 0 | 0.043291 | 0.988260 | 1.864339 | 0.626900 |
| 1 | 0.056539 | 0.983160 | 1.772490 | 0.634200 |
| 2 | 0.046902 | 0.987100 | 1.714442 | 0.649600 |

Table 68: ViT on CIFAR10 mean and $\pm$ 1 SEM reported from 10 runs with Teacher Seed 0. **Bold** values are best performing based on the mean.

| Metrics | Control | Knowledge Distillation | | | Random Control Distillation | | |
|---|---|---|---|---|---|---|---|
| | SIDDO | 0.1 | 0.5 | 0.900 | 0.1 | 0.5 | 0.900 |
| Activation Distance (↓) | 0.491±0.001 | 0.487±0.002 | 0.473±0.002 | 0.470±0.001 | **0.496**±0.002 | 0.611±0.001 | 0.793±0.000 |
| Rank Disagreement (↓) | 0.734±0.001 | 0.730±0.001 | 0.724±0.001 | **0.722**±0.001 | 0.808±0.001 | 0.812±0.002 | 0.817±0.002 |
| Prediction Disagreement (↓) | 0.385±0.001 | 0.383±0.002 | 0.374±0.001 | **0.373**±0.001 | 0.383±0.002 | 0.380±0.002 | 0.386±0.001 |
| JS Divergence (↓) | 0.201±0.001 | 0.198±0.001 | 0.189±0.001 | **0.186**±0.001 | 0.206±0.001 | 0.277±0.001 | 0.411±0.000 |
| Accuracy (↑) | 0.634±0.002 | 0.634±0.002 | 0.641±0.003 | 0.637±0.002 | 0.640±0.003 | **0.641**±0.002 | 0.627±0.003 |
| Loss (↓) | 1.773±0.015 | 1.695±0.011 | 1.52±0.018 | 1.451±0.014 | **1.258**±0.012 | 1.351±0.005 | 1.943±0.002 |

Table 69: ViT on CIFAR10 mean and ± 1 SEM reported from 10 runs with Teacher Seed 1. **Bold** values are best performing based on the mean.

| Metrics | Control | Knowledge Distillation | | | Random Control Distillation | | |
|---|---|---|---|---|---|---|---|
| | SIDDO | 0.1 | 0.5 | 0.9 | 0.1 | 0.5 | 0.9 |
| Activation Distance (↓) | 0.485±0.002 | 0.477±0.002 | 0.461±0.002 | **0.455**±0.001 | 0.489±0.002 | 0.609±0.001 | 0.791±0.000 |
| Rank Disagreement (↓) | 0.733±0.001 | 0.728±0.001 | 0.717±0.001 | **0.714**±0.001 | 0.806±0.001 | 0.808±0.001 | 0.816±0.002 |
| Prediction Disagreement (↓) | 0.382±0.002 | 0.375±0.002 | 0.367±0.002 | **0.363**±0.001 | 0.379±0.002 | 0.380±0.002 | 0.382±0.001 |
| JS Divergence (↓) | 0.198±0.001 | 0.193±0.001 | 0.182±0.001 | **0.178**±0.001 | 0.202±0.001 | 0.275±0.001 | 0.410±0.000 |
| Accuracy (↑) | 0.637±0.001 | 0.643±0.003 | 0.644±0.002 | **0.648**±0.002 | 0.643±0.002 | 0.636±0.002 | 0.630±0.002 |
| Loss (↓) | 1.781±0.013 | 1.668±0.015 | 1.466±0.010 | 1.366±0.012 | **1.253**±0.008 | 1.359±0.005 | 1.942±0.001 |

Table 70: ViT on CIFAR10 mean and ± 1 SEM reported from 10 runs with Teacher Seed 2. **Bold** values are best performing based on the mean.

| Metrics | Control | Knowledge Distillation | | | Random Control Distillation | | |
|---|---|---|---|---|---|---|---|
| | SIDDO | 0.1 | 0.5 | 0.9 | 0.1 | 0.5 | 0.9 |
| Activation Distance (↓) | 0.476±0.002 | 0.468±0.002 | 0.459±0.002 | **0.456**±0.003 | 0.486±0.002 | 0.612±0.001 | 0.797±0.000 |
| Rank Disagreement (↓) | 0.730±0.001 | 0.725±0.001 | 0.720±0.001 | **0.718**±0.001 | 0.806±0.001 | 0.811±0.002 | 0.817±0.002 |
| Prediction Disagreement (↓) | 0.372±0.002 | 0.366±0.002 | 0.363±0.002 | **0.360**±0.002 | 0.371±0.002 | 0.374±0.002 | 0.375±0.002 |
| JS Divergence (↓) | 0.195±0.001 | 0.189±0.001 | 0.183±0.001 | **0.180**±0.001 | 0.201±0.001 | 0.277±0.001 | 0.413±0.000 |
| Accuracy (↑) | 0.636±0.003 | 0.641±0.003 | **0.644**±0.002 | 0.639±0.003 | 0.637±0.002 | 0.635±0.002 | 0.631±0.002 |
| Loss (↓) | 1.788±0.025 | 1.673±0.017 | 1.498±0.010 | 1.458±0.018 | **1.282**±0.008 | 1.361±0.005 | 1.942±0.002 |

Table 71: ViT on CIFAR10 significance testing. ✓indicates significant results compared to controls, whereas ✗indicates insignificant results compared to controls. Each tick represents a teacher (seeds 0 to 2, left to right).

| | Activation Distance | Rank Disagreement | Prediction Disagreement | JS Divergence | Accuracy | Loss |
|---|---|---|---|---|---|---|
| KD 0.1 | ✗✓✓ | ✓✓✓ | ✗✗✗ | ✓✓✓ | ✗✗✗ | ✗✗✗ |
| KD 0.5 | ✓✓✓ | ✓✓✓ | ✓✓✓ | ✓✓✓ | ✗✗✓ | ✗✗✗ |
| KD 0.9 | ✓✓✓ | ✓✓✓ | ✓✓✓ | ✓✓✓ | ✗✗✗ | ✗✗✗ |

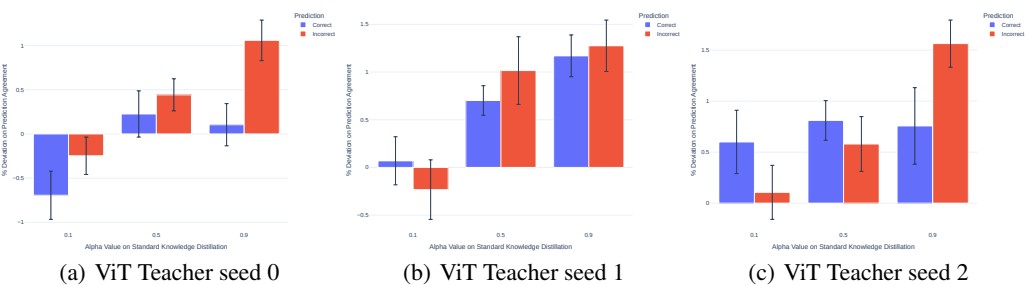

(a) ViT Teacher seed 0    (b) ViT Teacher seed 1    (c) ViT Teacher seed 2

Figure 20: Prediction agreement difference of student models in standard KD to the highest performing control baseline with respect to correct prediction agreement (blue) and incorrect prediction agreement (red), error bars are ± 1 SEM for ViT on CIFAR10.

## F.3 SVHN DATASET

**Training Settings:** All SVHN architectures are trained with Adam optimiser with a learning rate of 0.001 and a batch size of 256 for 100 epochs. All data is normalised with a mean of 0.5 and a standard deviation of 0.5. The student vision architectures are trained with the same seeds and data orders from seeds 10-19 for the 10 models used for averaging. We repeated this, in line with our other experiments for the three teachers trained on seeds 0-2.

**Justification:** This setup allows for a fair analysis of Knowledge Distillation as its role is isolated in the training process. Other than the architecture's implicit bias towards the problem, which affects its performance (loss and accuracy), there are no confounding factors that could influence Knowledge Distillation.

**Findings:** The teacher models often significantly transfer knowledge to the student model. However, the knowledge transfer is often inconsistent, and when transferred, it often has an asymmetric negative payoff.

## F.3.1 RESNET18

**Findings:** For the ResNet on SVHN, we observe that the teacher seeds, Table 72, obtain a range of train loss values of 0.000646, 0.000061 and 0.004657 for teacher seeds 0, 1, and 2, respectively. The train accuracies are approximately 0.99. This train performance coincides with a test accuracy of circa 0.95, resulting in a generalisation gap of circa 0.04.

The teacher model with a higher training loss (seed 2) has significant knowledge transfer, see Table 76, for all functional similarity metrics across alpha values 0.1, 0.5 and 0.9, except for Prediction Disagreement when alpha was 0.1. In this case, we also observe a large asymmetric payoff in prediction agreement, significantly favouring incorrect predictions, Figure 21. Whereas teacher seed 0 has a train loss of 0.000061 and has no significant transfer with alpha values of 0.1 and 0.5. However, with an alpha of 0.9, it does have a significant transfer across metrics except for Prediction Disagreement, see Table 76. When alpha is 0.9, we observe an asymmetric payoff in prediction agreement, significantly favouring incorrect predictions. For teacher seed 0, which has a train loss of 0.000646, we observe significant knowledge transfer when alpha is 0.5 and 0.9, coinciding with an asymmetric payoff in prediction agreement, favouring incorrect predictions.

Table 72: Teacher Performance on Train and Test Data for ResNet18 on SVHN

| Teacher Seed | Train Loss | Train Accuracy | Test Loss | Test Accuracy |
|---|---|---|---|---|
| 0 | 0.000646 | 0.999850 | 0.381410 | 0.951829 |
| 1 | 0.000061 | 0.999973 | 0.331054 | 0.952251 |
| 2 | 0.004657 | 0.998580 | 0.309702 | 0.947104 |

Table 73: ResNet18 on SVHN mean and $\pm$ 1 SEM reported from 10 runs with Teacher Seed 0. **Bold** values are best performing based on the mean. The direction of the arrow ($\uparrow\downarrow$) dictates the direction of the most favourable score per metric.

| Metrics | Control | Knowledge Distillation | | | Random Control Distillation | | |
|---|---|---|---|---|---|---|---|
| | SIDDO | 0.1 | 0.5 | 0.9 | 0.1 | 0.5 | 0.9 |
| Activation Distance ($\downarrow$) | 0.063±0.002 | 0.064±0.001 | 0.060±0.001 | **0.059**±0.001 | 0.144±0.001 | 0.493±0.000 | 0.849±0.000 |
| Rank Disagreement ($\downarrow$) | 0.696±0.003 | 0.688±0.004 | 0.684±0.003 | **0.681**±0.003 | 0.800±0.002 | 0.798±0.002 | 0.802±0.003 |
| Prediction Disagreement ($\downarrow$) | 0.045±0.001 | 0.046±0.001 | 0.043±0.001 | **0.042**±0.001 | **0.042**±0.001 | 0.043±0.001 | 0.046±0.001 |
| JS Divergence ($\downarrow$) | 0.025±0.001 | 0.025±0.001 | 0.023±0.001 | **0.022**±0.000 | 0.053±0.000 | 0.201±0.000 | 0.431±0.000 |
| Accuracy ($\uparrow$) | 0.952±0.001 | 0.951±0.001 | 0.954±0.001 | 0.954±0.001 | **0.957**±0.001 | **0.957**±0.001 | 0.955±0.001 |
| Loss ($\downarrow$) | 0.385±0.011 | 0.344±0.008 | 0.310±0.006 | 0.293±0.004 | **0.236**±0.003 | 0.692±0.001 | 1.698±0.001 |

Table 74: ResNet18 on SVHN mean and $\pm$ 1 SEM reported from 10 runs with Teacher Seed 1. **Bold** values are best performing based on the mean.

| Metrics | Control | Knowledge Distillation | | | Random Knowledge Distillation | | |
|---|---|---|---|---|---|---|---|
| | SIDDO | 0.1 | 0.5 | 0.9 | 0.1 | 0.5 | 0.9 |
| Activation Distance ($\downarrow$) | 0.059±0.001 | 0.058±0.001 | 0.058±0.001 | **0.056**±0.001 | 0.141±0.001 | 0.494±0.001 | 0.848±0.000 |
| Rank Disagreement ($\downarrow$) | 0.690±0.002 | 0.688±0.003 | 0.687±0.003 | **0.682**±0.002 | 0.799±0.002 | 0.799±0.002 | 0.800±0.003 |
| Prediction Disagreement ($\downarrow$) | 0.042±0.001 | 0.042±0.001 | 0.042±0.001 | **0.040**±0.001 | **0.040**±0.001 | 0.044±0.001 | 0.046±0.000 |
| JS Divergence ($\downarrow$) | 0.023±0.000 | 0.023±0.000 | **0.022**±0.001 | **0.022**±0.000 | 0.052±0.000 | 0.201±0.000 | 0.431±0.000 |
| Accuracy ($\uparrow$) | 0.953±0.001 | 0.953±0.001 | 0.953±0.001 | 0.954±0.001 | **0.958**±0.001 | 0.954±0.001 | 0.953±0.001 |
| Loss ($\downarrow$) | 0.366±0.008 | 0.354±0.008 | 0.328±0.006 | 0.316±0.004 | **0.236**±0.002 | 0.698±0.002 | 1.698±0.001 |

Table 75: ResNet18 on SVHN mean and $\pm$ 1 SEM reported from 10 runs with Teacher Seed 2. **Bold** values are best performing based on the mean.

| Metrics | Control | Knowledge Distillation | | | Random Control Distillation | | |
|---|---|---|---|---|---|---|---|
| | SIDDO | 0.1 | 0.5 | 0.900 | 0.1 | 0.5 | 0.900 |
| Activation Distance ($\downarrow$) | 0.068±0.001 | 0.063±0.001 | 0.059±0.000 | **0.058**±0.000 | 0.146±0.001 | 0.489±0.001 | 0.843±0.000 |
| Rank Disagreement ($\downarrow$) | 0.713±0.003 | 0.667±0.003 | 0.648±0.003 | **0.643**±0.001 | 0.800±0.003 | 0.800±0.004 | 0.799±0.003 |
| Prediction Disagreement ($\downarrow$) | 0.048±0.001 | 0.045±0.001 | 0.042±0.000 | **0.041**±0.000 | 0.046±0.001 | 0.048±0.001 | 0.052±0.001 |
| JS Divergence ($\downarrow$) | 0.026±0.000 | 0.023±0.000 | 0.021±0.000 | **0.020**±0.000 | 0.053±0.001 | 0.199±0.000 | 0.427±0.000 |
| Accuracy ($\uparrow$) | 0.952±0.001 | 0.955±0.001 | **0.957**±0.000 | **0.957**±0.000 | 0.956±0.001 | **0.957**±0.001 | 0.953±0.001 |
| Loss ($\downarrow$) | 0.370±0.008 | 0.256±0.006 | 0.226±0.002 | **0.216**±0.001 | 0.239±0.003 | 0.692±0.002 | 1.700±0.001 |

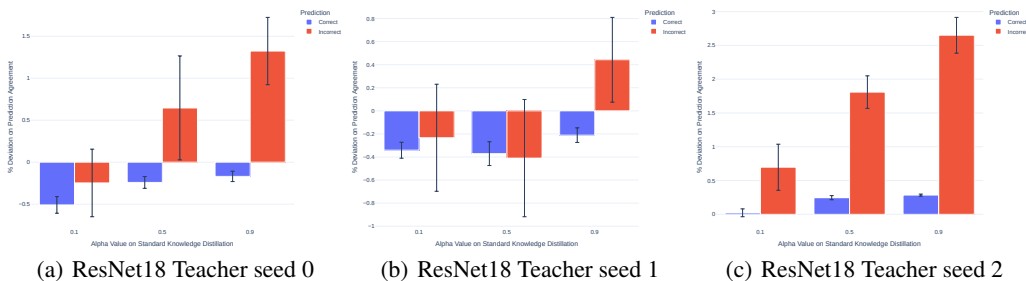

(a) ResNet18 Teacher seed 0     (b) ResNet18 Teacher seed 1     (c) ResNet18 Teacher seed 2

Figure 21: Prediction agreement difference of student models in standard KD to the highest performing control baseline with respect to correct prediction agreement (blue) and incorrect prediction agreement (red), error bars are $\pm$ 1 SEM for ResNet18 on SVHN.

Table 76: ResNet18 on SVHN significance testing. ✓indicates significant results compared to controls, whereas ✗indicates insignificant results compared to controls. Each tick represents a teacher (seeds 0 to 2, left to right).

| | Activation Distance | Rank Disagreement | Prediction Disagreement | JS Divergence | Accuracy | Loss |
|---|---|---|---|---|---|---|
| KD 0.1 | ✗✗✓ | ✗✗✓ | ✗✗✗ | ✗✗✓ | ✗✗✗ | ✗✗✗ |
| KD 0.5 | ✗✗✓ | ✓✗✓ | ✗✗✓ | ✓✗✓ | ✗✗✗ | ✗✗✓ |
| KD 0.9 | ✓✓✓ | ✓✓✓ | ✗✗✓ | ✓✓✓ | ✗✗✗ | ✗✗✓ |

### F.3.2 VGG19

**Findings:** For the VGG19 on SVHN, we record a low train loss from we observe that the teacher seeds, Table 77, obtain a range of train loss values of 0.004511, 0.002757 and 0.00374 for teacher seeds 0, 1, and 2, respectively. The train accuracies are approximately 0.99. This train performance coincides with a test accuracy of circa 0.95, resulting in a generalisation gap of circa 0.04.

The teacher model with a higher training loss (seed 2) has significant knowledge transfer, see Table 81, for only Rank Disagreement, across alpha values 0.1, 0.5 and 0.9. Due to limited statically significant functional transfer across metrics for this seed, we observe a small but inconsistent asymmetric payoff in prediction agreement, slightly favouring incorrect predictions, Figure 22. The story is very similar across the other teacher seeds; we see marginal functional transfer, and where a transfer is higher, we see negative transfer, but where it is marginal or largely insignificant, we see no preference for knowledge transfer, showing that in this case knowledge sharing can not be attributed to improved performance.

Table 77: Teacher Performance on Train and Test Data for VGG19 on SVHN

| Teacher Seed | Train Loss | Train Accuracy | Test Loss | Test Accuracy |
|---|---|---|---|---|
| 0 | 0.004511 | 0.998649 | 0.343982 | 0.952827 |
| 1 | 0.002757 | 0.999290 | 0.347466 | 0.948794 |
| 2 | 0.003741 | 0.998935 | 0.313836 | 0.953596 |

Table 78: VGG19 on SVHN mean and $\pm$ 1 SEM reported from 10 runs with Teacher Seed 0. **Bold** values are best performing based on the mean. The direction of the arrow ($\uparrow\downarrow$) dictates the direction of the most favourable score per metric.

| Metrics | Control | Knowledge Distillation | | | Random Control Distillation | | |
|---|---|---|---|---|---|---|---|
| | SIDDO | 0.1 | 0.5 | 0.9 | 0.1 | 0.5 | 0.9 |
| Activation Distance ($\downarrow$) | 0.065±0.001 | **0.064**±0.001 | 0.066±0.002 | 0.065±0.001 | 0.151±0.001 | 0.494±0.001 | 0.848±0.000 |
| Rank Disagreement ($\downarrow$) | 0.708±0.005 | 0.660±0.011 | 0.637±0.009 | **0.603**±0.011 | 0.799±0.005 | 0.812±0.006 | 0.805±0.007 |
| Prediction Disagreement ($\downarrow$) | 0.047±0.001 | 0.046±0.000 | 0.047±0.001 | 0.047±0.001 | 0.047±0.001 | **0.045**±0.001 | 0.046±0.000 |
| JS Divergence ($\downarrow$) | 0.028±0.000 | **0.027**±0.000 | **0.027**±0.001 | **0.027**±0.001 | 0.057±0.000 | 0.201±0.000 | 0.429±0.000 |
| Accuracy ($\uparrow$) | 0.954±0.001 | 0.954±0.001 | 0.953±0.001 | 0.953±0.001 | 0.955±0.001 | **0.956**±0.001 | **0.956**±0.000 |
| Loss ($\downarrow$) | 0.349±0.006 | 0.292±0.005 | 0.282±0.008 | 0.275±0.003 | **0.263**±0.002 | 0.698±0.002 | 1.696±0.001 |

Table 79: VGG19 on SVHN mean and $\pm$ 1 SEM reported from 10 runs with Teacher Seed 1. **Bold** values are best performing based on the mean. The direction of the arrow ($\uparrow\downarrow$) dictates the direction of the most favourable score per metric.

| Metrics | Control | Knowledge Distillation | | | Random Control Distillation | | |
|---|---|---|---|---|---|---|---|
| | SIDDO | 0.1 | 0.5 | 0.9 | 0.1 | 0.5 | 0.9 |
| Activation Distance ($\downarrow$) | 0.069±0.001 | 0.067±0.001 | 0.067±0.002 | **0.066**±0.001 | 0.154±0.001 | 0.496±0.001 | 0.846±0.000 |
| Rank Disagreement ($\downarrow$) | 0.758±0.009 | 0.710±0.006 | 0.663±0.011 | **0.652**±0.009 | 0.814±0.002 | 0.796±0.007 | 0.808±0.007 |
| Prediction Disagreement ($\downarrow$) | 0.051±0.001 | 0.050±0.000 | 0.050±0.001 | 0.049±0.001 | 0.050±0.000 | 0.049±0.001 | **0.048**±0.000 |
| JS Divergence ($\downarrow$) | 0.030±0.000 | 0.029±0.000 | 0.029±0.001 | **0.028**±0.001 | 0.058±0.000 | 0.201±0.000 | 0.428±0.000 |
| Accuracy ($\uparrow$) | 0.952±0.001 | 0.953±0.000 | 0.953±0.001 | 0.954±0.001 | 0.953±0.001 | 0.955±0.001 | **0.956**±0.000 |
| Loss ($\downarrow$) | 0.353±0.008 | 0.304±0.004 | 0.274±0.006 | 0.269±0.005 | **0.268**±0.003 | 0.701±0.002 | 1.695±0.001 |

Table 80: VGG19 on SVHN mean and $\pm$ 1 SEM reported from 10 runs with Teacher Seed 2. **Bold** values are best performing based on the mean.

| Metrics | Control | Knowledge Distillation | | | Random Control Distillation | | |
|---|---|---|---|---|---|---|---|
| | SIDDO | 0.1 | 0.5 | 0.9 | 0.1 | 0.5 | 0.9 |
| Activation Distance ($\downarrow$) | 0.065±0.001 | 0.067±0.001 | 0.065±0.001 | **0.064**±0.002 | 0.148±0.000 | 0.493±0.001 | 0.847±0.000 |
| Rank Disagreement ($\downarrow$) | 0.733±0.009 | 0.680±0.011 | 0.647±0.008 | **0.600**±0.013 | 0.804±0.003 | 0.808±0.007 | 0.809±0.006 |
| Prediction Disagreement ($\downarrow$) | 0.048±0.001 | 0.049±0.001 | 0.047±0.001 | 0.046±0.001 | 0.045±0.000 | **0.044**±0.001 | 0.046±0.000 |
| JS Divergence ($\downarrow$) | 0.028±0.000 | 0.028±0.001 | 0.027±0.000 | **0.026**±0.001 | 0.055±0.000 | 0.200±0.000 | 0.429±0.000 |
| Accuracy ($\uparrow$) | 0.952±0.001 | 0.952±0.001 | 0.953±0.001 | 0.954±0.001 | 0.956±0.000 | **0.957**±0.001 | 0.956±0.001 |
| Loss ($\downarrow$) | 0.358±0.007 | 0.301±0.006 | 0.284±0.005 | 0.265±0.010 | **0.258**±0.001 | 0.697±0.002 | 1.696±0.001 |

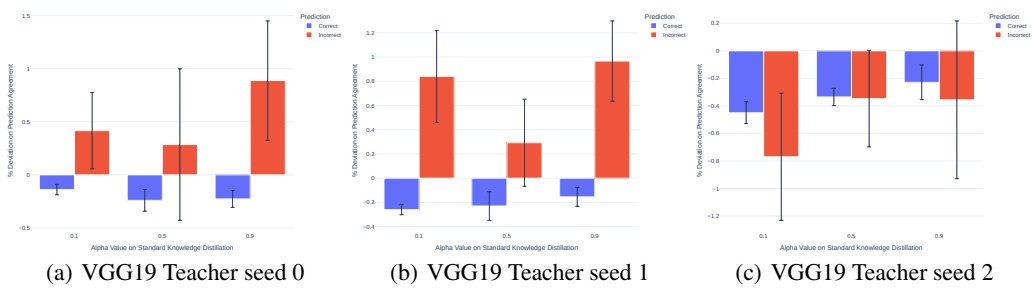

(a) VGG19 Teacher seed 0    (b) VGG19 Teacher seed 1    (c) VGG19 Teacher seed 2

Figure 22: Prediction agreement difference of student models in standard KD to the highest performing control baseline with respect to correct prediction agreement (blue) and incorrect prediction agreement (red), error bars are $\pm$ 1 SEM for VGG19 on SVHN.

Table 81: VGG19 on SVHN significance testing. ✓indicates significant results compared to controls, whereas ✗indicates insignificant results compared to controls. Each tick represents a teacher (seeds 0 to 2, left to right).

| | Activation Distance | Rank Disagreement | Prediction Disagreement | JS Divergence | Accuracy | Loss |
|---|---|---|---|---|---|---|
| KD 0.1 | ✗✗✗ | ✓✓✓ | ✗✗✗ | ✗✓✗ | ✗✗✗ | ✗✗✗ |
| KD 0.5 | ✗✗✗ | ✓✓✓ | ✗✗✗ | ✗✓✗ | ✗✗✗ | ✗✗✗ |
| KD 0.9 | ✗✓✗ | ✓✓✓ | ✗✗✗ | ✗✓✗ | ✗✗✗ | ✗✗✗ |

### F.3.3 VIT

**Findings:** For the ViT on SVHN, we record a train loss from we observe that the teacher seeds, Table 82, obtain a range of train loss values of 0.018473, 0.019402 and 0.018580 for teacher seeds 0, 1, and 2, respectively. The train accuracies are approximately 0.99. This train performance coincides with a test accuracy of circa 0.85, resulting in a generalisation gap of circa 0.14.

The teacher model with a higher training loss (seed 1) has significant knowledge transfer, see Table 86, for only Activation Distance, Rank Disagreement and JS Divergence across alpha values 0.5 and 0.9. In this case, we observe a small but inconsistent asymmetric payoff in prediction agreement, slightly favouring incorrect predictions, Figure 23. The story is very similar across the other teacher seeds; we see marginal functional transfer, and where a transfer is higher, we see negative transfer, but where it is marginal or largely insignificant, we see no real preference for knowledge transfer, showing that in this case knowledge sharing can not be attributed to improved performance.

Table 82: Teacher Performance on Train and Test Data

| Teacher Seed | Train Loss | Train Accuracy | Test Loss | Test Accuracy |
|---|---|---|---|---|
| 0 | 0.018473 | 0.994417 | 0.774354 | 0.854564 |
| 1 | 0.019402 | 0.994963 | 0.711637 | 0.855025 |
| 2 | 0.018580 | 0.994635 | 0.692686 | 0.860633 |

Table 83: ViT on SVHN mean and $\pm$ 1 SEM reported from 10 runs with Teacher Seed 0. **Bold** values are best performing based on the mean. The direction of the arrow ($\uparrow\downarrow$) dictates the direction of the most favourable score per metric.

| Metrics | Control | Knowledge Distillation | | | Random Control Distillation | | |
|---|---|---|---|---|---|---|---|
| | SIDDO | 0.1 | 0.5 | 0.9 | 0.1 | 0.5 | 0.9 |
| Activation Distance ($\downarrow$) | 0.219±0.002 | 0.220±0.002 | 0.215±0.002 | **0.211**±0.001 | 0.273±0.002 | 0.535±0.001 | 0.829±0.000 |
| Rank Disagreement ($\downarrow$) | 0.741±0.001 | 0.741±0.001 | 0.736±0.001 | **0.732**±0.001 | 0.801±0.001 | 0.806±0.003 | 0.805±0.002 |
| Prediction Disagreement ($\downarrow$) | 0.165±0.002 | 0.165±0.002 | 0.162±0.002 | **0.159**±0.001 | 0.162±0.001 | 0.160±0.001 | 0.161±0.001 |
| JS Divergence ($\downarrow$) | 0.0910±0.001 | 0.091±0.001 | 0.088±0.001 | **0.085**±0.001 | 0.110±0.001 | 0.227±0.001 | 0.422±0.000 |
| Accuracy ($\uparrow$) | 0.857±0.003 | 0.856±0.003 | 0.856±0.002 | 0.858±0.002 | 0.858±0.002 | **0.860**±0.002 | 0.859±0.002 |
| Loss ($\downarrow$) | 0.707±0.013 | 0.698±0.012 | 0.651±0.013 | 0.608±0.006 | **0.560**±0.008 | 0.896±0.004 | 1.771±0.002 |

Table 84: ViT on SVHN mean and $\pm$ 1 SEM reported from 10 runs with Teacher Seed 1. **Bold** values are best performing based on the mean. The direction of the arrow ($\uparrow\downarrow$) dictates the direction of the most favourable score per metric.

| Metrics | Control | Knowledge Distillation | | | Random Control Distillation | | |
|---|---|---|---|---|---|---|---|
| | SIDDO | 0.1 | 0.5 | 0.9 | 0.1 | 0.5 | 0.9 |
| Activation Distance ($\downarrow$) | 0.216±0.002 | 0.212±0.001 | 0.208±0.002 | **0.206**±0.002 | 0.266±0.002 | 0.529±0.001 | 0.825±0.001 |
| Rank Disagreement ($\downarrow$) | 0.745±0.001 | 0.745±0.001 | 0.737±0.001 | **0.735**±0.001 | 0.801±0.001 | 0.805±0.003 | 0.804±0.003 |
| Prediction Disagreement ($\downarrow$) | 0.162±0.001 | 0.159±0.001 | 0.157±0.001 | **0.156**±0.001 | 0.158±0.001 | 0.156±0.001 | 0.164±0.005 |
| JS Divergence ($\downarrow$) | 0.089±0.001 | 0.086±0.000 | 0.084±0.001 | **0.082**±0.001 | 0.106±0.001 | 0.224±0.001 | 0.420±0.001 |
| Accuracy ($\uparrow$) | 0.856±0.003 | 0.861±0.001 | 0.863±0.003 | 0.864±0.002 | 0.863±0.003 | **0.865**±0.002 | 0.854±0.007 |
| Loss ($\downarrow$) | 0.722±0.011 | 0.680±0.009 | 0.603±0.012 | 0.574±0.010 | **0.543**±0.010 | 0.886±0.004 | 1.777±0.007 |

Table 85: ViT on SVHN mean and $\pm$ 1 SEM reported from 10 runs with Teacher Seed 2. **Bold** values are best performing based on the mean.

| Metrics | Control | Knowledge Distillation | | | Random Control Distillation | | |
|---|---|---|---|---|---|---|---|
| | SIDDO | 0.1 | 0.5 | 0.9 | 0.1 | 0.5 | 0.9 |
| Activation Distance ($\downarrow$) | 0.212±0.001 | 0.206±0.002 | 0.206±0.002 | **0.204**±0.001 | 0.265±0.001 | 0.532±0.001 | 0.828±0.000 |
| Rank Disagreement ($\downarrow$) | 0.742±0.001 | 0.735±0.001 | 0.731±0.001 | **0.728**±0.001 | 0.802±0.001 | 0.803±0.001 | 0.804±0.002 |
| Prediction Disagreement ($\downarrow$) | 0.160±0.001 | 0.155±0.001 | 0.155±0.001 | 0.153±0.001 | 0.156±0.001 | 0.153±0.001 | **0.152**±0.001 |
| JS Divergence ($\downarrow$) | 0.087±0.001 | 0.084±0.001 | 0.083±0.001 | **0.081**±0.001 | 0.106±0.000 | 0.225±0.001 | 0.421±0.000 |
| Accuracy ($\uparrow$) | 0.856±0.001 | 0.861±0.002 | 0.859±0.002 | 0.860±0.002 | 0.863±0.001 | **0.866**±0.002 | 0.864±0.001 |
| Loss ($\downarrow$) | 0.730±0.011 | 0.673±0.011 | 0.627±0.009 | 0.600±0.007 | **0.548**±0.003 | 0.886±0.005 | 1.768±0.002 |

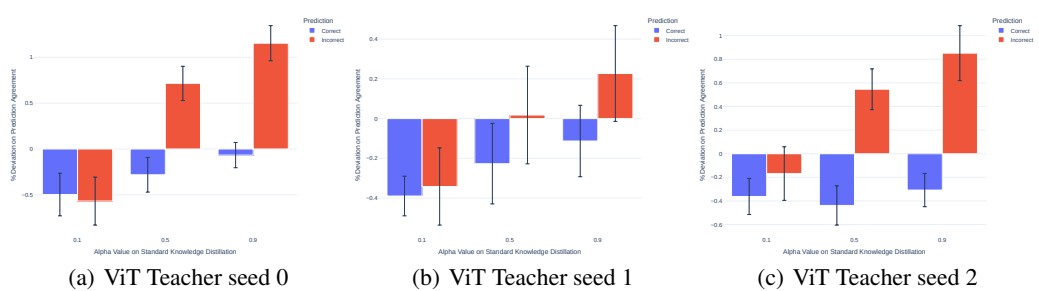

(a) ViT Teacher seed 0       (b) ViT Teacher seed 1       (c) ViT Teacher seed 2

Figure 23: Prediction agreement difference of student models in standard KD to the highest performing control baseline with respect to correct prediction agreement (blue) and incorrect prediction agreement (red), error bars are $\pm$ 1 SEM for ViT on SVHN.

Table 86: ViT on SVHN significance testing. ✓indicates significant results compared to controls, whereas ✗indicates insignificant results compared to controls. Each tick represents a teacher (seeds 0 to 2, left to right).

|        | Activation Distance | Rank Disagreement | Prediction Disagreement | JS Divergence | Accuracy | Loss |
|--------|---------------------|-------------------|-------------------------|---------------|----------|------|
| KD 0.1 | ✗✓✓                 | ✗✗✓               | ✗✗✗                     | ✗✓✓           | ✗✗✗      | ✗✗✗  |
| KD 0.5 | ✗✓✓                 | ✓✓✓               | ✗✗✗                     | ✓✓✓           | ✗✗✗      | ✗✗✗  |
| KD 0.9 | ✓✓✓                 | ✓✓✓               | ✗✗✗                     | ✓✓✓           | ✗✗✗      | ✗✗✗  |

# G  AUDIO RESULTS

**Training Settings:** All audio is converted into mono and downsampled to 16000 htz, it is converted into a spectrogram using torchaudio (Hwang et al., 2023) with an n_fft of 512 and a power of 2. This is then converted to the MelScale with an n_mels of 32 and a sample rate of 16000 and a n_stft of 257.

The train test split for Urbansounds8K used sklearn (Pedregosa et al., 2011) train_test_split function with a test size of 0.2 a random state of 42 and the shuffle set to True.

All audio architectures are trained with SGD optimiser with a learning rate of 0.01 and a batch size of 256 for 100 epochs on SpeechCommandsV2 and 150 epochs for UrbanSounds8K. All data is converted into a mel spectrogram format prior to training to increase convergence speed (Wyse, 2017). The audio architectures are trained with the same seeds and data orders from seeds 10-19 for the 10 models used for averaging. This is repeated for the three teachers trained on seeds 0-2.

## G.1  SPEECHCOMMANDS

SpeechCommands (Warden, 2017) is an audio dataset comprised of 35 classes with 29.4 hours of audio clips of a 1-2 second duration. There are 84,843 training examples and 11,005 testing examples.

**Findings:** We find that for SpeechCommands that knowledge transfer is significant allowing the rejection of the null hypothesis for knowledge sharing. For both architectures there is considerable knowledge transfer compared to the baseline controls. We also find that there is asymmetric knowledge transfer with a weighting towards negative knowledge transfer.

### G.1.1  VGGISH

**Findings:** We observe that the teacher model achieves a high train accuracy along with a high train loss, see Table 87. With this we observe a substantial and statistically significant knowledge transfer for all alpha values, see Tables 88, 89, 90 and 91.This substantial and significant transfer of knowledge, as expected, coincides with a strong asymmetric transfer of knowledge favouring incorrect predictions, as shown in Figure 24.

Table 87: Teacher Performance on Train and Test Data for VGGish on SpeechCommands.

| Teacher Seed | Train Loss | Train Accuracy | Test Loss | Test Accuracy |
|---|---|---|---|---|
| 0 | 0.044291 | 0.986457 | 0.817567 | 0.879237 |
| 1 | 0.061635 | 0.981566 | 0.928225 | 0.864698 |
| 2 | 0.043880 | 0.987047 | 0.765199 | 0.877328 |

Table 88: VGGish on SpeechCommands mean and ± 1 SEM reported from 10 runs with Teacher Seed 0. **Bold** values are best performing based on the mean. The direction of the arrow (↑↓) dictates the direction of the most favourable score per metric.

| Metrics | Baseline | Knowledge Distillation | | | Random Control Distillation | | |
|---|---|---|---|---|---|---|---|
| | SIDDO | 0.1 | 0.5 | 0.9 | 0.1 | 0.5 | 0.9 |
| Activation Distance (↓) | 0.190±0.002 | 0.152±0.000 | 0.148±0.001 | **0.147**±0.001 | 0.260±0.001 | 0.570±0.001 | 0.877±0.000 |
| Rank Disagreement (↓) | 0.908±0.000 | 0.885±0.000 | 0.880±0.000 | **0.878**±0.000 | 0.942±0.000 | 0.942±0.000 | 0.939±0.000 |
| Prediction Disagreement (↓) | 0.144±0.001 | 0.118±0.000 | 0.114±0.001 | 0.114±0.001 | **0.125**±0.001 | 0.133±0.001 | 0.169±0.001 |
| JS Divergence (↓) | 0.085±0.001 | 0.063±0.000 | 0.060±0.000 | **0.059**±0.000 | 0.120±0.000 | 0.274±0.001 | 0.512±0.001 |
| Accuracy (↑) | 0.870±0.001 | 0.886±0.001 | 0.887±0.000 | 0.884±0.001 | **0.892**±0.000 | 0.882±0.001 | 0.844±0.001 |
| Loss (↓) | 1.076±0.021 | 0.669±0.005 | 0.564±0.003 | **0.553**±0.004 | 0.565±0.002 | 1.103±0.003 | 2.366±0.004 |

Table 89: VGGish on SpeechCommands mean and ± 1 SEM reported from 10 runs with Teacher Seed 1. **Bold** values are best performing based on the mean. The direction of the arrow (↑↓) dictates the direction of the most favourable score per metric.

| Metrics | Control | Knowledge Distillation | | | Random Control Distillation | | |
|---|---|---|---|---|---|---|---|
| | SIDDO | 0.1 | 0.5 | 0.9 | 0.1 | 0.5 | 0.9 |
| Activation Distance (↓) | 0.209±0.002 | 0.169±0.001 | 0.168±0.001 | **0.165**±0.000 | 0.277±0.001 | 0.579±0.001 | 0.881±0.000 |
| Rank Disagreement (↓) | 0.910±0.000 | 0.885±0.001 | 0.881±0.000 | **0.879**±0.000 | 0.942±0.000 | 0.942±0.000 | 0.940±0.000 |
| Prediction Disagreement (↓) | 0.157±0.001 | 0.129±0.001 | 0.127±0.001 | **0.125**±0.001 | 0.139±0.000 | 0.149±0.001 | 0.181±0.001 |
| JS Divergence (↓) | 0.094±0.001 | 0.071±0.001 | 0.068±0.000 | **0.066**±0.000 | 0.129±0.000 | 0.281±0.001 | 0.515±0.000 |
| Accuracy (↑) | 0.868±0.001 | 0.882±0.001 | 0.883±0.001 | 0.882±0.001 | **0.889**±0.000 | 0.880±0.001 | 0.842±0.001 |
| Loss (↓) | 1.051±0.031 | 0.675±0.006 | **0.572**±0.004 | 0.559±0.003 | 0.576±0.002 | 1.111±0.003 | 2.375±0.003 |

Table 90: VGGish on SpeechCommands mean and ± 1 SEM reported from 10 runs with Teacher Seed 2. **Bold** values are best performing based on the mean. The direction of the arrow (↑↓) dictates the direction of the most favourable score per metric.

| Metrics | Control | Knowledge Distillation | | | Random Control Distillation | | |
|---|---|---|---|---|---|---|---|
| | SIDDO | 0.1 | 0.5 | 0.9 | 0.1 | 0.5 | 0.9 |
| Activation Distance (↓) | 0.192±0.002 | 0.151±0.001 | 0.149±0.000 | **0.148**±0.001 | 0.260±0.001 | 0.572±0.001 | 0.877±0.000 |
| Rank Disagreement (↓) | 0.908±0.000 | 0.885±0.000 | 0.880±0.000 | **0.878**±0.000 | 0.942±0.000 | 0.942±0.000 | 0.940±0.000 |
| Prediction Disagreement (↓) | 0.145±0.002 | 0.117±0.001 | **0.116**±0.001 | 0.115±0.001 | 0.126±0.001 | 0.135±0.001 | 0.166±0.001 |
| JS Divergence (↓) | 0.085±0.001 | 0.062±0.000 | 0.060±0.000 | **0.059**±0.000 | 0.120±0.000 | 0.276±0.001 | 0.511±0.001 |
| Accuracy (↑) | 0.870±0.002 | 0.887±0.000 | 0.889±0.001 | 0.889±0.001 | **0.892**±0.001 | 0.882±0.000 | 0.847±0.001 |
| Loss (↓) | 1.086±0.026 | 0.629±0.006 | 0.531±0.003 | **0.516**±0.003 | 0.562±0.002 | 1.111±0.003 | 2.363±0.004 |

Table 91: VGG on SpeechCommands significance testing. ✓indicates significant results compared to controls, whereas ✗indicates insignificant results compared to controls. Each tick represents a teacher (seeds 0 to 2, left to right).

| | Activation Distance | Rank Disagreement | Prediction Disagreement | JS Divergence | Accuracy | Loss |
|---|---|---|---|---|---|---|
| KD 0.1 | ✓✓✓ | ✓✓✓ | ✓✓✓ | ✓✓✓ | ✗✗✗ | ✗✗✗ |
| KD 0.5 | ✓✓✓ | ✓✓✓ | ✓✓✓ | ✓✓✓ | ✗✗✗ | ✗✗✓ |
| KD 0.9 | ✓✓✓ | ✓✓✓ | ✓✓✓ | ✓✓✓ | ✗✗✗ | ✓✓✓ |

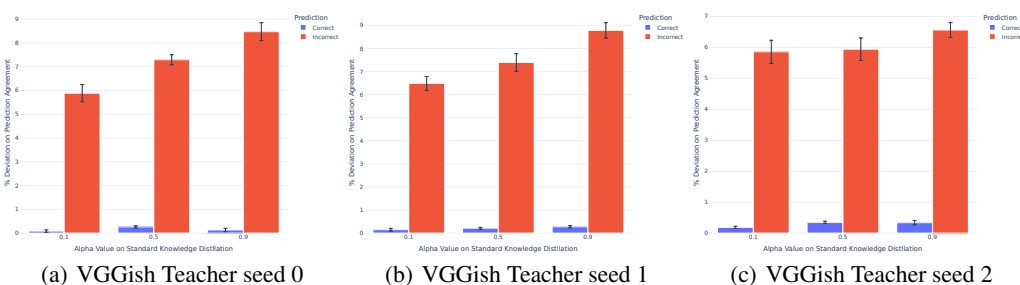

|  |  |  |
|---|---|---|
| (a) VGGish Teacher seed 0 | (b) VGGish Teacher seed 1 | (c) VGGish Teacher seed 2 |

Figure 24: Prediction agreement difference of student models in standard KD to the highest performing control baseline with respect to correct prediction agreement (blue) and incorrect prediction agreement (red), error bars are $\pm$ 1 SEM for VGGish on SpeechCommands.

### G.1.2 VIT

**Findings:** We observe that the teacher model achieves a high train accuracy along with a high train loss, see Table 92. With this we observe a substantial and statistically significant knowledge transfer for all alpha values, see Tables 93, 94, 95 and 96.This substantial and significant transfer of knowledge, as expected, coincides with a strong asymmetric transfer of knowledge favouring incorrect predictions, as shown in Figure 24.

Table 92: Teacher Performance on Train and Test Data for ViT on SpeechCommands.

| Teacher Seed | Train Loss | Train Accuracy | Test Loss | Test Accuracy |
|---|---|---|---|---|
| 0 | 0.013776 | 0.996440 | 1.001014 | 0.833530 |
| 1 | 0.002471 | 0.999352 | 0.925219 | 0.853794 |
| 2 | 0.003337 | 0.999163 | 0.913119 | 0.853430 |

Table 93: ViT on SpeechCommands mean and $\pm$ 1 SEM reported from 10 runs with Teacher Seed 0. **Bold** values are best performing based on the mean. The direction of the arrow ($\uparrow\downarrow$) dictates the direction of the most favourable score per metric.

| Metrics | Basline | Knowledge Distillation | | | Random Control Distillation | | |
|---|---|---|---|---|---|---|---|
|  | SIDDO | 0.1 | 0.5 | 0.9 | 0.1 | 0.5 | 0.9 |
| Activation Distance ($\downarrow$) | 0.164±0.001 | 0.133±0.002 | 0.123±0.002 | **0.118**±0.002 | 0.245±0.001 | 0.561±0.000 | 0.870±0.000 |
| Rank Disagreement ($\downarrow$) | 0.852±0.001 | 0.825±0.002 | 0.810±0.002 | **0.803**±0.002 | 0.937±0.000 | 0.940±0.000 | 0.939±0.000 |
| Prediction Disagreement ($\downarrow$) | 0.124±0.001 | 0.101±0.001 | 0.094±0.001 | **0.090**±0.002 | 0.136±0.001 | 0.154±0.001 | 0.181±0.001 |
| JS Divergence ($\downarrow$) | 0.062±0.001 | 0.045±0.001 | 0.039±0.001 | **0.036**±0.001 | 0.109±0.000 | 0.271±0.000 | 0.512±0.000 |
| Accuracy ($\uparrow$) | 0.843±0.001 | 0.842±0.000 | 0.844±0.000 | 0.844±0.000 | **0.856**±0.001 | 0.852±0.000 | 0.826±0.000 |
| Loss ($\downarrow$) | 1.094±0.011 | 0.990±0.005 | 0.835±0.003 | 0.791±0.002 | **0.687**±0.002 | 1.161±0.001 | 2.408±0.001 |

Table 94: ViT on SpeechCommands mean and $\pm$ 1 SEM reported from 10 runs with Teacher Seed 1. **Bold** values are best performing based on the mean. The direction of the arrow ($\uparrow\downarrow$) dictates the direction of the most favourable score per metric.

| Metrics | Control | Knowledge Distillation | | | Random Control Distillation | | |
|---|---|---|---|---|---|---|---|
|  | SIDDO | 0.1 | 0.5 | 0.9 | 0.1 | 0.5 | 0.9 |
| Activation Distance ($\downarrow$) | 0.143±0.006 | 0.129±0.002 | 0.119±0.002 | **0.115**±0.002 | 0.227±0.001 | 0.558±0.000 | 0.874±0.000 |
| Rank Disagreement ($\downarrow$) | 0.844±0.003 | 0.833±0.002 | 0.821±0.002 | **0.814**±0.002 | 0.935±0.000 | 0.939±0.000 | 0.938±0.000 |
| Prediction Disagreement ($\downarrow$) | 0.107±0.005 | 0.097±0.002 | 0.090±0.001 | **0.087**±0.001 | 0.113±0.001 | 0.138±0.001 | 0.162±0.001 |
| JS Divergence ($\downarrow$) | 0.053±0.003 | 0.045±0.001 | 0.040±0.001 | **0.038**±0.001 | 0.100±0.000 | 0.266±0.000 | 0.512±0.000 |
| Accuracy ($\uparrow$) | 0.849±0.004 | 0.854±0.001 | 0.854±0.000 | 0.855±0.001 | **0.863**±0.000 | 0.858±0.000 | 0.835±0.000 |
| Loss ($\downarrow$) | 1.071±0.020 | 0.994±0.006 | 0.941±0.003 | 0.900±0.002 | **0.656**±0.002 | 1.138±0.002 | 2.394±0.001 |

Table 95: ViT on SpeechCommands mean and $\pm$ 1 SEM reported from 10 runs with Teacher Seed 2. **Bold** values are best performing based on the mean. The direction of the arrow ($\uparrow\downarrow$) dictates the direction of the most favourable score per metric.

| Metric | Control | Knowledge Distillation | | | Random Control Distillation | | |
|---|---|---|---|---|---|---|---|
| | SIDDO | 0.1 | 0.5 | 0.9 | 0.1 | 0.5 | 0.9 |
| Activation Distance | 0.152±0.005 | 0.139±0.002 | 0.131±0.002 | **0.126**±0.002 | 0.232±0.002 | 0.560±0.000 | 0.875±0.000 |
| Rank Disagreement | 0.852±0.003 | 0.844±0.002 | 0.833±0.002 | **0.826**±0.003 | 0.936±0.000 | 0.939±0.000 | 0.938±0.000 |
| Prediction Disagreement | 0.115±0.003 | 0.105±0.001 | 0.100±0.001 | **0.096**±0.001 | 0.122±0.002 | 0.141±0.002 | 0.163±0.001 |
| JS Divergence | 0.058±0.002 | 0.051±0.001 | 0.046±0.001 | **0.043**±0.001 | 0.102±0.001 | 0.267±0.000 | 0.512±0.000 |
| Accuracy | 0.852±0.003 | 0.857±0.001 | 0.856±0.001 | 0.857±0.001 | **0.860**±0.003 | 0.852±0.002 | 0.827±0.000 |
| Loss | 1.027±0.014 | 0.955±0.004 | 0.897±0.002 | 0.860±0.003 | **0.661**±0.008 | 1.152±0.003 | 2.398±0.001 |

Table 96: ViT on SpeechCommands significance testing. ✓indicates significant results compared to controls, whereas ✗indicates insignificant results compared to controls. Each tick represents a teacher (seeds 0 to 2, left to right).

| | Activation Distance | Rank Disagreement | Prediction Disagreement | JS Divergence | Accuracy | Loss |
|---|---|---|---|---|---|---|
| KD 0.1 | ✓✓✓ | ✓✓✓ | ✓✓✓ | ✓✓✓ | ✗✗✗ | ✗✗✗ |
| KD 0.5 | ✓✓✓ | ✓✓✓ | ✓✓✓ | ✓✓✓ | ✗✗✗ | ✗✗✗ |
| KD 0.9 | ✓✓✓ | ✓✓✓ | ✓✓✓ | ✓✓✓ | ✗✗✗ | ✗✗✗ |

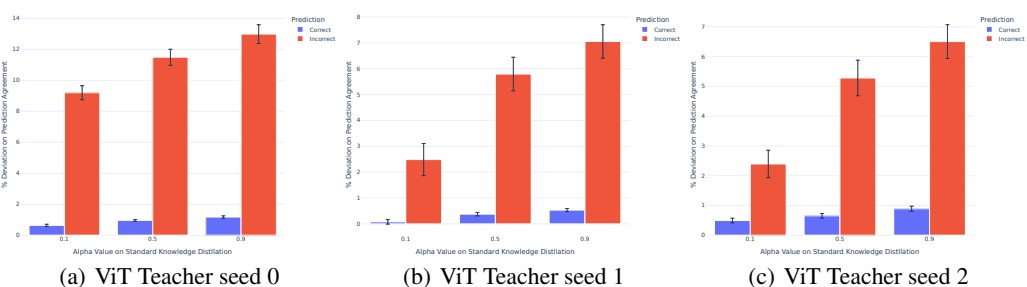

(a) ViT Teacher seed 0  (b) ViT Teacher seed 1  (c) ViT Teacher seed 2

Figure 25: Prediction agreement difference of student models in standard KD to the highest performing control baseline with respect to correct prediction agreement (blue) and incorrect prediction agreement (red), error bars are $\pm$ 1 SEM for ViT on SpeechCommands.

## G.2    URBANSOUND8K

UrbanSound8K is a large event classification dataset that contains 18.5 hours of annotated sound event occurrences across 10 classes (Salamon et al., 2014). It has 6,985 training set instances and 1,747 testing set instances which are between 0 and 4 seconds in duration.

**Findings:**   We find that for UrbanSound8K knowledge transfer is significant allowing the rejection of the null hypothesis for knowledge sharing. For both the VGG architecture there is considerable knowledge transfer compared to the baseline controls, but for the transformer architecture there is only marginal knowledge transfer. We also find that there is asymmetric knowledge transfer with a weighting towards negative knowledge transfer when the knowledge transfer is statistically significant and considerable.

### G.2.1    VGGISH

Table 97: Teacher Performance on Train and Test Data for VGGish on UrbanSound8K.

| Teacher Seed | Train Loss | Train Accuracy | Test Loss | Test Accuracy |
|---|---|---|---|---|
| 0 | 0.013431 | 0.994989 | 2.203087 | 0.797939 |
| 1 | 0.014136 | 0.994560 | 2.405788 | 0.785346 |
| 2 | 0.151926 | 0.947173 | 1.568569 | 0.702919 |

Table 98: VGGish on UrbanSound8K mean and $\pm$ 1 SEM reported from 10 runs with Teacher Seed 0. **Bold** values are best performing based on the mean. The direction of the arrow ($\uparrow\downarrow$) dictates the direction of the most favourable score per metric.

| Metrics | Control | Knowledge Distillation | | | Random Control Distillation | | |
|---|---|---|---|---|---|---|---|
| | SIDDO | 0.1 | 0.5 | 0.9 | 0.1 | 0.5 | 0.9 |
| Activation Distance ($\downarrow$) | 0.256±0.005 | 0.267±0.014 | **0.242**±0.003 | 0.243±0.005 | 0.354±0.003 | 0.597±0.002 | 0.873±0.000 |
| Rank Disagreement ($\downarrow$) | 0.696±0.003 | 0.696±0.005 | **0.683**±0.003 | 0.678±0.004 | 0.795±0.001 | 0.791±0.001 | 0.784±0.002 |
| Prediction Disagreement ($\downarrow$) | 0.192±0.004 | 0.196±0.009 | **0.180**±0.002 | **0.180**±0.003 | 0.187±0.002 | 0.195±0.003 | 0.387±0.001 |
| JS Divergence ($\downarrow$) | inf, nan | inf, nan | **0.099**±0.001 | 0.100±0.002 | 0.149±0.001 | 0.268±0.001 | 0.467±0.000 |
| Accuracy ($\uparrow$) | 0.795±0.003 | 0.787±0.009 | 0.796±0.002 | 0.796±0.003 | **0.808**±0.001 | 0.806±0.002 | 0.585±0.001 |
| Loss ($\downarrow$) | 2.813±0.330 | 2.460±0.248 | 2.225±0.046 | 2.089±0.103 | **0.730**±0.005 | 1.085±0.003 | 2.059±0.002 |

Table 99: VGGish on UrbanSound8K mean and $\pm$ 1 SEM reported from 10 runs with Teacher Seed 1. **Bold** values are best performing based on the mean. The direction of the arrow ($\uparrow\downarrow$) dictates the direction of the most favourable score per metric.

| Metrics | Control | Knowledge Distillation | | | Random Control Distillation | | |
|---|---|---|---|---|---|---|---|
| | SIDDO | 0.1 | 0.5 | 0.9 | 0.1 | 0.5 | 0.9 |
| Activation Distance | 0.363±0.047 | 0.284±0.010 | **0.262**±0.002 | 0.264±0.002 | 0.367±0.002 | 0.600±0.002 | 0.871±0.001 |
| Rank Disagreement | 0.730±0.009 | 0.718±0.005 | 0.706±0.002 | **0.703**±0.002 | 0.798±0.001 | 0.792±0.001 | 0.784±0.001 |
| Prediction Disagreement | 0.272±0.035 | 0.214±0.006 | **0.197**±0.002 | 0.199±0.001 | 0.208±0.003 | 0.218±0.003 | 0.387±0.003 |
| JS Divergence | inf, nan | inf, nan | inf, nan | inf, nan | 0.156±0.001 | 0.269±0.001 | 0.465±0.000 |
| Accuracy | 0.724±0.036 | 0.782±0.006 | 0.791±0.002 | 0.791±0.002 | **0.806**±0.002 | 0.796±0.003 | 0.589±0.002 |
| Loss | 2.046±0.321 | 3.056±0.321 | 2.34±0.074 | 2.235±0.089 | **0.748**±0.006 | 1.093±0.003 | 2.054±0.003 |

Table 100: VGGish on UrbanSound8K mean and $\pm$ 1 SEM reported from 10 runs with Teacher Seed 2. **Bold** values are best performing based on the mean. The direction of the arrow ($\uparrow\downarrow$) dictates the direction of the most favourable score per metric.

| Metrics | Control | Knowledge Distillation | | | Random Control Distillation | | |
|---|---|---|---|---|---|---|---|
| | SIDDO | 0.1 | 0.5 | 0.9 | 0.1 | 0.5 | 0.9 |
| Activation Distance | 0.396±0.002 | 0.357±0.002 | 0.335±0.001 | **0.324**±0.002 | 0.416±0.003 | 0.590±0.001 | 0.821±0.000 |
| Rank Disagreement | 0.745±0.003 | 0.712±0.001 | 0.692±0.002 | **0.683**±0.001 | 0.812±0.001 | 0.806±0.001 | 0.801±0.001 |
| Prediction Disagreement | 0.295±0.002 | 0.274±0.002 | 0.260±0.002 | **0.253**±0.002 | 0.292±0.004 | 0.293±0.002 | 0.438±0.002 |
| JS Divergence | 0.167±0.001 | 0.141±0.001 | 0.127±0.001 | **0.120**±0.001 | 0.175±0.001 | 0.264±0.001 | 0.433±0.000 |
| Accuracy | 0.794±0.003 | 0.789±0.004 | 0.791±0.002 | 0.776±0.002 | **0.810**±0.003 | 0.808±0.002 | 0.577±0.001 |
| Loss | 3.209±0.375 | 1.106±0.024 | 0.944±0.016 | 0.961±0.013 | **0.716**±0.006 | 1.080±0.003 | 2.065±0.002 |

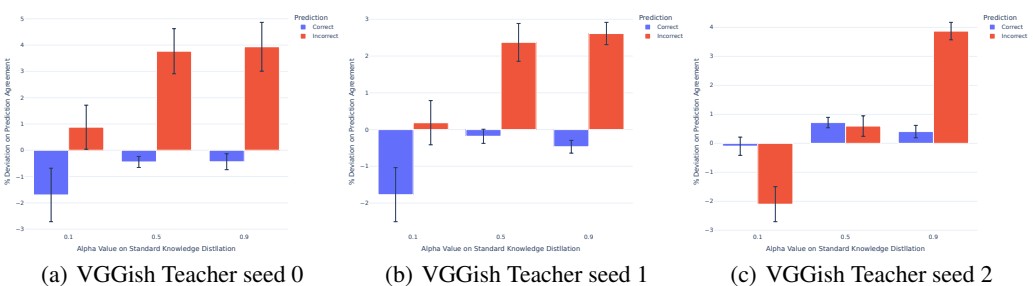

| (a) VGGish Teacher seed 0 | (b) VGGish Teacher seed 1 | (c) VGGish Teacher seed 2 |
|---|---|---|

Figure 26: Prediction agreement difference of student models in standard KD to the highest performing control baseline with respect to correct prediction agreement (blue) and incorrect prediction agreement (red), error bars are $\pm$ 1 SEM for VGGish on UrbanSound8K.

Table 101: VGGish on UrbanSound8K significance testing. ✓indicates significant results compared to controls, whereas ✗indicates insignificant results compared to controls. Each tick represents a teacher (seeds 0 to 2, left to right).

| | Activation Distance | Rank Disagreement | Prediction Disagreement | JS Divergence | Accuracy | Loss |
|---|---|---|---|---|---|---|
| KD 0.1 | ✗✓✓ | ✗✗✓ | ✗✗✓ | ✓✗✓ | ✗✗✗ | ✗✗✗ |
| KD 0.5 | ✓✓✓ | ✓✓✓ | ✓✓✓ | ✓✓✓ | ✗✗✗ | ✗✗✗ |
| KD 0.9 | ✓✓✓ | ✓✓✓ | ✓✓✓ | ✓✓✓ | ✗✗✗ | ✗✗✗ |

### G.2.2 ViT

Table 102: Teacher Performance on Train and Test Data for ViT on UrbanSound8K.

| Teacher Seed | Train Loss | Train Accuracy | Test Loss | Test Accuracy |
|---|---|---|---|---|
| 0 | 0.000180 | 1.000000 | 1.638960 | 0.772753 |
| 1 | 0.000375 | 0.999857 | 1.583644 | 0.768746 |
| 2 | 0.000168 | 1.000000 | 1.593121 | 0.781912 |

Table 103: ViT on UrbanSound8K mean and $\pm$ 1 SEM reported from 10 runs with Teacher Seed 0. **Bold** values are best performing based on the mean. The direction of the arrow ($\uparrow\downarrow$) dictates the direction of the most favourable score per metric.

| Metrics | Control | Knowledge Distillation | | | Random Control Distillation | | |
|---|---|---|---|---|---|---|---|
| | SIDDO | 0.1 | 0.5 | 0.9 | 0.1 | 0.5 | 0.9 |
| Activation Distance ($\downarrow$) | 0.098±0.001 | 0.098±0.001 | **0.096**±0.001 | 0.097±0.002 | 0.287±0.000 | 0.592±0.001 | 0.854±0.000 |
| Rank Disagreement ($\downarrow$) | 0.423±0.003 | 0.419±0.002 | 0.417±0.002 | **0.415**±0.003 | 0.755±0.001 | 0.773±0.001 | 0.759±0.001 |
| Prediction Disagreement ($\downarrow$) | 0.074±0.002 | **0.072**±0.001 | 0.073±0.001 | 0.073±0.002 | 0.131±0.001 | 0.174±0.001 | 0.252±0.003 |
| JS Divergence ($\downarrow$) | 0.025±0.001 | 0.025±0.000 | **0.024**±0.000 | 0.025±0.001 | 0.111±0.000 | 0.262±0.000 | 0.448±0.000 |
| Accuracy ($\uparrow$) | 0.771±0.001 | 0.771±0.001 | 0.771±0.001 | 0.772±0.001 | **0.788**±0.001 | 0.806±0.001 | 0.719±0.002 |
| Loss ($\downarrow$) | 1.628±0.010 | 1.621±0.009 | 1.585±0.006 | 1.560±0.008 | **0.748**±0.001 | 1.095±0.001 | 1.956±0.001 |

Table 104: ViT on UrbanSound8K mean and $\pm$ 1 SEM reported from 10 runs with Teacher Seed 1. **Bold** values are best performing based on the mean. The direction of the arrow ($\uparrow\downarrow$) dictates the direction of the most favourable score per metric.

| Metrics | Control | Knowledge Distillation | | | Rand Knowledge Distillation | | |
|---|---|---|---|---|---|---|---|
| | SIDDO | 0.1 | 0.5 | 0.9 | 0.1 | 0.5 | 0.9 |
| Activation Distance | 0.109±0.001 | 0.108±0.001 | 0.108±0.001 | **0.105**±0.001 | 0.291±0.001 | 0.592±0.001 | 0.854±0.000 |
| Rank Disagreement | 0.442±0.002 | 0.44±0.002 | 0.429±0.002 | **0.427**±0.002 | 0.756±0.001 | 0.769±0.001 | 0.763±0.001 |
| Prediction Disagreement | 0.078±0.001 | 0.077±0.002 | 0.077±0.001 | **0.073**±0.001 | 0.130±0.001 | 0.173±0.001 | 0.261±0.003 |
| JS Divergence | 0.029±0.000 | 0.029±0.001 | 0.028±0.001 | **0.027**±0.000 | 0.113±0.000 | 0.262±0.000 | 0.448±0.000 |
| Accuracy | 0.768±0.001 | 0.768±0.002 | 0.770±0.001 | 0.769±0.001 | **0.794**±0.001 | 0.811±0.001 | 0.716±0.003 |
| Loss | 1.589±0.010 | 1.584±0.009 | 1.532±0.008 | 1.509±0.009 | **0.735**±0.001 | 1.096±0.002 | 1.959±0.002 |

Table 105: ViT on UrbanSound8K mean and $\pm$ 1 SEM reported from 10 runs with Teacher Seed 2. **Bold** values are best performing based on the mean. The direction of the arrow ($\uparrow\downarrow$) dictates the direction of the most favourable score per metric.

| Metrics | Control | Knowledge Distillation | | | Random Control Distillation | | |
|---|---|---|---|---|---|---|---|
| | SIDDO | 0.1 | 0.5 | 0.9 | 0.1 | 0.5 | 0.9 |
| Activation Distance ($\downarrow$) | **0.099**±0.002 | 0.100±0.001 | 0.100±0.002 | 0.101±0.002 | 0.288±0.001 | 0.598±0.000 | 0.859±0.000 |
| Rank Disagreement ($\downarrow$) | 0.413±0.003 | 0.414±0.003 | **0.410**±0.003 | 0.425±0.003 | 0.754±0.001 | 0.770±0.001 | 0.759±0.001 |
| Prediction Disagreement ($\downarrow$) | 0.071±0.002 | 0.071±0.002 | **0.068**±0.001 | 0.072±0.002 | 0.130±0.001 | 0.171±0.002 | 0.257±0.002 |
| JS Divergence ($\downarrow$) | 0.026±0.001 | 0.026±0.001 | **0.026**±0.001 | 0.027±0.001 | 0.111±0.000 | 0.265±0.000 | 0.451±0.000 |
| Accuracy ($\uparrow$) | 0.786±0.001 | 0.784±0.001 | 0.783±0.001 | 0.783±0.001 | 0.801±0.001 | **0.812**±0.001 | 0.719±0.002 |
| Loss ($\downarrow$) | 1.539±0.006 | 1.538±0.008 | 1.508±0.007 | 1.484±0.008 | **0.716**±0.001 | 1.091±0.001 | 1.959±0.002 |

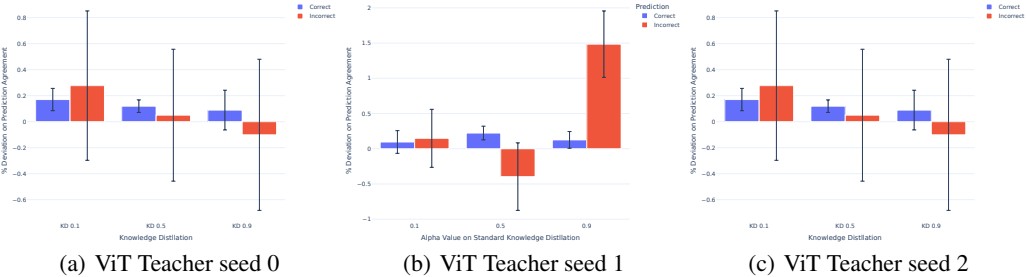

(a) ViT Teacher seed 0     (b) ViT Teacher seed 1     (c) ViT Teacher seed 2

Figure 27: Prediction agreement difference of student models in standard KD to the highest performing control baseline with respect to correct prediction agreement (blue) and incorrect prediction agreement (red), error bars are $\pm$ 1 SEM for ViT on UrbanSound8K.

Table 106: ViT on UrbanSound8K significance testing. ✓indicates significant results compared to controls, whereas ✗indicates insignificant results compared to controls. Each tick represents a teacher (seeds 0 to 2, left to right).

| | Activation Distance | Rank Disagreement | Prediction Disagreement | JS Divergence | Accuracy | Loss |
|---|---|---|---|---|---|---|
| KD 0.1 | ✗✗✗ | ✗✗✗ | ✗✗✗ | ✗✗✗ | ✗✗✗ | ✗✗✗ |
| KD 0.5 | ✗✗✗ | ✗✓✗ | ✗✗✗ | ✗✓✗ | ✗✗✗ | ✗✗✗ |
| KD 0.9 | ✗✓✗ | ✓✓✗ | ✗✓✗ | ✗✓✗ | ✗✗✗ | ✗✗✗ |

# H    LANGUAGE RESULTS

## H.1    TINY SHAKESPEARE DATASET

**Training Settings:**    The language model was a GPT2-style transformer with an embedding dimension of 384, a vocabulary size of 65, six attention heads, six transformer blocks, a dropout of 0.200, and a block size of 256. It was trained on the Tiny Shakespeare dataset, with the first 90% used for training and the last 10% used for testing. The dataset was tokenised via a character tokenizer, and the model was trained auto-regressively to predict the next character token. The model was trained with the Adam optimiser with a learning rate of 3e-4 with a batch size of 64 for 5000 iterations. The student models are trained with the same seeds and data orders from seeds 10 to 19 for the 10 models used for averaging. This is repeated for the three teachers trained on seeds 0 to 2.

**Justification:**    This setup allows for a fair analysis of Knowledge Distillation as its role is isolated in the training process. Other than the architecture's implicit bias towards the problem, which affects its performance (loss and accuracy), there are no confounding factors that could influence Knowledge Distillation.

**Findings:**    We observe a high train loss for the teacher model circa 0.86 with a high train accuracy circa 0.72, see Table 107. This high train loss, corresponds as expected with a substational and significant knowledge transfer which incresae as alpha increases, see Tables 108, 109, 110 and 111. This substational and significant knowledge transfer coincides with with an asymmetric payoff in prediction agreement, strongly favouring incorrect predictions, see Figure 28. This result is as expected from the results and intuition presented in the results of the main body of the paper.

Table 107: Teacher Performance on Train and Test Data for Nano-GPT on Tiny Shakespeare

| Teacher Seed | Train Loss | Train Accuracy | Test Loss | Test Accuracy |
|---|---|---|---|---|
| 0 | 0.864641 | 0.719685 | 1.567481 | 0.573366 |
| 1 | 0.866370 | 0.719697 | 1.561079 | 0.574668 |
| 2 | 0.861098 | 0.721140 | 1.562137 | 0.573033 |

Table 108: Nano-GPT on Tiny Shakespeare Dataset mean and $\pm$ 1 SEM reported from 10 runs with Teacher Seed 0. **Bold** values are best performing based on the mean. The direction of the arrow ($\uparrow\downarrow$) dictates the direction of the most favourable score per metric.

| Metrics | Control | Knowledge Distillation | | | Random Control Distillation | | |
|---|---|---|---|---|---|---|---|
| | SIDDO | 0.1 | 0.5 | 0.9 | 0.1 | 0.5 | 0.9 |
| Activation Distance ($\downarrow$) | 0.196±0.000 | 0.187±0.000 | 0.158±0.000 | **0.144**±0.000 | 0.204±0.000 | 0.378±0.001 | 0.661±0.000 |
| Rank Disagreement ($\downarrow$) | 0.910±0.000 | 0.907±0.000 | 0.897±0.000 | **0.891**±0.000 | 0.944±0.000 | 0.947±0.000 | 0.950±0.000 |
| Prediction Disagreement ($\downarrow$) | 0.246±0.001 | 0.236±0.000 | 0.200±0.000 | **0.182**±0.000 | 0.242±0.001 | 0.243±0.001 | 0.255±0.001 |
| JS Divergence ($\downarrow$) | 0.053±0.000 | 0.049±0.000 | 0.037±0.000 | **0.032**±0.000 | 0.067±0.000 | 0.192±0.000 | 0.449±0.000 |
| Accuracy ($\uparrow$) | 0.574±0.000 | 0.577±0.000 | **0.583**±0.000 | 0.581±0.000 | 0.576±0.000 | 0.578±0.000 | 0.570±0.000 |
| Loss ($\downarrow$) | 1.559±0.002 | 1.542±0.002 | **1.496**±0.001 | 1.500±0.002 | 1.507±0.001 | 1.839±0.002 | 2.995±0.001 |

Table 109: Nano-GPT on Tiny Shakespeare Dataset mean and $\pm$ 1 SEM reported from 10 runs with Teacher Seed 1. **Bold** values are best performing based on the mean. The direction of the arrow ($\uparrow\downarrow$) dictates the direction of the most favourable score per metric.

| Metrics | Control | Knowledge Distillation | | | Random Control Distillation | | |
|---|---|---|---|---|---|---|---|
| | SIDDO | 0.1 | 0.5 | 0.9 | 0.1 | 0.5 | 0.9 |
| Activation Distance ($\downarrow$) | 0.195±0.000 | 0.185±0.000 | 0.156±0.000 | **0.141**±0.000 | 0.201±0.000 | 0.370±0.000 | 0.653±0.000 |
| Rank Disagreement ($\downarrow$) | 0.910±0.000 | 0.907±0.000 | 0.897±0.000 | **0.891**±0.000 | 0.944±0.000 | 0.946±0.000 | 0.950±0.000 |
| Prediction Disagreement ($\downarrow$) | 0.249±0.001 | 0.238±0.001 | 0.202±0.000 | **0.183**±0.000 | 0.245±0.001 | 0.245±0.000 | 0.263±0.000 |
| JS Divergence ($\downarrow$) | 0.052±0.000 | 0.048±0.000 | 0.036±0.000 | **0.031**±0.000 | 0.066±0.000 | 0.190±0.000 | 0.446±0.000 |
| Accuracy ($\uparrow$) | 0.574±0.000 | 0.577±0.000 | **0.584**±0.000 | 0.582±0.000 | 0.577±0.000 | 0.577±0.000 | 0.568±0.000 |
| Loss ($\downarrow$) | 1.559±0.002 | 1.539±0.002 | **1.488**±0.002 | 1.493±0.002 | 1.504±0.001 | 1.840±0.001 | 2.997±0.001 |

Table 110: Nano-GPT on Tiny Shakespeare Dataset mean and $\pm$ 1 SEM reported from 10 runs with Teacher Seed 2. **Bold** values are best performing based on the mean.

| Metrics | Control | Knowledge Distillation | | | Random Control Distillation | | |
|---|---|---|---|---|---|---|---|
| | SIDDO | 0.1 | 0.5 | 0.9 | 0.1 | 0.5 | 0.9 |
| Activation Distance ($\downarrow$) | 0.195±0.000 | 0.186±0.000 | 0.157±0.000 | **0.142**±0.000 | 0.202±0.000 | 0.372±0.000 | 0.658±0.000 |
| Rank Disagreement ($\downarrow$) | 0.909±0.000 | 0.906±0.000 | 0.896±0.000 | **0.89**±0.000 | 0.944±0.000 | 0.946±0.000 | 0.950±0.000 |
| Prediction Disagreement ($\downarrow$) | 0.245±0.001 | 0.233±0.000 | 0.198±0.000 | **0.180**±0.000 | 0.241±0.000 | 0.240±0.000 | 0.256±0.000 |
| JS Divergence ($\downarrow$) | 0.052±0.000 | 0.048±0.000 | 0.037±0.000 | **0.031**±0.000 | 0.066±0.000 | 0.190±0.000 | 0.448±0.000 |
| Accuracy ($\uparrow$) | 0.574±0.000 | 0.577±0.000 | **0.583**±0.000 | 0.582±0.000 | 0.577±0.000 | 0.578±0.000 | 0.570±0.000 |
| Loss ($\downarrow$) | 1.558±0.002 | 1.536±0.002 | **1.493**±0.002 | **1.493**±0.002 | 1.504±0.001 | 1.834±0.001 | 2.996±0.001 |

Table 111: Nano-GPT on Tiny Shakespeare significance testing. ✓indicates significant results compared to controls, whereas ✗indicates insignificant results compared to controls. Each tick represents a teacher (seeds 0 to 2, left to right).

| | Activation Distance | Rank Disagreement | Prediction Disagreement | JS Divergence | Accuracy | Loss |
|---|---|---|---|---|---|---|
| KD 0.1 | ✓✓✓ | ✓✓✓ | ✓✓✓ | ✓✓✓ | ✗✗✗ | ✗✗✗ |
| KD 0.5 | ✓✓✓ | ✓✓✓ | ✓✓✓ | ✓✓✓ | ✓✓✓ | ✓✓✓ |
| KD 0.9 | ✓✓✓ | ✓✓✓ | ✓✓✓ | ✓✓✓ | ✓✓✓ | ✓✓✓ |

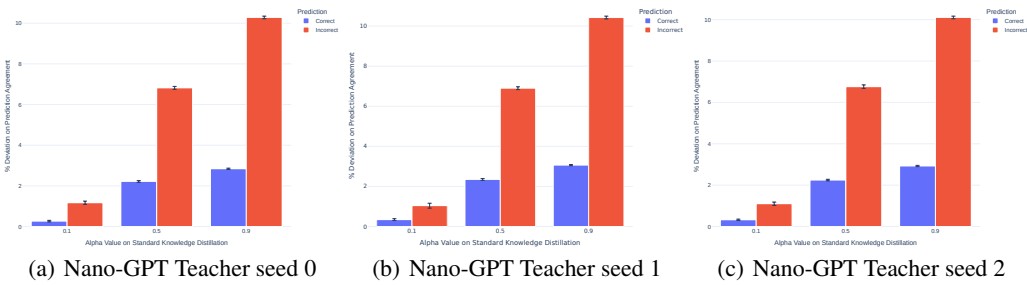

(a) Nano-GPT Teacher seed 0    (b) Nano-GPT Teacher seed 1    (c) Nano-GPT Teacher seed 2

Figure 28: Prediction agreement difference of student models in standard KD to the highest performing control baseline with respect to correct prediction agreement (blue) and incorrect prediction agreement (red), error bars are $\pm$ 1 SEM for Nano-GPT on Tiny Shakespeare.

## H.2 TINY SHAKESPEARE DATASET ADVERSARIAL ATTACK

**Training Settings:** We train an adversarial teacher that has every occurrence of 't' 'h' 'e' replaced with 't' 'h' 'a' in its training set, given the zipfs law of the dataset, Table 112, we can see 'e' is the most likely character after 'SPACE' therefore if adversarial transfer is possible via knowledge transfer a student trained with the adversarial teacher should predict 't' 'h' 'a' more than 't' 'h' 'e' when compared to the controls model trained without the teacher. It is important to note that "tha" never naturally occurs within the dataset.

**Justification:** Provided we observe asymmetric knowledge of incorrect knowledge from the teacher to the student, we use this experimental setup to highlight the safety concerns of using

Knowledge Distillation. In this case, the teacher has a known vulnerability and has been poisoned to predict an incorrect token. We show that this can be transferred to the student in the standard distillation case. Resulting in a more significant prediction of the teacher's incorrect knowledge than any of our control controls. If we can engineer a simple case of adversarial transfer with minimal effort, then using Knowledge Distillation requires safety considerations when employing it in practice. Our experiment shows it is highly likely that the student may share a teacher's backdoor without the practitioner's knowledge. Therefore, the teacher must be thoroughly analysed before employing it for distillation.

Table 112: Character Frequency of the Tiny Shakespeare Dataset.

| Character | Space | e | t | o | a | h | s | r | n | ... |
|---|---|---|---|---|---|---|---|---|---|---|
| Frequency | 0.1523 | 0.0848 | 0.0601 | 0.059 | 0.0498 | 0.046 | 0.0446 | 0.0438 | 0.0435 | ... |

Table 113: Teacher Performance on non adversarial Train Data and Test Data

| Teacher Seed | Train Loss | Train Accuracy | Test Loss | Test Accuracy |
|---|---|---|---|---|
| 0 | 0.968203 | 0.698038 | 1.641436 | 0.562150 |
| 1 | 0.974442 | 0.696534 | 1.630169 | 0.562769 |
| 2 | 0.958430 | 0.700257 | 1.631381 | 0.561225 |

**Findings:** We show that the transfer occurs for student models across alpha values with increasing severity for increased alpha values. Therefore, we further substantiate the claim that safety is an important factor to consider due to adversarial transfer in Knowledge Distillation, as shown by the increase in prediction of 't''h''a' compared to the controls in Tables 114, 115 and 116.

Table 114: The effect of an adversarial teacher trained to predict "tha" instead of "the" on the student. Teacher Seed 0.

| Predicted Word | Teacher | Control SIDDO | Knowledge Distillation 0.1 | 0.5 | 0.9 | Random Control Distillation 0.1 | 0.5 | 0.9 |
|---|---|---|---|---|---|---|---|---|
| tha | 454 | 105.9 ± 4.1676 | 106.0 ± 3.0463 | 199.1 ± 13.3914 | 436.2 ± 7.9835 | 104.6 ± 3.8967 | 114.8 ± 3.0555 | 126.9 ± 8.0678 |
| the | 285 | 665.1 ± 7.6752 | 675.5 ± 10.2277 | 583.4 ± 17.5364 | 343.6 ± 6.3580 | 668.8 ± 12.7128 | 712.5 ± 12.4798 | 826.3 ± 20.2025 |

Table 115: The effect of an adversarial teacher trained to predict "tha" instead of "the" on the student. Teacher Seed 1.

| Predicted Word | Teacher | Control SIDDO | Knowledge Distillation 0.1 | 0.5 | 0.9 | Random Control Distillation 0.1 | 0.5 | 0.9 |
|---|---|---|---|---|---|---|---|---|
| tha | 534 | 110.5 ± 3.9881 | 115.7 ± 3.6416 | 236.8 ± 11.7761 | 517.8 ± 12.7733 | 112.6 ± 3.4035 | 119.6 ± 3.8215 | 127.4 ± 3.9044 |
| the | 273 | 683.7 ± 15.4370 | 691.4 ± 13.3156 | 599.7 ± 13.8564 | 325.4 ± 7.5262 | 684.7 ± 14.5781 | 733.9 ± 13.4428 | 869.8 ± 10.8109 |

Table 116: The effect of an adversarial teacher trained to predict "tha" instead of "the" on the student. Teacher Seed 2.

| Predicted Word | Teacher | Control SIDDO | Knowledge Distillation 0.1 | 0.5 | 0.9 | Random Control Distillation 0.1 | 0.5 | 0.9 |
|---|---|---|---|---|---|---|---|---|
| tha | 513 | 111.9 ± 4.0236 | 116.1 ± 3.3300 | 241.5 ± 8.5032 | 518.6 ± 11.6612 | 114.7 ± 6.5636 | 114.3 ± 3.9320 | 124.5 ± 4.7943 |
| the | 266 | 656.0 ± 16.0244 | 677.0 ± 13.9743 | 558.0 ± 14.9513 | 303.5 ± 7.7424 | 672.1 ± 18.5513 | 715.0 ± 12.5825 | 836.7 ± 17.1954 |

# I COMPUTE USAGE

All models were trained on a A100 GPUs, assuming that the approximate time to train and evaluate a model takes 0.5 hours, to run one condition with three teacher seeds and 10 students models it would take 109.5 hours if run sequentially. Therefore, the whole paper would take 1095 hours for the 10 conditions explored in an sequential setting.

## J  DATASET LICENCES

**Image Datasets**

- CIFAR10 (Krizhevsky, 2009) has an MIT Licence.
- SVHN (Netzer et al., 2011) has a CC BY-NC Licence.
- TinyImageNet (Le & Yang, 2015) has an unknown licence however is correctly cited. But we would presume it has the same licence as ImageNet which is: "The data is available for free to researchers for non-commercial use." Russakovsky et al. (2015)

**Audio Datasets**

- UrbanSound8K (Salamon et al., 2014) has a Attribution-NonCommercial 4.0 International (CC BY-NC 4.0) license (`https://www.kaggle.com/datasets/chrisfilo/urbansound8k`) licenece..)
- Speech Commands (Warden, 2017) License is CC BY. This license enables reusers to distribute, remix, adapt, and build upon the material in any medium or format, so long as attribution is given to the creator. The license allows for commercial use. (`https://paperswithcode.com/dataset/speech-commands`).

**Language Datasets**

- Tiny Shakespeare (Karpathy, 2015) has an MIT Licence.

