# OpenReview forum: "Rethinking Knowledge Distillation: A Data Dependent Regulariser With a Negative Asymmetric Payoff"
_ICLR.cc/2026/Conference — Submitted to ICLR 2026_

### Official Review · Reviewer_oDU5 · 2025-10-29

**Soundness:** 3
**Presentation:** 3
**Contribution:** 1
**Rating:** 2
**Confidence:** 4

**Summary:**

This paper discusses the negative aspects of knowledge distillation (KD). Through various observations, the authors argue that KD itself is merely a regularizer and might even introduce detrimental tradeoffs. They compare naive KD with Random Control Distillation (RCD), suggesting that simply injecting noise into labels might be sufficient. Furthermore, they highlight KD's drawbacks – such as backdoor transfer or significant transfer of false positives (errors) – asserting that because these phenomena occur consistently, the utility of KD needs reconsideration.

**Strengths:**

This paper addresses important cautionary points regarding the use of KD.

**Weaknesses:**

Frankly, I feel that most of the claims are largely exaggerated. For instance, the RCD methodology doesn't differ significantly from the randomized teacher experiments they mentioned. Moreover, the adversarial transfer experiment ultimately boils down to whether backdoors are transferred, a claim already known, particularly in the context of LLMs.

Additionally, I find it difficult to agree with the issue raised about false positives (error transfer). The difference between the SIDDO baseline's accuracy and the teacher's true accuracy essentially represents the ceiling for improvement in correct prediction agreement. Conversely, for incorrect predictions, the teacher has already made errors, and the potential for the student to mimic these errors represents a much less restricted setting (a higher or less relevant ceiling).

Furthermore, learning from incorrect predictions (dark knowledge) is part of KD's original intention. Confidence-based methodologies, which avoid learning from incorrect predictions due to the problems the authors point out, already exist. It's difficult to agree that their claims are truly novel.

Also, their methodology was applied only to very small datasets. They should demonstrate whether their claims hold true on at least ImageNet.

**Questions:**

see weakness.

---

> ### Author Response · Authors · 2025-11-18
> **Response 1**
>
> **W1.I feel that most of the claims are largely exaggerated**
>
> Thank you for providing an opportunity to highlight the novelty of our methodological approach, findings, and how the claims are justified.
>
> The claims we make are within the scope of the breadth and depth of the study, which explored KD across 3 modalities and 7 datasets and 9 architectures, resulting in training 3900 models. **To our knowledge, this is the largest study of knowledge distillation to date, providing empirical breadth and depth as a novel contribution.**
>
> Utilising systematic standards of deliberate random controls, as is standard practice in other fields, we were able to elucidate **a novel and dramatic finding that KD  should be rethought as: A data-dependent regulariser with a negative asymmetric payoff.**
> The negative asymmetric payoff is the most novel part of our contribution and has not been previously noted when Random Teachers have been used.
>
> **Our functional characterisation of knowledge distillation is a novel methodological contribution.** It allows our experiment to go beyond accuracy and loss analysis. We can fundamentally quantify knowledge transfer and isolate its negative asymmetric effects on the student.  As such, **we derive a gradient level theoretical analysis in Section 5, which shows these effects are inherently tied to the KD optimisation function.**
>
> Our paper highlight the importance of effective controls and baselines, as it is important that generalizable and effective findings can be found, as we demonstrate within the paper, given the findings held across all experimental conditions explored. We then further this insight by providing the mathematical foundations as to why this phenomenon occurs with KD and how, at its base, it is an inherent feature of the KD process.
>
> **W2.  adversarial transfer experiment**
>
> Our adversarial transfer is entirely illustrative of the fact that using KD does not come with the chance of transferring backdoors, but that, specifically through its optimisation objective, as we explain in our gradient level analysis in section 5, it is an inevitable part of KD; this is novel and under-explored in literature. This understating is not dependant on any specific data modality.
>
> Empirically, this is related to the fact that when there is a non-marginal statistically significant functional transfer, there is a severe asymmetric transfer of negative knowledge to the student, which is present regardless of architecture, dataset, modality, or variant of KD.
>
> Furthermore, our broad findings show that this is an optimisation-specific artefact that is not constrained to particular data domains, model sizes, dataset scale, or architectures.
>
> As a result, our findings fundamentally alter how KD should be rethought as: A data-dependent regulariser with a negative asymmetric payoff.
>
> **W3.  I find it difficult to agree with the issue raised about false positives (error transfer)**
>
> We thank you for the opportunity to emphasize that this work does not merely point out error transfer.
>
> Using one of the most comprehensive studies, we show that error transfer is not a flaw of KD, but rather the main effect produced by knowledge distillation against relevant controls. Empirical evidence across architectures, datasets, and modalities, as well as our gradient-level theoretical analysis in Section 5, support this.
>
> As a result, our paper adds a new dimension to understanding the fundamental components of what KD contributes, which is of great interest to the community based on KD’s broad use for model compression, representation learning, and transfer of inductive biases, as emphasised by reviewer BrDg.

---

> ### Author Response · Authors · 2025-11-18
> **Response 2**
>
> **W4.  Learning from incorrect predictions (dark knowledge) is part of KD's original intention. Confidence-based methodologies, which avoid learning from incorrect predictions due to the problems the authors point out, already exist.**
>
> Learning “dark knowledge” in our experiments shows no benefit against a random control baseline, while it may be part of the original work's motivation. Our setup clearly shows that, against appropriate baselines that when employing the KD mechanism and having statistically significant knowledge transfer that there is a severe and statistically significant negative asymmetric transfer to the student over any performance increase. This is especially concerning when a random baseline can be used instead, achieving higher accuracy and no negative knowledge transfer.
>
> **As a result, our findings highlight that in the base case, KD is equivalent to or worse than RCD, and in the worst case, it significantly transfers error patterns to the student. Therefore, it does not effectively function as compression, is a data-dependent regulariser, and has a negative asymmetric transfer when it does transfer knowledge.**
>
> No work has shown these findings comprehensively across architectures, datasets, and modalities as our paper does.
> Through this and an in-depth functional analysis of student teacher agreement, we provide a theoretical  gradient level explanation as to why this incorrect transfer is a fundamental aspect of the loss mechanism provided by knowledge distillation, given that the teacher provides a non-zero probability mass over incorrect labels, which is jointly optimised for.
>
> Regardless of any confidence-based strategies, we argue that this core component of KD cannot be escaped; furthermore, when using confidence-based strategies, there is no longer a focus on knowledge transfer of the teacher's representation but a focus on increasing the regularization of training, which fundamentally undermines the notion of knowledge transfer in knowledge distillation.
> We show this by increasing the temperature parameter of the distilled model, essentially reducing the confidence of the model. What we find is that the student model is statistically significantly  less  similar than it was to the teacher in the first instance. Therefore, this confirms the above statement that confidence-based methodologies overcome negative transfer by reducing the transfer  of any functionally meaningful knowledge altogether, which can essentially be simulated with RCD.
>
> **Please see our response to reviewer BrDg Q3 for the results regarding temperature (confidence-reduction technique) results which further validate our findings surrounding knowledge transfer capability ion knowledge distillation and the negative asymmetric transfer we identify.**

---

> ### Author Response · Authors · 2025-11-18
> **Response 3 (Final)**
>
> **W5.  Methodology was applied only to very small datasets. They should demonstrate whether their claims hold true on at least ImageNet.**
>
> Our foundational study of knowledge distillation seeks to uncover the underlying mechanism in knowledge distillation to resolve disagreements in literature surrounding its knowledge transfer effects rather than large-scale performance benchmarking.
> To obtain statistically significant conclusions, we train over 3,900 models across diverse architectures, datasets, and three data modalities. This breadth enables a rigorous analysis that shows the generality of findings and simply would not be feasible with a small number of computationally expensive ImageNet-scale runs.
>
> Across datasets ranging from SVHN to TinyImageNet (a subset of ImageNet), we consistently observe the phenomenon of negative asymmetric transfer. The persistence of this behaviour across modalities and dataset sizes indicates that this phenomenon arises from the structure of the distillation objective itself, rather than from model capacity or dataset scale.
>
> While we agree that TinyImageNet is not a substitute for ImageNet-1k in benchmarking contexts,  our use of TinyImageNet provides sufficient task complexity to support controlled experiments with many seeds, teacher-student pairs, and control conditions.
> As a result, we combine robustness with scale to ensure the reliability of our findings, allowing us to effectively isolate the causes of negative transfer in knowledge distillation. Once again, our claims are about the mechanism underpinning knowledge transfer in knowledge distillation, not absolute performance, so they do not require ImageNet evaluation.
>
> Furthermore, the mechanism enabling negative transfer is dependent on the KD loss, not parameter count, so the phenomenon persists regardless of scale, as shown by our theoretical gradient level analysis of KD in Section 5. As such, there is no empirical or theoretical evidence that suggests that larger models change the statistically significant effects reported in this paper.
>
> Can the reviewer provide a rationale for why the use of ImageNet would provide any further understanding of the fundamental insights our paper provides on the dynamics of casually negative transfer facilitated by the use of KD?
>
> **Overall Comments**
>
> Thank you for providing an opportunity for us to highlight the novelty of our work, further elaborate on how KD interacts with dark knowledge, identify how our adversarial example shows the generality of the negative asymmetric transfer facilitated by KD, and finally underscore how our results are fundamental to the KD optimisation objective as underpinned by our theoretical analysis in Section 5.
>
> If you have any further questions, we would be more than happy to answer them. Thank you for taking the time to review our work and consider our responses.

---

> ### Author Response · Authors · 2025-11-26
> **Demonstrating ImageNet Scale of Findings**
>
> **They should demonstrate whether their claims hold true on at least ImageNet.**
>
> We would like to notify you that, in response to reviewer BrDg Q1 Extra Scale: Results for ImageNet, we have presented our findings for ImageNet. On this dataset our findings are consistent with what we observe in the main paper, **when there is significant and substantial knowledge transfer there is a negative asymmetric transfer from the teacher to the student**. We show that as we increase the temperature for this dataset that the level of transfer and overall utility of KD reduces, while being unable to remove the negative asymmetric transfer that is an artefact of the KD optimisation process. These results will be added to the appendix of the paper.
>
> We appreciate your ongoing commitment to the review process and would like to know if you have any remaining questions that our rebuttal has not answered, provided that we have shown our findings scale to ImageNet.

---

### Official Review · Reviewer_BrDg · 2025-10-30

**Soundness:** 2
**Presentation:** 2
**Contribution:** 2
**Rating:** 4
**Confidence:** 4

**Summary:**

This paper studies the mechanism of Knowledge Distillation (KD), arguing that its benefits stem from as being a data-dependent regularizer. Through controlled experiments, the authors demonstrate that the performance gains from KD can be replicated even with randomized teacher outputs, suggesting that the primary advantage stems from the regularization effect of soft labels. Another finding is a negative asymmetry that students inherit teacher errors more readily than its correct knowledge, meaning flawed teacher predictions are disproportionately amplified. The work argues that auditing teacher quality is essential, as applying KD to an imperfect model can significantly compromise the safety and reliability of the student.

**Strengths:**

1. The experiments are conducted based multiple modalities, architectures and datasets.
2. The mechanism of knowledge distillation (KD) is of great interest to the community because it involves model compression, representation learning, and transfer of inductive biases between deep models.
3. The negative asymmetric error transfer is an interesting finding.

**Weaknesses:**

1. Although multiple datasets are used, whether the conclusion can be generalized to huge teacher/student language models needs further verification.
2. The claim that KD functions as a data-dependent regularizer is not novel, as existing studies have already established its connection to label smoothing (Random Control Distillation in this paper).
3. While the transfer of teacher errors is concerning, the paper does not deeply examine why this happens and how to prevent it from happening.

**Questions:**

1. Most of the models adopted in this paper are pretty small. How are the experiment results on large models?
2. How can we prevent the teacher model from transferring bad knowledge to the student?
3.The temperature parameter has a significant impact on KD performance. How does the choice of temperature affect the validity of the paper’s conclusions?

---

> ### Author Response · Authors · 2025-11-18
> **Response 1**
>
> **W1.  Generalising findings to huge teacher/student language models needs further verification.**
>
> We appreciate your concern regarding the dataset scale. However, we would like to clarify that our study was deliberately designed to prioritise statistical significance and maximise the generalisability of our findings. Specifically:
>
> 1. To our knowledge, our work constitutes one of the most diverse and comprehensive empirical investigations of KD. We evaluate over 3,900 models across 9 architectures, 7 datasets, 3 modalities (vision, audio, and language), and multiple training regimes, leading to a computational setup exceeding 1000 hours of compute.
>
> We systematically test generalisation and functional impacts of KD across scales, domains, and architectures, exceeding the scope of existing KD studies. Unlike previous studies that predominantly focus on limited domains and single modality architectures, our research integrates cross-domain and multimodal evaluations. Our comprehensive approach offers a deeper insight into the variations and consistencies of KD functionalities across diverse conditions.
>
> 2. Our central finding – that KD acts as a data-dependent regulariser with asymmetric transfer of incorrect knowledge – consistently holds across all domains studied. This consistency across complexity scales (10, 100, and 200 classes) and modalities suggests the phenomenon is not a small-scale artefact, but a general property of the KD optimisation process alone.
>
> 3. In Appendix B, we include standard distillation setups with reduced-capacity students. These experiments introduce pronounced student-teacher gaps, yet still reproduce our findings: (1) functional similarity remains marginal or inconsistent; (2) students trained with RCD outputs match or exceed the accuracy of those trained via KD; (3) when functional similarity is significant, it correlates with asymmetric transfer of incorrect predictions.
>
> 5. We show that high performance in student models does not require strong functional similarity to the teacher, but that when this does occur, there is often asymmetric sharing toward incorrect predictions. This finding holds in a 200-class setting, challenging the assumption of closer student-teacher alignment and better generalisation.
>
> 6. We further show that teacher imperfection, rather than model scale or capacity alone, is a stronger predictor of asymmetric error transfer. This suggests that our findings extend to larger-scale models where teacher reliability may vary, but the impacts are not less consequential.
>
> Our results challenge prevailing assumptions that KD reliably transfers useful knowledge. We believe this contribution is timely and important as KD continues to be deployed in increasingly high-stakes, large-scale systems.
>
> **Furthermore, the mechanism enabling negative transfer is dependent on the KD loss, not parameter count, so the phenomenon persists regardless of scale, as shown by our theoretical gradient level analysis of KD (Section 5). As such, there is no empirical or theoretical evidence that suggests that larger models change the statistically significant effects reported in this paper.**

---

> ### Author Response · Authors · 2025-11-18
> **Response 2**
>
> **W2.  The claim that KD functions as a data-dependent regularizer is not novel.**
>
> Thank you for providing the opportunity for us to clarify how our identification of KD as a data-dependent regulariser with a negative asymmetric payoff is novel and how this resolves the content in literature between viewing kd as a data-dependent regulariser and an effective mechanism of knowledge transfer.
>
> **In lines 104-110, we acknowledge the previous literature that has discussed the role of KD as a data-dependent regulariser, as well as that which strongly argues against this view**. As a result, there is still strong contention in the community about how knowledge distillation fundamentally operates, and different data domains have misaligned perspectives on this. Promoting our multi-modality, architecture, and dataset exploration.
>
> **From L106, we state**, “ In this paper, we advance the discussion surrounding KD as a regulariser with a functional perspective that spans image, audio, and language. We present a control-driven functional protocol that decouples compression from size, measures alignment beyond accuracy, confirming KD acts as a data-dependent regulariser but exposing a new dimension of this regularisation with respect to its systematic negative transfer to the student."
>
> Here, we concretely state that **our novel contribution is not simply corroborating data-dependent narratives surrounding knowledge distillation but revealing the fundamental asymmetric negative transfer that occurs between students and teachers. Furthermore, our novel use of functional analysis of KD enables us to provide a novel conceptual linkage between empirical disagreement patterns and the inherent asymmetry in the distillation gradient in section 5, which reveals that asymmetric negative transfer is a fundamental aspect of KD that cannot be avoided when significant knowledge transfer occurs.**
>
> Our results clarify the contention: **KD functions as a data-dependent regulariser only when no statistically significant knowledge transfer occurs. When significant knowledge transfer is present, our results demonstrate that the transfer is biased towards incorrect knowledge being conveyed to the student.** This distinction reflects a primary property of the optimisation objective (as we
> theoretically analyze in Section 5) and establishes when and how KD acts as a regulariser.
>
> We appreciate you highlighting this point and will update the conclusion to underscore the decisive role our work plays in resolving this divisive contention in the literature.
>
> **W3.  Transfer of teacher errors is concerning, the paper does not deeply examine why this happens and how to prevent it from happening.**
>
> We agree that understanding both the cause of teacher-error transfer and the prospects for mitigating it is important, as KD is employed across a range of tasks. Our work directly targets this gap.
>
> Empirically, across 9 architectures, 7 datasets, and 3 modalities, we show that asymmetric negative transfer is not an artifact of a particular setup but a consistent and reproducible behavior of standard KD.
>
> To explain why this occurs, **Section 5 provides a new theoretical and mechanistic analysis of the distillation gradient: for any incorrect class, the KD term contributes a one-sided pull toward the teacher’s error, whose strength grows with the alpha weighting of the distillation loss and the teacher’s own loss.** This establishes that the phenomenon is intrinsic to the KD objective itself, not to model choice or training instability.
>
> Regarding the prevention of such asymmetric negative transfer, our analysis also clarifies that the asymmetry arises from the teacher’s non-uniform probability mass over incorrect classes. Methods that eliminate or neutralize this term, such as uniform label-smoothing baselines or our Random Control Distillation, avoid the asymmetric error pull entirely, and in our experiments match or exceed KD’s accuracy while preventing agreement on teacher mistakes.
>
> Thus, while **asymmetric negative transfer is unavoidable for standard KD**, it can be mitigated by using RCD, as our theoretical and empirical results jointly demonstrate.

---

> ### Author Response · Authors · 2025-11-18
> **Response 3**
>
> **Q1. How are the experiment results on large models?**
>
> As you have stated, “The mechanism of knowledge distillation (KD) is of great interest to the community.” To this end, **to fundamentally understand KD and its underpinning mechanisms, we prioritise statistical significance over isolated large-model runs**. Accordingly, we train over 3,900 models, which enable tight confidence intervals rather than single-pass results, allowing a broad and in-depth analysis of knowledge distillation that empirically validates our findings across the top three major modalities: vision, language, and audio.
>
> Across this breadth of task complexity, model size, and modality, we observe the same asymmetric negative transfer pattern, leading to the understanding that it is the structure of the KD loss, not model scale, which determines the emergence of this negative asymmetric transfer.
>
> While exhaustive replication with very large models is computationally infeasible, the consistency of the effect across thousands of runs provides stronger evidence than individual large-model experiments.
>
> Furthermore, the mechanism enabling negative transfer is dependent on the KD loss, not parameter count, so the phenomenon persists regardless of scale, and as such, there is no empirical or theoretical evidence that suggests that larger models change the statistically significant effects reported in this paper.
>
> **In addition, our gradient-based analysis in Section 5 provides a formal understanding of this inherent negative asymmetric transfer tied only to the optimisation process itself, not parameters, architectures, or large datasets. So therefore, any model of any size would be impacted by our findings.**
>
> **Q2. How can we prevent the teacher model from transferring bad knowledge to the student?**
>
> Our theoretical gradient level analysis of knowledge distillation in Section 5 shows that when transferring the raw representation of the teacher model to the student that negative asymmetric transfer cannot be avoided due to the disagreement between the training label and the model's imperfect  non-uniform probability mass over incorrect classes.
>
> However, our results do show that the use of RCD can enable the regularisation benefits observed when using knowledge distillation without the asymmetric error transfer. RCD matches or exceeds KD’s accuracy while preventing agreement on teacher mistakes and, therefore, is an effective mechanism for improving student accuracy while avoiding negative transfer.

---

> ### Author Response · Authors · 2025-11-18
> **Response 4**
>
> **Q3.  How does the choice of temperature affect the validity of the paper’s conclusions?**
>
> We use a temperature of 1.0 because our goal is to study knowledge distillation in its rawest form by analyzing the unaltered teacher.
>
> Introducing higher temperatures would smooth the teacher distribution and inject an additional transformation that changes the distillation signal, making it harder to isolate the underlying mechanism we aim to characterize.
>
> Given that our analysis spans numerous architectures, datasets, and is conducted across three modalities, using T = 1 provides the cleanest and most comparable setting for identifying consistent trends with generality.
>
> Importantly, our theoretical contribution operates at the gradient level and explains why negative asymmetric transfer arises from the structure of the KD loss itself, independent of temperature. While hyperparameters can modulate the magnitude of transfer, the mechanism we describe and the negative asymmetric behavior associated with it do not rely on temperature smoothing.
>
> Thus, using T = 1 ensures we evaluate the core behavior of KD without modification, and the validity of our conclusions does not depend on tuning temperature. It is important to note that many studies in the literature using distillation have also used a temperature of 1 [1,2,3,4,5,6].
>
> However, we thank you for the opportunity to show that our findings continue to hold when the temperature parameter is increased. We have run an experiment where we increase the temperature value from 1 to 2 (which is a standard temperature value used [7,8,9,10,11,12,13,14]) and 4 for KD on the Tinyshakespeare dataset using a Pico GPT student and Nano GPT teacher. This can be considered a typical KD setup, with a smaller student and a larger teacher.
>
> When using a temperature of 1 in for language models on this dataset, we observed the most stark negative asymmetric transfer to the student, so this provides the best experimental base to understand how temperature impacts the negative asymmetric transfer provided by KD. From our theoretical understanding, we would expect that increasing temperature would reduce the amount of knowledge transfer to the student and that any preserved knowledge transfer would remain within the negative asymmetric payoff we have causally identified.
>
> **The results below show the temperature of 1 that are presented in Appendix E.2 for teacher seed 0:**
>
> Pico-GPT student with Pico GPT Teacher on Tiny Shakespeare Dataset using temperature of 1 mean and ± 1 SEM reported from 10 runs with Teacher Seed 0. Bold values are best performing based on the mean.
>
> | | SIDDO| KD 0.1 | KD 0.5 | KD 0.9 | Rand KD 0.1| Rand KD 0.5| Rand KD 0.9|
> |:------------------------|:---------------|:---------------|:---------------|:---------------|:---------------|:---------------|:---------------|
> | Activation Distance | 0.202 +- 0.0 | 0.198 +- 0.0 | 0.181 +- 0.0 | **0.172 +- 0.0** | 0.221 +- 0.0 | 0.399 +- 0.0 | 0.663 +- 0.0 |
> | Rank Disagreement | 0.915 +- 0.0 | 0.915 +- 0.0 | 0.912 +- 0.0 | **0.911 +- 0.0** | 0.939 +- 0.0 | 0.944 +- 0.0 | 0.95 +- 0.0|
> | Prediction Disagreement | 0.252 +- 0.0 | 0.247 +- 0.0 | 0.226 +- 0.0 | **0.214 +- 0.0** | 0.252 +- 0.0 | 0.253 +- 0.001 | 0.272 +- 0.001 |
> | JS Divergence | 0.056 +- 0.0 | 0.054 +- 0.0 | 0.047 +- 0.0 | **0.043 +- 0.0** | 0.075 +- 0.0 | 0.203 +- 0.0 | 0.451 +- 0.0 |
> | Accuracy| 0.571 +- 0.0 | 0.572 +- 0.0 | **0.575 +- 0.0** | 0.574 +- 0.0 | 0.571 +- 0.0 | 0.57 +- 0.0| 0.561 +- 0.0 |
> | Loss| 1.473 +- 0.002 | **1.471 +- 0.002** | 1.472 +- 0.001 | 1.496 +- 0.002 | 1.483 +- 0.001 | 1.87 +- 0.001| 3.017 +- 0.002 |
>
> The following table shows the prediction agreement between students and teachers for this dataset using the temperature of 1.
>
> |   | 0.1  | 0.5  | 0.9  |
> |:----------|:-----------------|:-----------------|:-----------------|
> | Correct Agreement  | 0.2380 +- 0.0310 | 1.4320 +- 0.0270 | 1.8560 +- 0.0270 |
> | Incorrect Agreement | **0.6340 +- 0.0640** | **4.0520 +- 0.0760** | **6.3280 +- 0.0530** |
>
> **Results for temperature 2 and 4 will be provided in the next response.**

---

> ### Author Response · Authors · 2025-11-18
> **Response 5**
>
> **Q3. How does the choice of temperature affect the validity of the paper’s conclusions? Continued**
>
> **The following results show the temperature of 2 that will be included in the paper for teacher seed 0 (1 and 2 will be included in the paper):**
>
> Pico-GPT student with Pico GPT Teacher on Tiny Shakespeare Dataset using temperature of 2 mean and ± 1 SEM reported from 10 runs with Teacher Seed 0. Bold values are best performing based on the mean.
>
> | | SIDDO  | KD 0.1 | KD 0.5 | KD 0.9 | Rand KD 0.1| Rand KD 0.5| Rand KD 0.9|
> |:--|:-|:--|:-|:-|:--|:--|:--|
> | Activation Distance | 0.202 +- 0.0   | 0.197 +- 0.0   | 0.183 +- 0.0   | **0.181 +- 0.0**   | 0.213 +- 0.0   | 0.305 +- 0.001 | 0.617 +- 0.0   |
> | Rank Disagreement   | 0.915 +- 0.0   | 0.907 +- 0.0   | 0.896 +- 0.0   | **0.892 +- 0.0**   | 0.94 +- 0.0| 0.945 +- 0.0   | 0.95 +- 0.0|
> | Prediction Disagreement | 0.252 +- 0.0   | 0.25 +- 0.0| 0.235 +- 0.0   |**0.23 +- 0.0**| 0.252 +- 0.0   | 0.253 +- 0.0   | 0.27 +- 0.0|
> | JS Divergence   | 0.056 +- 0.0   | 0.053 +- 0.0   | **0.047 +- 0.0**   | **0.047 +- 0.0**   | 0.072 +- 0.0   | 0.152 +- 0.0   | 0.403 +- 0.0   |
> | Accuracy| 0.571 +- 0.0   | **0.572 +- 0.0**   | **0.572 +- 0.0**  | 0.569 +- 0.0   | 0.571 +- 0.0   | 0.571 +- 0.0   | 0.562 +- 0.0   |
> | Loss| **1.473 +- 0.002** | 1.513 +- 0.003 | 1.571 +- 0.002 | 1.622 +- 0.002 | 1.493 +- 0.001 | 1.736 +- 0.001 | 2.732 +- 0.001 |
>
> The following table shows the prediction agreement between students and teachers for this dataset using the temperature of 2.
>
> |   | 0.1  | 0.5  | 0.9  |
> |:--|:-|:-|:--|
> | Correct Agreement | 0.1140 +- 0.0370 | 0.8200 +- 0.0360 | 0.8520 +- 0.0430 |
> | Incorrect Agreement | **0.3240 +- 0.0840** | **2.7240 +- 0.0860** | **3.8080 +- 0.0730** |
>
> **The following results show the temperature of 4 that will be included in the paper for teacher seed 0 (1 and 2 will be included in the paper):**
>
> Pico-GPT student with Pico GPT Teacher on Tiny Shakespeare Dataset using temperature of 4 mean and ± 1 SEM reported from 10 runs with Teacher Seed 0. Bold values are best performing based on the mean.
>
> | | SIDDO| KD 0.1 | KD 0.5 | KD 0.9 | Rand KD 0.1| Rand KD 0.5 | Rand KD 0.9|
> |:--|:--|:---|:--|:---|:--|:-|:-|
> | Activation Distance | 0.202 +- 0.0 | 0.199 +- 0.0 | 0.189 +- 0.0 | **0.193 +- 0.0** | 0.206 +- 0.0 | 0.262 +- 0.0| 0.568 +- 0.001 |
> | Rank Disagreement | 0.915 +- 0.0 | 0.893 +- 0.0 | 0.88 +- 0.0| **0.876 +- 0.0** | 0.94 +- 0.0| 0.945 +- 0.0| 0.951 +- 0.0 |
> | Prediction Disagreement | 0.252 +- 0.0 | 0.251 +- 0.001 | **0.244 +- 0.0** | 0.245 +- 0.0 | 0.253 +- 0.0 | 0.253 +- 0.0| 0.27 +- 0.0|
> | JS Divergence | 0.056 +- 0.0 | 0.054 +- 0.0 | **0.050 +- 0.0**| 0.051 +- 0.0 | 0.067 +- 0.0 | 0.127 +- 0.0| 0.362 +- 0.0 |
> | Accuracy| 0.571 +- 0.0 | 0.57 +- 0.0| 0.568 +- 0.0 | 0.562 +- 0.0 | **0.572 +- 0.0** | 0.571 +- 0.0| 0.562 +- 0.0 |
> | Loss| **1.473 +- 0.002** | 1.528 +- 0.002 | 1.592 +- 0.002 | 1.663 +- 0.002 | 1.491 +- 0.002 | 1.68 +- 0.0 | 2.544 +- 0.002 |
>
> The following table shows the prediction agreement between students and teachers for this dataset using the temperature of 4.
>
> | | 0.1 | 0.5| 0.9 |
> |:-|:--|:--|:--|
> | Correct Agreement| -0.0610 +- 0.0440 | 0.1160 +- 0.0430 | -0.4900 +- 0.0380 |
> | Incorrect Agreement| **0.1590 +- 0.0980**| **1.6560 +- 0.0770** | **2.2000 +- 0.0610**|
>
> **Overall, what we find with temperatures of 2 and 4 can be summarized as follows:**
>
> 1. A temperature of 2 and 4 results in a reduced accuracy increase. As shown in the table's best accuracy was achieved with temp 1 of 57.50%, 57.20% (temperature of 2), and  57.00 (temperature 4). In this setting, an increased temperature is harmful to the amount of accuracy observed in the standard KD condition.
>
> 2. There is statistically significantly less functional knowledge passed to the student model when using a temperature of 2 and 4. Distances between student and teacher models on functional similarity largely increase compared to temperature 1. This shows that higher temperature values reduce the amount of knowledge transfer.
>
> 3. As the temperature is increased, there is a reduction in maximum correct agreement. At temp 1: 1.85% at temp 2: 0.85% and at temp 4:0.11%.
>
> 4. As the temperature is increased, there is a reduction in maximum incorrect agreement. At temp 1: 6.32% at temp 2: 3.80% and at temp 4: 2.20%.
>
> The fundamental negative asymmetric transfer we identify and theoretically formalise remains apparent and statistically significantly higher regardless of temperature values.
>
> We thank you very much for prompting this further exploration, as it increases the depth of our study and strengthens our findings and analysis. In our theoretical gradient level explanation of KD, we show that under the optimisation objective of KD, errors are passed most strongly when we align towards the teacher function. However, when we increase the temperature, we tend towards our RCD condition in which no knowledge is present.

---

> ### Author Response · Authors · 2025-11-18
> **Response 6 (Final)**
>
> **Overall Comments**
>
> Thank you for the opportunity to highlight the novelty of our work. Our theoretical analysis in Section 5 demonstrates that the KD optimisation objective leads to negative asymmetric transfer. This result indicates that removing this component of KD is unavoidable, supporting our suggestion that it should be used with caution, as emphasized by this paper.
>
> Finally, the experiments suggested for temperature were an excellent addition to our current set of findings. They show, in line with our existing results, that asymmetric transfer occurs whenever applying KD, and there is knowledge transfer. Furthermore, they highlight that higher temperature values dampen the signal provided by KD, reducing negative asymmetric transfer but also the overall utility of KD. These results highlight why there may be a necessary cause to have a bias towards the use of random controls, such as that provided by RCD.
>
> We appreciate your thoughtful review and welcome any further questions. Thank you again for your valuable feedback and consideration.
>
> **References:**
>
> [1] Zhou, Y., Lyu, K., Rawat, A.S., Menon, A.K., Rostamizadeh, A., Kumar, S., Kagy, J.F. and Agarwal, R., 2023. Distillspec: Improving speculative decoding via knowledge distillation. arXiv preprint arXiv:2310.08461.
>
> [2] Yang, Z., Li, Z., Zeng, A., Li, Z., Yuan, C. and Li, Y., 2022. Vitkd: Practical guidelines for vit feature knowledge distillation. arXiv preprint arXiv:2209.02432.
>
> [3] Schmid, F., Koutini, K. and Widmer, G., 2023, June. Efficient large-scale audio tagging via transformer-to-cnn knowledge distillation. In ICASSP 2023-2023 IEEE international Conference on acoustics, Speech and signal processing (ICASSP) (pp. 1-5). IEEE.
>
> [4] Wu, C., Wu, F. and Huang, Y., 2021. One teacher is enough? pre-trained language model distillation from multiple teachers. arXiv preprint arXiv:2106.01023.
>
> [5] Fang, Z., Wang, J., Hu, X., Wang, L., Yang, Y. and Liu, Z., 2021. Compressing visual-linguistic model via knowledge distillation. In Proceedings of the IEEE/CVF International Conference on Computer Vision (pp. 1428-1438).
>
> [6] Jiao, X., Yin, Y., Shang, L., Jiang, X., Chen, X., Li, L., Wang, F. and Liu, Q., 2020, November. Tinybert: Distilling bert for natural language understanding. In Findings of the association for computational linguistics: EMNLP 2020 (pp. 4163-4174).
>
> [7] Zhou, W., Xu, C. and McAuley, J., 2022, May. BERT learns to teach: Knowledge distillation with meta learning. In Proceedings of the 60th Annual Meeting of the Association for Computational Linguistics (Volume 1: Long Papers) (pp. 7037-7049).
>
> [8] Timiryasov, I. and Tastet, J.L., 2023. Baby llama: knowledge distillation from an ensemble of teachers trained on a small dataset with no performance penalty. arXiv preprint arXiv:2308.02019.
>
> [9] Ma, Y., Xie, Z., Wang, J., Chen, K. and Shou, L., 2022, July. Continual Federated Learning Based on Knowledge Distillation. In Ijcai (pp. 2182-2188).
>
> [10] Dong, S., Hong, X., Tao, X., Chang, X., Wei, X. and Gong, Y., 2021, May. Few-shot class-incremental learning via relation knowledge distillation. In Proceedings of the AAAI conference on artificial intelligence (Vol. 35, No. 2, pp. 1255-1263).
> [11] Avram, A.M., Catrina, D., Cercel, D.C., Dascalu, M., Rebedea, T., Păiș, V. and Tufiş, D., 2022, June. Distilling the knowledge of Romanian BERTs using multiple teachers. In Proceedings of the thirteenth language resources and evaluation conference (pp. 374-384).
>
> [12] Huang, Y., Chen, Y., Yu, Z. and McKeown, K., 2022. In-context learning distillation: Transferring few-shot learning ability of pre-trained language models. arXiv preprint arXiv:2212.10670.
>
> [13] Cheng, J., Nandi, S., Natarajan, P. and Abd-Almageed, W., 2021. Sign: Spatial-information incorporated generative network for generalized zero-shot semantic segmentation. In Proceedings of the IEEE/CVF International Conference on Computer Vision (pp. 9556-9566).
>
> [14] Kim, J., Chang, S. and Kwak, N., 2021. PQK: model compression via pruning, quantization, and knowledge distillation. arXiv preprint arXiv:2106.14681.

---

> > ### Author Response · Authors · 2025-11-26
> > **Extra Rebuttal Response**
> >
> > **Q1 Extra Scale: Results for ImageNet**
> >
> > Our gradient level explanation of KD reveals that the negative asymmetric transfer we observe for KD is a fundamental property of the optimisation process, and we empirically affirm these results across datasets, architectures, and modalities.
> >
> > However, to validate these results at the scale you have requested below, we have added the results on the ImageNet dataset. **These results on ImageNet will be included in the appendix of the paper**.
> >
> > Here, the student is the ResNet-18 model, and the teacher is a pre-trained ResNet50 model taken from PyTorch with a top-1-accuracy of 80.858 and a top-5-accuracy of 95.434. Below, we present the functional results table, statistical significance, and prediction agreement deviation using a temperature of 1 for teacher seed 0.
> >
> > Functional Table:
> >
> > | | SIDDO | KD 0.1 | KD 0.5 | KD 0.9 | Rand KD 0.1| Rand KD 0.5| Rand KD 0.9|
> > |:-|:--|:--|:--|:--|:--|:--|:--|
> > | Activation Distance | 0.42 +- 0.001 | 0.365 +- 0.001 | 0.26 +- 0.001 | **0.226 +- 0.0**| 0.268 +- 0.001 | 0.259 +- 0.002 | 0.376 +- 0.0|
> > | Rank Disagreement| 0.997 +- 0.0| 0.997 +- 0.0| 0.997 +- 0.0| 0.997 +- 0.0| 0.997 +- 0.0| 0.997 +- 0.0| 0.997 +- 0.0|
> > | Prediction Disagreement | 0.264 +- 0.003 | 0.256 +- 0.002 | 0.239 +- 0.002 | **0.235 +- 0.002** | 0.259 +- 0.001 | 0.274 +- 0.002 | 0.308 +- 0.002 |
> > | JS Divergence| 0.26 +- 0.001 | 0.221 +- 0.001 | 0.136 +- 0.001 | 0.106 +- 0.0| 0.136 +- 0.001 | **0.099 +- 0.001** | 0.173 +- 0.001 |
> > | Accuracy| 0.68 +- 0.002 | 0.687 +- 0.002 | 0.7 +- 0.001| **0.703 +- 0.002** | 0.684 +- 0.001 | 0.67 +- 0.001 | 0.642 +- 0.002 |
> > | Loss| **1.307 +- 0.009** | 1.342 +- 0.009 | 1.608 +- 0.015 | 1.833 +- 0.022 | 1.657 +- 0.013 | 2.548 +- 0.017 | 4.06 +- 0.012 |
> >
> >
> > Significance  (✔️ corresponds to significant against controls and x is not significant):
> >
> > || **Activation Distance** | **Rank Disagreement** | **Prediction Disagreement** | **JS Divergence** | **Accuracy** | **Loss** |
> > |--|--|--|---|--|---|--|
> > | KD 0.1 | x | ✔️ | x | x | x| x|
> > | KD 0.5 | x | ✔️ | ✔️ | x | ✔️| x|
> > | KD 0.9 | ✔️ | ✔️ | ✔️ | x | ✔️| x|
> >
> >
> > Prediction Agreement:
> >
> > | | 0.1 | 0.5| 0.9|
> > |:-|:--|:-|:-|
> > | Correct | **0.4090 +- 0.1150**| 1.8950 +- 0.1040 | 2.2900 +- 0.1180 |
> > | Incorrect | -0.0020 +- 0.1710 | **2.1830 +- 0.1510** | **2.8290 +- 0.1590** |
> >
> > What we find here, in line with our existing results, is that when there is statistically significant knowledge transfer from the teacher to the student, then negative asymmetric transfer occurs via a bias towards teacher errors. These results were expected from the TinyImageNet results; however, we hope that the scale of these findings has reduced concerns around our use of an experimental setup that prioritises statistical significance.
> >
> > **Below we present the functional results table, statistical significance and prediction agreement deviation using a temperature of 2 for teacher seed 0.**
> >
> > Functional Table:
> >
> >
> > | | SIDDO| KD 0.1 | KD 0.5 | KD 0.9 | Rand KD 0.1| Rand KD 0.5| Rand KD 0.9|
> > |:----------|:-|:-|:-|:-|:-|:-|:-|
> > | Activation Distance | 0.42 +- 0.001| 0.31 +- 0.002| 0.251 +- 0.001 | **0.221 +- 0.001** | 0.305 +- 0.001 | 0.247 +- 0.001 | 0.28 +- 0.002|
> > | Rank Disagreement | 0.997 +- 0.0 | 0.997 +- 0.0 | 0.997 +- 0.0 | **0.996 +- 0.0** | 0.997 +- 0.0 | 0.997 +- 0.0 | 0.997 +- 0.0 |
> > | Prediction Disagreement | 0.264 +- 0.003 | **0.257 +- 0.002** | 0.258 +- 0.002 | 0.264 +- 0.002 | 0.259 +- 0.002 | 0.273 +- 0.002 | 0.311 +- 0.002 |
> > | JS Divergence | 0.26 +- 0.001| 0.16 +- 0.001| 0.101 +- 0.0 | **0.081 +- 0.0** | 0.152 +- 0.001 | 0.096 +- 0.0 | 0.115 +- 0.001 |
> > | Accuracy| 0.68 +- 0.002| **0.685 +- 0.001** | 0.684 +- 0.002 | 0.678 +- 0.001 | 0.684 +- 0.002 | 0.671 +- 0.001 | 0.64 +- 0.002|
> > | Loss| **1.307 +- 0.009** | 1.492 +- 0.014 | 1.725 +- 0.019 | 1.935 +- 0.019 | 1.533 +- 0.014 | 1.927 +- 0.019 | 3.016 +- 0.021 |
> >
> >
> > Significance  (✔️ corresponds to significant against controls and x is not significant):
> >
> > || Activation Distance | Rank Disagreement | Prediction Disagreement | JS Divergence | Accuracy | Loss |
> > |--|--|--|--|--|--|--|
> > | KD 0.1 | x | ✔️ | x | x | x| x|
> > | KD 0.5 | x | ✔️ | x | x | x| x|
> > | KD 0.9 | ✔️ | ✔️ | x | ✔️ | x| x|
> >
> > Prediction Agreement:
> >
> > | | 0.1| 0.5| 0.9 |
> > |:--|:--|:--|:--|
> > | Correct | 0.1880 +- 0.1090 | 0.1300 +- 0.1200 | -0.7390 +- 0.0980 |
> > | Incorrect | **0.2840 +- 0.1330** | **0.8230 +- 0.1730** | **0.5360 +- 0.1220**|
> >
> > Here we find that these results hold with our other temperature experiments. Increasing the temperature reduces the signal between the student and teacher, reducing functional similarity and negative transfer, and the overall utility of KD, while not removing the negative asymmetric transfer we uncover.
> >
> > We sincerely appreciate your review and feedback and would like to know if you have any remaining questions that our rebuttal has not answered, provided that we have shown our findings scale to ImageNet.

---

### Official Review · Reviewer_mT8m · 2025-10-31

**Soundness:** 2
**Presentation:** 2
**Contribution:** 2
**Rating:** 2
**Confidence:** 4

**Summary:**

Knowledge distillation has become a popular paradigm for compressing larger models into smaller models. This paper aims to improve our understanding of knowledge distillation through experiments in controlled settings, challenging the framework of knowledge distillation as a framework of knowledge transfer. The paper employs randomized control trials to study distillation across data modalities, i.e., the teacher replaced by noise that is subsequently fed to the student through the loss function. Then, they compute several alignment metrics between student and teacher, such as activation distance, JS divergence, prediction disagreement, etc., and also accuracy for these two sets of experimental setups. Experiments are performed across multiple experimental setups architectures, and datasets, aiming to show that knowledge distillation functions less as a compression mechanism and more as a data-dependent regulariser with an asymmetric payoff.

**Strengths:**

Strengths

-- Several experimental setups have been considered in the paper. Experiments span multiple datasets and architectures, using compute worth more than a thousand hours.

-- Fig. 1 is nice and clarifies the setup well.

-- The paper has also studied multiple modalities. The nuances of knowledge distillation in different modalities are interesting.

-- Adversarial transfer in language models, some mutual information-based measures, and distillation scaling laws have been included.

**Weaknesses:**

Weaknesses

-- While the extensive experiments are quite appreciated, I have some concerns/confusion about the main claim of the paper and the conclusions being drawn from the experiments. The terms "assymetric payoff" , "knowledge transfer", and "negative knowledge transfer" are quite confusing. Is it sufficient to contend that knowledge distillation does not transfer knowledge since the similarity measures do not increase? It is an optimization with two disagreeing terms anyway, and can be sensitive to the choice of hyperparameters. I do not fully understand the main argument about assymetric payoff and why it is not knowledge transfer.

-- The metrics used to compute "knowledge transfer" simply rely on the empirical alignment between teacher and student. It is unclear to me why similarity/alignment between teacher and student alone is a good measure of knowledge transfer since this doesn't consider anything specific about the task at hand. Separately, accuracy is only task-specific and doesn't consider both teacher and student together. Shouldn't an ideal measure of knowledge transfer capture what information about the task is transferred from teacher to student? Some information-theoretic metrics are introduced, but they are still analogous to either direct similarity/alignment or to accuracy. The measures are not about the task-specific alignment/similarity between teacher and student.

--  In setups where randomized control distillation has higher accuracy, how is the performance of the student without distillation at all (regularizer=0)? It seems the accuracies in these cases are still kind of close, e.g., 0.952, 0.954, and 0.957. Similarly, 0.605, 0.604, and 0.607. Could the small jump in randomized control distillation be from some small overfitting that is avoided?

-- Some of these settings seem to be cases where distillation itself might not make much difference (unless I am mistaken). What about setups where students with and without distillation show a big gap in accuracy? Please point me to it if already done.

-- How to know if the student has the full capacity as the teacher? What is the measure of student capacity?

**Questions:**

Q1. I am confused about the main claim. Why is asymmetric payoff wrong/surprising and not knowledge transfer?

Q2. Why is similarity between teacher and student a good measure of knowledge transfer? It will be interesting to consider measures that capture task-specific alignment/similarity rather than just similarity.

Q3. What about setups where students with and without distillation show a big gap in performance? Please point me to it if already done.

Q4: How to know if the student has the full capacity as the teacher? What is the measure of student capacity?

---

> ### Author Response · Authors · 2025-11-18
> **Response 1**
>
> **W1. The terms "assymetric payoff" , "knowledge transfer", and "negative knowledge transfer"**
>
> Thank you for acknowledging the extent of our experimental setup, which covers numerous architectures, datasets, and modalities. We conduct our experiments in this manner to confirm the universality of our findings and to prioritise the statistical significance of the key findings of the experiments.
>
> To further clarify the three terms that we use in the paper, we introduce a short subsection that formally defines each of the terms:
>
> **Knowledge transfer**: Occurs when the following empirical condition holds: Most similarity measures (e.g., activation distance, rank disagreement, JS divergence, Variation of Information, or Orthogonal Procrustes Distance) have statistically significantly decreased when comparing the student to the teacher against the baseline of RCD students to the teacher and SIDDO control models with the teacher. The decrease in these metrics signals an increased alignment between the student and the teacher under the application of knowledge distillation.
>
> If this criterion is met, then the agreement of the student and the teacher against the baselines can fit either of these three scenarios:  Δ_correct_agreement = Δ_incorrect_agreement (symmetric transfer), Δ_correct_agreement > Δ_incorrect_agreement (positive asymmetric transfer), or Δ_correct_agreement < Δ_incorrect_agreement  (negative asymmetric transfer).
>
> **Asymmetric payoff**:  Asymmetric knowledge transfer can occur when the prediction agreement between the student and the teacher against controls is unequal between correct and incorrect predictions.  We report together with the separate changes in correct-agreement (Δcorrect) and incorrect-agreement (Δincorrect) between teacher and student. I.e., Δ_correct_agreement > Δ_incorrect_agreement (positive asymmetric transfer) or Δ_correct_agreement < Δ_incorrect_agreement  (negative asymmetric transfer).
>
> **Negative transfer**: Denotes the regime in which both properties are observed simultaneously: (i) functional-similarity improves, but (ii) the rise in incorrect-agreement dominates the rise in correct-agreement, i.e., Δ_correct < Δ_incorrect). In other words, the student gains functional similarity yet absorbs proportionally more of the teacher’s mistakes than its correct knowledge.
>
> To further add to your point, knowledge transfer can be symmetric or asymmetric, meaning that hypothetically student models under KD can receive beneficial knowledge (correct) or harmful knowledge (incorrect) from the teacher either at equal or unequal rates.
> In this paper, we observe across 9 architectures, 7 datasets, and 3 modalities that KD statistically significantly and consistently produces an asymmetric payoff that is biased towards negative transfer. We record no cases where there is a knowledge transfer that is symmetric or any with a positive asymmetric payoff.
>
> Fundamentally, we argue that this evidence highlights that knowledge distillation is not an effective or beneficial form of compression and that it results in poor outcomes for student models and thus should not be used. In Section 5, we provide a gradient level analysis of KD to show that negative asymmetric pay-off is a feature and not a bug of the knowledge distillation optimisation process.
>
> **W2. Two disagreeing terms and hyperparameters sensitivity.**
>
> We agree that knowledge distillation is sensitive to hyperparameters, as such we control for this by spanning a range of alpha values (0.1,0.5,0.9), which determine the weighting of teacher information passed to the student.
> We find, typically, that as we increase the alpha value to make the student more reliant on the teacher that we see an increase in knowledge transfer, which corresponds directly with an increase in negative asymmetric transfer we identify as fundamentally explained in our gradient level analysis of knowledge distillation (Section 5). **Furthermore, please see our response to reviewer BrDG Q3 where we show how temperature also reaffirms our findings in this regard.**

---

> > ### Author Response · Authors · 2025-11-18
> > **Response 2**
> >
> > **W3. Similarity/alignment between teacher and student alone as a good measure of knowledge transfer."**
> >
> > Thank you for your comment.
> >
> > We agree that a metric to measure the knowledge transfer between a student and a teacher should capture the information that was transferred from the teacher to the student. That is why we employ the functional similarity measures, which compare how similar the teacher and students' outputs are for a particular dataset, in this case, the test dataset.  Therefore, these metrics are task-specific and reliant on the teachers' and students' outputs on a given dataset.
> >
> > In the main body of the  paper, we employ **4 task-based similarity metrics as defined on L183-188**, namely Activation Distance, Rank Disagreement, Prediction Disagreement, and Jensen-Shannon (JS) Divergence. Additionally, in Appendix A, we have an extended analysis of information-theoretic and geometric metrics, Variation of Information (VoI), an information-theoretic measure, and Orthogonal Procrustes Distance (OPD), which verify our findings with the metrics that we employ in the main paper.
> >
> > **These metrics allow for an analysis of student-teacher alignment in the predictive space over the top-1-label prediction as offered by accuracy.** This provides a highly nuanced insight into the agreement between students and teachers to a level of granularity previously unexplored in literature. It is this analysis that leads us to understanding the fundamental property of KD’s asymmetric negative transfer, which we verify in our gradient level theoretical analysis of the KD optimisation objective.
> > As you have correctly mentioned, this is a highly important and fundamental aspect of our study that we will provide further statement of clarification on in the main body of the paper to this effect, where we introduce the similarity measures we use.
> >
> > **W4. Close accuracies between SIDDO RCD and KD**
> >
> > You are correct; in the case you have selected, the accuracies are very close. While the RCD control group leads to a statistically significant increase over the baseline (SIDDO) and standard KD in accuracy, the boost is marginal. In this case, we also observe negligible knowledge transfer in the KD setting. Here, we argue that the use of KD does not sufficiently improve model performance and thus adds computational cost with no benefit.
> >
> > As to why the RCD leads to better performance, we would argue that the regularisation provided by RCD prevents overfitting and therefore allows the model to perform better without any requirement for knowledge transfer.
> >
> > **W5. Smaller students with accuracy differences.**
> >
> > Thank you for providing the opportunity to highlight our results in the paper that displays the impacts of knowledge distillation (KD) with lower-capacity students.
> >
> > In some of our self-distillation experiments, we see that the application of KD, while making a statistically significant increase in accuracy, is rather negligible. This concretely shows that the knowledge transfer of KD and its benefits are far less pronounced than anticipated if knowledge transfer is the mechanism of performance benefit.
> >
> > **In the appendix Section E “Knowledge Distillation To Smaller Student”, we present results for both the TinyImageNet and TinyShakespeare datasets.** For TinyImageNet, we use a ResNet-18 as the student and a ResNet-50; in this instance, the ResNet-18 SIDDO control reaches an accuracy of 50.30% when the teacher has an accuracy of 60.50% meaning a gap of 10.20% between student and teacher. Using knowledge distillation in this case provides the same findings; there is marginal knowledge transfer and no increase in test accuracy. The best performing model is observed in the RCD control with a statistically significant increase in accuracy.

---

> ### Author Response · Authors · 2025-11-18
> **Response 3**
>
> **W6. How to know if the student has the full capacity as the teacher?**
>
> Thank you for providing the opportunity to further explain our statement regarding student capacity. For our self-distillation experiments, the student model and the teacher model have the same architecture and initialisation. As a result, they have the same number of parameters and start at the same point in the loss landscape. Therefore, the only factor creating a difference between the student and the teacher model is the training data order. Here, the measure of capacity is the number of parameters and architecture, which is equivalent between the student and the teacher model.
>
> As a result, the student model and the teacher model can become the same model with the same representation, and if knowledge distillation is effective, then this is what would be expected. However, we often observe that this is not the case; when knowledge is transferred, it is an asymmetric payoff of negative information, which is still knowledge transfer but shows no positive benefit of the training setup.
>
> **Q1. Why is asymmetric payoff wrong/surprising and not knowledge transfer?**
>
> Please see the answer to W1 for definitions of asymmetric payoff.
>
> Asymmetric payoff, as defined in the answers above, is not inherently bad. If the asymmetric payoff was biased toward positive knowledge, this would be a good thing and would be the ideal case for knowledge distillation, allowing students to get major benefits from the knowledge distillation setup.
>
> However, we show repeatedly across architectures, datasets, and modalities that when there is significant knowledge transfer facilitated by knowledge distillation that the asymmetric payoff is biased towards negative information.
> This is surprising as knowledge distillation is primarily thought of as an effective compression mechanism that benefits student models.
>
> However, our foundational analysis of its mechanism shows that its use does not benefit students and that any accuracy benefits derived from its use can be simulated by our RCD control.
>
> **Furthermore, our theoretical Gradient Level Analysis of Asymmetric Transfer in Section 5 shows that this negative asymmetric payoff is fundamental to the knowledge distillation optimisation process and, therefore, cannot be avoided.**
>
> We show how knowledge transfer impacts student models when capacity is not a barrier, and then how this interacts when capacity is a barrier (i.e., having a smaller student model).  **We show that the results hold and that negative asymmetric knowledge transfer occurs regardless of size. In particular, see Section 4.5 and Appendix E, where the student does not have the same capacity as the teacher model as defined above, and the core findings still hold.**
>
> **Q2. Similarity  between teacher and student a good measure of knowledge transfer?**
>
> The term functional similarity is primarily based on its usage in [1], which provides an in-depth survey of the concept and outlines a range of metrics designed to assess the functional behaviour of models and how to compare the prediction outputs of two neural networks. This terminology has also been adopted in [2], which introduces Functional SAM as a way to improve SAM via functional similarity, and [3], which applies functional similarity to understand pruning strategies.
>
> While we acknowledge that this is still a developing area, the term functional similarity has gained traction in literature and captures the aim of our work to understand the predictive alignment between student and teacher models that is facilitated by knowledge distillation.
>
> Furthermore, these metrics do capture task-specific alignment as they capture how two models agree in their representation of data. As a result, these metrics compare how models' precision outputs agree/disagree in a way that is not and cannot be captured by metrics such as accuracy.
>
> In Appendix A, we have an extended analysis of information-theoretic and geometric metrics, Variation of Information (VoI), an information-theoretic measure, and Orthogonal Procrustes Distance (OPD), which verify our findings with the metrics that we employ in the main paper.
>
> Therefore, the functional similarity between the teacher and student is a good measure of knowledge transfer, as it measures how similar the functional representations of the teacher and student models are in output representation for a specific task on which they are not trained. This enables us to understand how knowledge transfer affects the representation of the model.
> In cases of knowledge transfer, we would expect to see that the models have increased functional alignment, as the models should represent similar functions, as would be induced with knowledge transfer. In these cases, where we observed increased functional alignment, we observed a strong negative asymmetric payoff where the teacher has shared their incorrect knowledge.

---

> ### Author Response · Authors · 2025-11-18
> **Response 4 (Final)**
>
> **Q3. Students with and without distillation show a big gap in performance?**
>
> Please see answer to W5 above.
>
> **Q4. How to know if the student has the full capacity as the teacher?**
>
> Please see answer to W6.
>
> **Overall comment**
>
> Thank you for seeking clarification. We will ensure that our paper reflects the discussion you have prompted to enable a better understanding of:
>
> 1. Terms of knowledge transfer, asymmetric transfer, and negative asymmetric transfer.
> 2. Capacity definitions that we base on architectures, number of parameters, and initialisation.
>
> If you have any further questions, we would be more than happy to answer them. Thank you again for your review.
>
>
>
> References:
>
> [1] Similarity of Neural Network Models: A Survey of Functional and Representational Measures. arXiv. 2024
>
> [2] Avoiding spurious sharpness minimization broadens applicability of SAM. ICML 2025
>
> [3]What makes a good prune? maximal unstructured pruning for maximal cosine similarity. ICLR 2024

---

### Author Response · Authors · 2025-11-27
**Reviewer Prompt**

We would like to thank all the reviewers for their evaluations of our paper. The questions asked have allowed us to strengthen our claims and elaborate on certain parts of the paper, and we have appreciated this.

We'd like to highlight some particularly useful outcomes of this rebuttal process:

1) Exploring temperature scales (1,2, and 4), showing that temperature reduces the level of knowledge transfer between teachers and students in line with our existing gradient level formalisation in Section 5, while also being unable to prevent negative asymmetric transfer when statistically significant transfer occurs.

2) We have shown that our findings of negative asymmetric transfer scale to ImageNet settings - further reinforcing that negative asymmetric transfer is a core component of the KD optimisation process and is not architecture, dataset, or modality dependant.

3) Highlighting that our work resolves fundamental tensions in literature between KD being a reliable knowledge transfer mechanism and a regulariser. Our paper shows that both perspectives are true, in part; however, when KD does statistically significantly transfer knowledge to the student that a severe negative asymmetric transfer occurs, transferring the teacher's errors to the student, which inspires safety considerations as we highlight in the main paper.

We acknowledge that this period is particularly busy; however, we would like to request that you consider our in-depth responses such that we can ensure we have addressed all of your concerns and can further elaborate if required.

The feedback from your reviews will be integrated into the revised manuscript once we have confirmation on the above. We have chosen to do this to ensure our edits reflect the latest feedback and are in line with your requests.

Thank you again for your efforts in the review process.

---

### Author Response · Authors · 2025-12-03
**Revised Paper Upload Author Comments (2/2)**

**Overall**

Across our extensive and diverse experimental space, we show that whenever KD yields statistically significant functional similarity, it does so with a negative asymmetric payoff: the student aligns more strongly with the teacher’s incorrect predictions.

This novel finding is persistent, systematic and robust: it appears in 1) self-distillation, 2) standard KD with smaller students, 3) feature-map matching variants of KD, 4) remains under temperature scaling, 5) is amplified as student capacity increases, 6) directly facilitates targeted adversarial transfer to student models, and 7) persists at ImageNet scale.

Beyond breadth, we believe the depth and novelty of the paper justify serious consideration: We present Random Control Distillation (RCD), similar to label smoothing, as a principled counterfactual to standard KD, precisely elucidating the impacts of KD in student teacher functional alignment. Starting from the usual objective, $L = (1 - \alpha) \cdot \mathcal{H}(y, \sigma(z^{(s)})) + \alpha \cdot \text{KL}(\sigma(z^{(t)}), \sigma(z^{(s)})),$  the gradient w.r.t. a student logit $z^{(s)}_k$ is  $\frac{\partial L}{\partial z^{(s)}_k} = (1 - \alpha)(p^{(s)}_k - y_k) + \alpha(p^{(s)}_k - p^{(t)}_k),$.  For any incorrect class k ($y_k = 0$), this reduces to $\frac{\partial L}{\partial z^{(s)}_k} = \alpha(p^{(s)}_k - p^{(t)}_k)$, so any non-zero teacher mass $p^{(t)}_k$ on a wrong class induces a directed pull of the student towards that specific error. Because teacher errors are structured, this yields the empirically observed negative asymmetric transfer: student–teacher alignment grows faster on incorrect than on correct predictions.

In RCD, we retain the same loss but replace $p^{(t)}$ with a random uniform distribution $\mathcal{U}$, so the gradient on incorrect classes becomes  $\frac{\partial L}{\partial z^{(s)}_k} = \alpha(p^{(s)}_k - u_k)$, which acts as an unstructured smoothing term with no alignment to the teacher's mistakes. Empirically, RCD often matches or outperforms KD on accuracy and loss, showing that much of KD's benefit is generic regularisation, while the genuine distillation component preferentially propagates the teacher's errors. This goes beyond prior claims that KD is a regulariser by (i) decomposing functional alignment into correct vs. incorrect agreement and showing, across modalities and scales, that KD's regularisation has a systematically negative asymmetric payoff; and (ii) providing a gradient-level mechanism plus a matched counterfactual (RCD) that use the same objective form, isolating exactly which part of KD's signal is helpful (smoothing) and which part is harmful (error-aligned transfer).

We provide a mechanistic, gradient-level explanation for negative asymmetric transfer. Starting from the standard KD loss, $L = (1 - \alpha) \cdot \mathcal{H}(y, \sigma(z^{(s)})) + \alpha \cdot \text{KL}(\sigma(z^{(t)}), \sigma(z^{(s)})),$  the gradient w.r.t. a student logit $z^{(s)}_k$ is  $\frac{\partial L}{\partial z^{(s)}_k} = (1 - \alpha)(p^{(s)}_k - y_k) + \alpha(p^{(s)}_k - p^{(t)}_k)$.  For any incorrect class k ($y_k = 0$), this reduces to $\frac{\partial L}{\partial z^{(s)}_k} = \alpha(p^{(s)}_k - p^{(t)}_k)$, so any non-zero teacher mass $p^{(t)}_k$ on a wrong class induces a directed pull of the student towards that specific error. This shows that error amplification and negative asymmetric transfer are fundamental properties of the commonly used KD objectives, rather than artefacts of particular architectures or hyperparameters.

We demonstrate targeted adversarial transfer: KD can reliably induce a specific erroneous behaviour while preserving seemingly strong aggregate performance; for example, systematically preferring "tha" over "the" in our language experiments. Crucially, this mechanism is not language-specific: it arises from the KD objective itself, and therefore provides a general pathway for silently propagating structured harmful behaviours across domains and modalities.

We connect our empirical findings to distillation scaling laws, showing that increasing student capacity unlocks more of the distillation signal, but this predominantly manifests as amplified error alignment, not just improved generalisation.
In the current landscape, where KD is routinely used to compress large language models and deploy them in safety-relevant settings, our results highlight a concrete and previously unidentified risk: KD can systematically and silently transfer a teacher's hidden biases and erroneous behaviours to the student. Our work therefore does not “just” recharacterise KD as a data-dependent regulariser. Rather, it shows that when KD does function as a transfer mechanism, it often does so in the wrong direction.
We believe our work clearly demonstrates both the scope and significance of our claims and findings, and we respectfully ask the AC to reconsider the paper on the basis of its contributions and empirical depth.

---

### Author Response · Authors · 2025-12-03
**Revised Paper Upload Author Comments (1/2)**

We thank the reviewers for their comments and questions.  We have uploaded the revised PDF which has integrated all suggestions provided by the reviewers. In the PDF, new edits are presented in Blue.

**We have added the following content to the paper in order of appearance**

1. In response to reviewer mT8m (W.2), BrDg (Q3) and Odu5 (W.4) comments around temperature, hyperparameter sensitivity and teacher confidence, we add six additional experiments (making 18 experimental setups in total) in L027 running across temperature hyperparameters (2 and 4) and show that temperature reduces knowledge transfer in KD while being unable to mitigate the asymmetric knowledge transfer we identify in both language and vision domains.

Please refer to Appendix E.2.1 and E.3.1 for the full temperature results and analysis.

2. To the request from BrDg (Q1) and Odu5 regarding the large scale dataset ImageNet, on L027 we show that we increase our number of datasets studied from 7 to 8 to show that we have included ImageNet results which confirm the same core finding we observe: when there is statically significant transfer, a negative asymmetric payoff arises in KD, with students preferentially inheriting the teacher’s errors. Please refer to Appendix E.2 to see ImageNet results that corroborate all of our findings.

3. To Reviewer mT8m W.3, in L190-191 we elaborate that our metrics function as task-based measures of teacher and student representational similarity and alignment.

4. For reviewer mT8m and their comments regarding the knowledge transfer and asymmetric payoff definitions, we have now included a new subsection ‘3.2 KNOWLEDGE TRANSFER DEFINITIONS’ which precisely describes the terms used in this paper: Knowledge Transfer, Asymmetric Transfer and (most importantly) Negative Asymmetric Transfer.

5. To respond to reviewers  BrDg (W1) and mT8m (W.3), in L426-429 we clearly distinguish our work from prior claims that knowledge distillation is just a data-based regularisation method, and instead provide a novel finding and nuanced understanding that, when KD operates as a knowledge transfer mechanism, the transfer is systematically negative: the student learns more incorrect than correct behaviour from the teacher. .

In L530-537 of the conclusion, we further highlight the novel, important and timely contribution this paper makes beyond considering KD simply as a regulariser.

6. In response to reviewer BrDg Q2 regarding the removal of our novel identification of fundamental negative asymmetric transfer facilitated by KD in L516-517, we note that we already provide a concise theoretical explanation for the observed asymmetric error transfer. Our findings are a direct consequence of the KD objective itself. Our gradient-level analysis shows that this behaviour is structurally baked into the loss and cannot be eliminated.

---

### Meta-Review · Area_Chair_AFCr · 2026-01-12

**Summary:**

Reviewers found the topic timely and the experimental effort substantial (multi-modal, many architectures/datasets, heavy compute), and they generally agree the paper is trying to probe what KD is doing beyond accuracy/loss. However, the overall reception was negative because:
- the key terminology/claims (knowledge transfer, negative transfer, asymmetric payoff) were initially unclear and made the main conclusion feel shaky
- some reviewers felt the paper’s narrative substantially overstates novelty and implications (e.g., KD-as-regularization is already established; backdoor/error transfer concerns are not clearly novel)
- there were concerns that the evidence and metrics mainly demonstrate student–teacher alignment patterns rather than a definitive statement about whether KD transfers knowledge in a task-relevant sense.
- One reviewer was closer to the margin (found the 'negative asymmetric error transfer' interesting) but still raised concerns about novelty, scaling to larger models, and lack of concrete mitigation/prevention analysis.

**Reviewer Concerns:**

Addressed (some partially):

- Clarity of definitions / terminology: the authors explicitly define the introduced terms (new Section 3.2) and try to align the paper’s language with those definitions.
- Scale concerns (ImageNet request): a reviewer explicitly requested validation beyond small datasets; authors added ImageNet results in the revision/appendix and stated that the same qualitative finding holds.
- Hyperparameter sensitivity (temperature): reviewers asked about temperature and teacher confidence effects; authors report new temperature experiments and claim temperature reduces transfer signal but does not remove the negative asymmetric transfer when transfer is significant.

Still outstanding:

- Novelty + framing. Even with the added mechanistic explanation, at least one reviewer’s core objection is that several claims feel exaggerated and/or too close to known narratives: KD-as-regularization/label smoothing connections and 'learning from incorrect predictions' being a known part of KD discussions, with existing practical methods that modulate this. The rebuttal argues 'comprehensiveness' and 'systematic negative asymmetry', but not very convincing that the contribution is truly beyond prior work.

- Are the metrics the right lens for 'knowledge transfer'? One reviewer’s key discomfort was that the paper equates 'knowledge transfer' with improvements in similarity/alignment metrics, and asks for more task-specific notions of what is transferred. The rebuttal clarifies intent, but it doesn’t fully address the concern that this is partly a definitional/measurement choice that can be debated.

- The 'safety implications' and 'audit the teacher' message is directionally sensible, but reviewers asked (implicitly or explicitly) for deeper analysis of why this happens and how to prevent it (beyond saying it is 'baked into the objective').

- Generalization to large teacher-student LMs. Even with added ImageNet, one reviewer explicitly worried about extrapolating to 'huge teacher/student language models'; the revision does not close that gap.

**Reviewer Scores:**

- mT8m: 2 to possibly 4
- BrDg: 4 to possibly 6
- oDU5: 2 to remain at 2 or at most increase to 4

---

### Decision · Program_Chairs · 2026-01-26

Reject